# ON THE MECHANISM AND DYNAMICS OF MODULAR ADDITION: FOURIER FEATURES, LOTTERY TICKET, AND GROKKING

## ABSTRACT

We present a comprehensive analysis of how two-layer neural networks learn features to solve the modular addition task. Our work provides a full mechanistic interpretation of the learned model and a theoretical explanation of its training dynamics. First, we empirically show that trained networks learn a sparse Fourier representation; each neuron's parameters form a trigonometric pattern corresponding to a single frequency. We identify two key structural properties: *phase alignment*, where a neuron's output phase is twice its input phase, and *model symmetry*, where phases are uniformly distributed among neurons sharing the same frequency, particularly when overparametrized. We prove that these properties allow the network to collectively approximate an indicator function on the correct logic for the modular addition task. While individual neurons produce noisy signals, the phase symmetry enables a *majority-voting scheme* that cancels out noise, allowing the network to robustly identify the correct sum. We then explain how these features are learned through a "*lottery ticket mechanism*". An analysis of the gradient flow reveals that frequencies compete within each neuron during training. The winning frequency that ultimately dominates is predictably determined by its initial magnitude and phase misalignment. Finally, we use these insights to demystify grokking, characterizing it as a three-stage process involving memorization followed by two generalization phases driven by feature sparsification.

## 1 INTRODUCTION

A central mystery in deep learning is how neural networks learn to generalize. While these models are trained to find patterns in data, the precise way they build internal representations through gradient-based training and make predictions on new, unseen data is not fully understood. The sheer complexity of modern networks often obscures the fundamental principles at work. To gain a clearer view, researchers often simplify the problem by studying how networks solve simple but rich tasks that can be precisely analyzed. By meticulously analyzing the learning process in these controlled "toy" settings, we can uncover basic mechanisms that may apply more broadly. The modular addition task, $(x, y) \mapsto (x + y) \pmod{p}$ has emerged as a canonical problem for this approach, as it is simple to define yet reveals surprisingly complex and insightful learning dynamics.

Prior work has established that neural networks trained on modular arithmetic discover a *Fourier feature* representation, embedding inputs onto a circle to transform addition into geometric rotation (Nanda et al., 2023; Zhong et al., 2023). These studies have also highlighted the intriguing *grokking* phenomenon, where a model suddenly generalizes long after it has memorized the training data (Power et al., 2022; Liu et al., 2022). While these observations are foundational, prior work has not yet offered a conclusive, end-to-end explanation of the learning process. Existing theoretical accounts often rely on mean-field approximations (Tian, 2024; Wang & Wang, 2025) or analyze non-standard loss functions (Morwani et al., 2023), leaving a gap in our understanding of the finite-neuron dynamics under standard training. This leaves three fundamental questions unanswered:

(i) **Mechanistic Interpretability:** How does the trained network leverage its learned Fourier features to implement the modular addition algorithm precisely?

(ii) **Training Dynamics:** How do these specific Fourier features reliably emerge from gradient-based training with random initialization?

(iii) **Grokking:** How do these mechanisms and dynamics explain the full timeline of grokking, from memorization to delayed generalization?

In this paper, we provide a comprehensive answer to these questions through extensive experiments and rigorous theoretical analysis of a two-layer neural network. First, we empirically demonstrate that trained networks learn a sparse Fourier representation characterized by two key structural properties: *phase alignment*, where a neuron's output phase is twice its input phase, and *phase symmetry*, where phases are uniformly distributed among neurons sharing the same frequency. Mechanistically, we prove that these properties enable the network to collectively approximate a *indicator function*. While individual neurons produce noisy signals, the phase symmetry facilitates a majority-voting scheme that cancels out noise, allowing the network to identify the correct sum robustly.

Second, we explain how these features are learned via a *lottery ticket mechanism*. An analysis of the gradient flow reveals that different frequencies compete within each neuron during training. We prove that the winning frequency that ultimately dominates is predictably determined by its initial conditions: the one with the largest initial magnitude and smallest phase misalignment grows much faster than its competitors. This gives a complete explanation for the emergence of single-frequency. Finally, armed with this mechanistic and dynamic understanding, we demystify *grokking*. We characterize it as a three-stage process: an initial memorization phase, followed by two distinct generalization phases driven by feature sparsification and refinement under weight decay. Our analysis also uncovers a *common-to-rare* memorization pattern, where the model prioritizes common training examples over rarer ones. By providing a complete, end-to-end theoretical and empirical account of this learning problem, our work offers a concrete foundation for understanding the interplay between feature learning, training dynamics, and generalization in neural networks.

## 1.1 RELATED WORK

**Modular Addition and Grokking.**   Studying simple tasks like modular addition has revealed deep insights into neural network mechanisms (e.g., Power et al., 2022). Reverse-engineering has shown models learn a Fourier feature, converting addition into a geometric rotation by embedding numbers on a circle (Nanda et al., 2023; Zhong et al., 2023; Gromov, 2023; Doshi et al., 2024; Yip et al., 2024; McCracken et al., 2025). This discovery is central to understanding grokking, a phenomenon where generalization suddenly emerges long after overfitting, which these papers study using specific train-test data splits (e.g., Liu et al., 2022; Doshi et al., 2023; Yip et al., 2024; Mallinar et al., 2024; Wu et al., 2025). A complete discussion on related works is deferred to §A.2 due to space limit.

## 2 PRELIMINARIES

**Modular Addition.**   In a *modular addition* task, we aim to learn the teacher model $\mathbb{Z}_p \times \mathbb{Z}_p \mapsto \mathbb{Z}_p$, whose form is given by $(x, y) \mapsto (x + y) \bmod p$. The complete dataset is given by $\mathcal{D}_{\text{full}} = \{(x, y, z) \mid x, y \in \mathbb{Z}_p, z = (x + y) \bmod p\}$ which consists of all possible input pairs $(x, y)$ and their corresponding modular sums $z$. This dataset is then partitioned into a training set for learning and a disjoint test set for evaluation. Such a training setup is widely used in the literature (e.g., Nanda et al., 2023; Morwani et al., 2023) in modular arithmetic tasks.

**Two-Layer Neural Network.**   We consider a two-layer neural network with $M$ hidden neurons and no bias terms. Each input pair $(x, y)$ is assigned to embedding vectors $h_x$ and $h_y$ in $\mathbb{R}^d$, where $h : \mathbb{Z}_p \mapsto \mathbb{R}^d$ is an embedding function of dimension $d \in \mathbb{N}$. Here, the embedding can be either the canonical embedding $e_x \in \mathbb{R}^p$ in which case $d = p$ or a trainable one $\{h_x\}_{x \in \mathbb{Z}_p} \subseteq \mathbb{R}^d$. Let $\theta = \{\theta_m\}_{m \in [M]}$ and $\xi = \{\xi_m\}_{m \in [M]}$ denote the parameters, where $\theta_m \in \mathbb{R}^d$ is the parameter vector of the $m$-th hidden neuron and $\xi_m \in \mathbb{R}^p$ is its corresponding output-layer weight. The network output follows

$$f(x, y; \xi, \theta) = \sum_{m=1}^{M} \xi_m \cdot \sigma(\langle h_x + h_y, \theta_m \rangle) \in \mathbb{R}^p, \tag{2.1}$$

where $\sigma(\cdot)$ is a nonlinear activation. In this paper, we primarily focus on the ReLU activation $\sigma(x) = \max\{x, 0\}$ for experiments and the quadratic activation $\sigma(x) = x^2$ for theoretical interpretations. Since the modular addition is essentially a classification problem, we apply the softmax function $\texttt{smax} : \mathbb{R}^d \mapsto \mathbb{R}^d$ to the network output and consider the cross-entropy (CE) loss:

$$\ell_{\mathcal{D}}(\xi, \theta) = - \sum_{(x,y) \in \mathcal{D}} \big\langle \log \circ \texttt{smax} \circ f(x, y; \xi, \theta), e_{(x+y) \bmod p} \big\rangle. \tag{2.2}$$

Here, $\log(\cdot)$ is applied entry-wise and $e_{(x+y) \bmod p}$ is the one-hot vector that corresponds to the correct label. Intuitively, each input pair $(x, y)$ is mapped to a hidden representation by $\sigma(\langle h_x + h_y, \theta_m \rangle)$ for each neuron $m$, then linearly combined by $\xi_m$'s to produce the logits $f(x, y; \xi, \theta)$, and finally processed via softmax function to yield a categorical distribution for classification.

## 3 EMPIRICAL FINDINGS

In this section, we set $p = 23$ and use a two-layer neural network with width $M = 512$ and ReLU activation. The network is trained using the AdamW optimizer with a constant step size of $\eta = 10^{-4}$. For stable training, we initialize all parameters using PyTorch's default method (Paszke et al., 2019), and then normalize. We use the CE loss averaged over the dataset.

Following prior work (Morwani et al., 2023; Tian, 2024), we primarily focus on training the model with the complete dataset $\mathcal{D}_{\text{full}}$ (without train-test splitting), as this yields more stable training dynamics and enhances model interpretability. While the train-test split setup exhibits the intriguing grokking phenomenon (e.g., Nanda et al., 2023; Doshi et al., 2023; Gromov, 2023)—wherein models suddenly achieve generalization after extensive training despite initial overfitting—we defer this analysis to §3.2, building upon the foundational results presented in subsequent sections.

### 3.1 EXPERIMENTAL OBSERVATIONS ON LEARNED WEIGHTS

We first summarize the main empirical findings of our experiments using ReLU activation (see Figures 7 and 1), formalized as four key observations. The first two—*trigonometric parameterization* and *phase alignment*—have been previously explored in the literature (Gromov, 2023; Nanda et al., 2023; Yip et al., 2024), and are included for completeness. For clarity, we focus on the case where inputs are one-hot embedded, i.e., $h_x = e_x \in \mathbb{R}^p$ and $\theta_m, \xi_m \in \mathbb{R}^p$. We begin with the most striking observation: a global *trigonometric pattern* in parameters that consistently emerges across all training runs with random initialization.

> **Observation 1 (Fourier Feature).** There exists a frequency mapping $\varphi : [M] \to [\frac{p-1}{2}]$, along with magnitudes $\alpha_m, \beta_m \in \mathbb{R}^+$ and phases $\phi_m, \psi_m \in [-\pi, \pi)$, such that
>
> $$\theta_m[j] = \alpha_m \cdot \cos(\omega_{\varphi(m)} j + \phi_m), \ \xi_m[j] = \beta_m \cdot \cos(\omega_{\varphi(m)} j + \psi_m), \ \forall (m, j) \in [M] \times [p], \quad (3.1)$$
>
> where we denote $\omega_k = 2\pi k/p$ for all $k \in [(p-1)/2]$.

This observation shows that the parameter vectors $\theta_m$ and $\xi_m$ simplify during training into a clean trigonometric pattern. In the frequency domain, this corresponds to a *sparse signal*. After applying a Discrete Fourier Transform (DFT, see §A.3), each neuron is represented by a *single active frequency* $\varphi(m)$. Given this single-frequency structure, we will henceforth refer to $\alpha_m$ and $\phi_m$ as the **input magnitude** and **phase**, and to $\beta_m$ and $\psi_m$ the **output magnitude** and **phase** for neuron $m$.

In Figure 15b, we zoom in on the learned parameters of the first three neurons, with each entry corresponding to the input or output value $j$. The plots show that these parameters are well approximated by cosine curves, shifted by phases $\phi_m, \psi_m$, and scaled by magnitudes $\alpha_m, \beta_m$. This suggests that the trained neural network learns to solve modular addition by embedding a trigonometric structure into its parameters. We further examine the local structure of individual neurons, and observe a highly structured *phase alignment* behavior.

> **Observation 2 (Doubled Phase).** For each neuron $m \in [M]$, the parameter exhibits a doubled phase relationship, where the output phase is twice the input phase, i.e., $(2\phi_m - \psi_m) \bmod 2\pi = 0$.

We visualize the relationship between $\phi_m$ and $\psi_m$ in Figure 16a. Specifically, the dots represent the pairs $(2\phi_m, \psi_m)$, which lie precisely on the line $y = x$, confirming the claim made in Observation 2. This indicates that the first-layer $\theta_m$ and second-layer $\xi_m$ learns to couple in the feature space, specifically the Fourier space, through training. Having studied both global and neuron-wise local parameter patterns, we now examine how neurons coordinate their collective operation. Consider a network with a sufficiently large number of neurons, then the phases exhibit clear *within-group uniformity* and the magnitudes display nearly *homogeneous scaling* across neurons.

> **Observation 3 (Model Symmetry).** Let $\mathcal{N}_k$ be the set of neurons for frequency $k$, defined as $\mathcal{N}_k = \{m \in [M] : \varphi(m) = k\}$. For large $M$, (i) phases are approximately uniform over $(-\pi, \pi)$ within frequency group $\mathcal{N}_k$, i.e., $\phi_m, \psi_m \overset{\text{i.i.d.}}{\sim} \text{Unif}(-\pi, \pi)$, (ii) every frequency $k$ is represented among the neurons, and (iii) the magnitudes $\alpha_m$'s and $\beta_m$ remains close across all neurons.

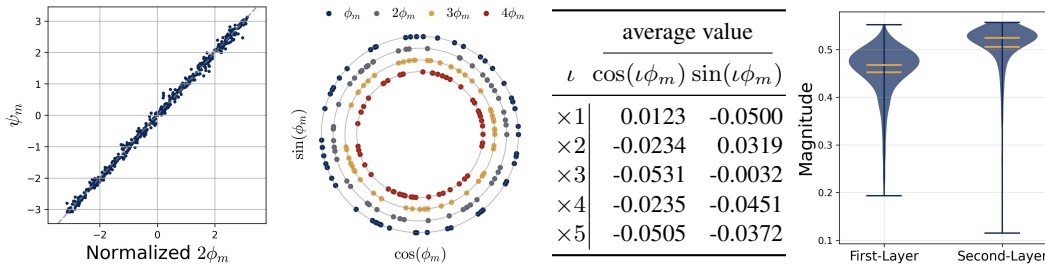

(a) Scatter of $(2\phi_m, \psi_m)$.   (b) Phase Symmetry within Frequency Group $\mathcal{N}_k$.   (c) Distribution of $\alpha_m, \beta_m$.

Figure 1: Visualizations of learned phases with $M = 512$ neurons. Figure (a) plots the relationship between the normalized $2\phi_m$ and $\psi_m$, with all points lying around $y = x$. Figure (b) shows the uniformity of the learned phases within $\mathcal{N}_k$. The right panel quantifies this symmetry by computing the averages of $\cos(\iota\phi_m)$ and $\sin(\iota\phi_m)$, all of which are close to zero. Figure (c) presents violin plots of the magnitudes $\alpha_m$ and $\beta_m$, suggesting that the neurons learn nearly identical magnitudes.

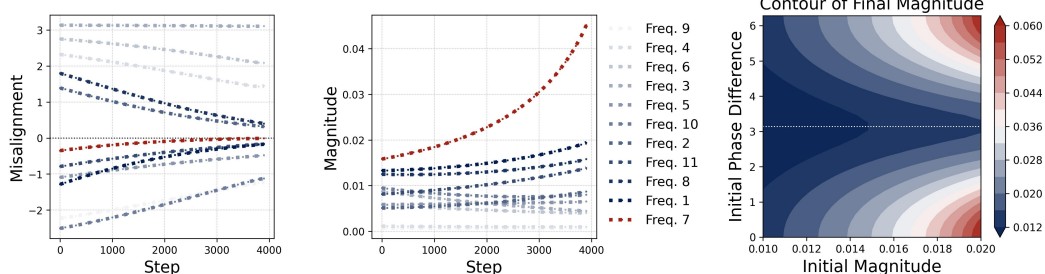

(a) Evolution of misalignment level $\mathcal{D}_m^k$ and magnitude $\beta_m^k$ of a specific neuron under gradient flow under small random initialization.   (b) Leaned magnitude $\beta_m^k$ under different initializations.

Figure 2: Illustration of the lottery-ticket mechanism. Figure (a) plots the dynamics of every frequency $k$ for a specific neuron, with the red curve tracing the trajectory of the frequency that eventually dominates. In the left-hand plot, misalignment levels $\mathcal{D}_m^k$ are rescaled to $[-\pi, \pi]$ for clarity. Figure (b) plots the contour of the magnitude $\beta_m^k$ with various $(\beta_m^k(0), \mathcal{D}_m^k(0))$ after $10,000$ steps.

Figure 16b illustrates the uniformity of phases within a specific frequency group $\mathcal{N}_k$ by examining the higher-order symmetry. In addition, the learned magnitudes are close to each other for the majority of the neurons, and no single neuron dominates (see Figure 16c). While previous work, notably Kumar et al. (2024), has introduced the concept of phase uniformity to provide a constructive model that solves modular addition, our findings significantly refine the understanding. Through empirical validation, we show that this phase uniformity is a consistent when $M$ is large. Furthermore, in §4, we derive and utilize a substantially weaker condition than strict uniformity to enable a more precise, joint analysis of noise cancellation across a diversified, finite set of neurons. Finally, we report a surprising *adaptivity* in the learned parametrization: the network continues to perform perfectly when ReLU is replaced with a broad class of alternative activations.

> **Observation 4 (Robustness to Activation Swapping).** A model trained with ReLU is robust to changes of activation function at inference time. This is because learning a good solution only relies on the activation's dominant *even-order components*. Consequently, functions with strong even components, such as the absolute value and quadratic, can be used interchangeably after training, all while maintaining perfect accuracy with a negligible change in loss.

Table 1 provides the empirical support. Hence, we can analyze the mechanism of the learned model or, furthermore, the training dynamics using more analytically tractable activations.

### 3.1.1 DYNAMICAL PERSPECTIVE: PHASE ALIGNMENT AND FEATURE EMERGENCE

We conduct an analysis of training dynamics in an analytically tractable setting, using quadratic activation with small random initialization, and focus on the early stages of training. Motivated by Observation 1, our analysis hinges on studying the training dynamics within the frequency domain. To do this, we use the Discrete Fourier Transform (DFT), which is formally discussed in §A.3, to decompose the model's parameters. Without loss of generality, any random initial parameter

vector can be exactly represented by its frequency components—magnitudes $(\alpha_m^k, \beta_m^k)$'s and phases $(\phi_m^k, \psi_m^k)$'s. This allows us to express the parameters as

$$\theta_m[j] = \alpha_m^0 + \sum_{k=1}^{(p-1)/2} \alpha_m^k \cdot \cos(\omega_k j + \phi_m^k), \ \xi_m[j] = \beta_m^0 + \sum_{k=1}^{(p-1)/2} \beta_m^k \cdot \cos(\omega_k j + \psi_m^k), \ \forall j \in [p], \quad (3.2)$$

Note that, under small initialization, the neurons and frequencies are *fully decoupled*. This results in the *parallel growth* of the magnitudes and phases for each neuron-frequency pair $(m, k)$. The central question is how the training process evolves this **complex, multi-frequency initial state** into the **simple, single-frequency pattern** observed at the end of training. Our finding is surprising:

*The final, dominant frequency learned by each neuron is entirely predictable from*
*a small subset of Fourier components in its initial parameters.*

It arises from a competitive dynamics among frequencies, as illustrated in Figure 2a. A frequency's success is determined by its initial conditions, primarily two key factors: its initial magnitudes and its initial phase misalignment level. To gain a more detailed understanding of the dynamics, we begin by tracking the evolution of phases. Motivated by the double phase phenomenon in Observation 2, we monitor the normalized phase difference $\mathcal{D}_m^k$, defined as $\mathcal{D}_m^k = (2\phi_m^k - \psi_m^k) \mod 2\pi \in [0, 2\pi)$. In the left-hand side of Figure 2a, we plot the dynamics of this phase difference, rescaling its range to $(-\pi, \pi]$ for visual clarity. This analysis leads to the following observation.

**Observation 5 (Phase-Aligning Dynamics).** The phase difference $\mathcal{D}_m^k(t)$ for each frequency converges monotonically to "zero" without crossing the axis. Generally, frequencies that start with an initial phase difference $\mathcal{D}_m^k(0)$ closer to zero converge faster.

To formalize the closeness of phase difference to zero, we define the phase misalignment $\widetilde{\mathcal{D}}_m^k$ as $\widetilde{\mathcal{D}}_m^k = \max\{\mathcal{D}_m^k, 2\pi - \mathcal{D}_m^k\}$. In the following, we outline the core dynamics of the training process. It reveals that the single-frequency pattern in Observation 1 is the direct result of a frequency competition, a process governed by the interplay of phase misalignment and magnitude.

**Observation 6 (Lottery Ticket Mechanism).** Under small initialization, neurons are decoupled. Each frequency $k$ draws a "lottery ticket" specified by its initial magnitudes $\alpha_m^k(0)$, $\beta_m^k(0)$ and its misalignment level $\widetilde{\mathcal{D}}_m^k(0)$. All frequencies grow in parallel, and the one with *the largest* $\alpha_m^k(0)$ *and* $\beta_m^k(0)$ and *the smallest* $\widetilde{\mathcal{D}}_m^k(0)$ ultimately wins—dominating the feature of specific neuron—due to the rapid acceleration once magnitudes become larger and $\widetilde{\mathcal{D}}_m^k(t)$ reaches zero.

Figure 2a provides a clear empirical illustration of the mechanism. The winning frequency, highlighted in red, begins with a highly advantageous initialization—a competitively large magnitude and a misalignment value close to zero. While other frequencies exhibit slow growth, the holder of this winning ticket undergoes a distinct phase of rapid, exponential acceleration in its magnitude. Figure 2b plots the magnitude under different initializations after a fixed time $t = 10$, verifying that frequencies with a larger magnitude and a smaller misalignment take advantage.

### 3.2 GROKKING: FROM MEMORIZATION TO GENERALIZATION

In this section, we provide empirical insights into grokking by analyzing the model's training dynamics using a progress measure designed based on our prior observations. Prior work, such as Nanda et al. (2023), identifies two key factors for inducing grokking: a distinct train-test data split and the application of weight decay. We randomly partition the entire dataset of $p^2$ points, using a training fraction of 0.75, and apply a weight decay of 1.0. As shown in Figure 3a, this elicits a clear grokking: the training loss drops quickly to zero. In contrast, the test loss initially remains high before gradually decreasing, signaling a delayed generalization. We track four key progress measures: (i) **train-test loss** and **accuracy**, to measure memorization and generalization; (ii) **phase difference** $|\sin(\mathcal{D}_m^\star)|$, where $\mathcal{D}_m^\star := 2\phi_m^\star - \psi_m^\star \mod 2\pi$, to track layer-wise phase alignment; (iii) **frequency sparsity**, measured by inverse participation ratio (IPR), defined as $\text{IPR}(\nu) = (\|\nu\|_{2r}/\|\nu\|_2)^{2r}$ with $r = 2$, to capture the single-frequency emergence of Fourier coefficients; and (iv) $\ell_2$**-norm** of parameter, which serves as a proxy for the effect of weight decay.

Building upon Figure 3, we identify two primary driving forces of the dynamics: *loss minimization* and *weight decay*. These forces guide the training process through an initial memorization phase followed by two generalization stages.

The memorization phase is dominated by loss minimization, causing the model to fit the training data with its parameter norms increasing rapidly. As a result, the model achieves perfect accuracy

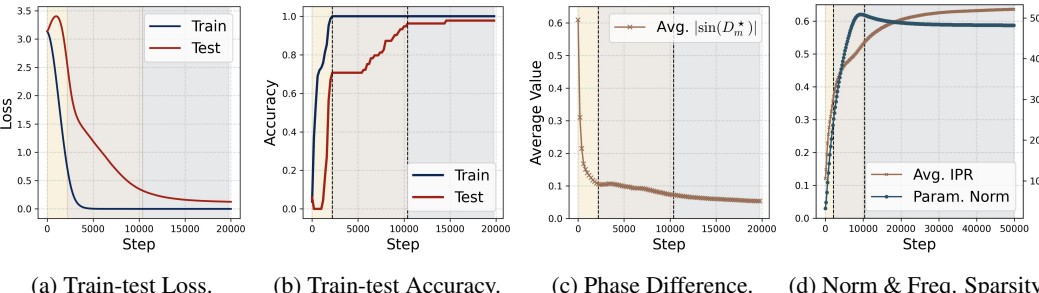

(a) Train-test Loss.  (b) Train-test Accuracy.  (c) Phase Difference.  (d) Norm & Freq. Sparsity.

Figure 3: Progress measure of grokking behavior. The shaded regions mark three distinct phases: an initial memorization phase. Figures (a) and (b) plot the train-test loss and accuracy curve, where the network first overfits the training data while the test loss remains high. Figure (c) visualizes the dynamics of phase alignment level, measured by $\frac{1}{m}\sum_{m=1}^{M}|\sin(\mathcal{D}_m^\star)|$. Figure (d) tracks the evolution of the average neuron-wise frequency sparsity level, as measured by the inverse participation ratio (IPR) of the Fourier coefficients, alongside the $\ell_2$-norm of the parameter.

on the training data and their symmetric counterparts in the test set (due to the exchangability of the two input numbers), but completely fails to generalize to truly "unseen" test points (see Figure 9). At this phase, all the frequency components in one neuron keep growing but at different pace similar to the *lottery ticket mechanism* described previously. Next, the model enters the first generalization stage, which is characterized by a precise interplay between the two forces. We conclude that both forces are active because the parameter norms continue to grow, which is a clear indicator of ongoing loss minimization. At the same time, weight decay induces a *sparsification effect* in the frequency domain. Specifically, the one frequency component that dominates in the lottery ticket mechanism continues growing, while weight decay refines the learned sparse features by pruning the remaining components, making it closer to the clean single-frequency solution for each neuron and causing the test loss to drop sharply. This dynamic culminates in a turning point around step 10,000, which marks the onset of the second and final generalization stage. From this point, weight decay becomes the dominant force, slowly pushing the test accuracy toward a perfect score.

**Common-to-Rare Memorization.** Early in training, as training accuracy rises, test accuracy falls from an initial 5% (due to small random initialization) to 0% (see Figure 3b). By Step 1000, when training accuracy peaks, the first phase is evident: the model prioritizes memorizing common data, specifically symmetric pairs where both $(i, j)$ and its counterpart $(j, i)$ are in the training set. This intense focus comes at a cost, as the model actively *suppresses performance on rare examples* within the same training set, driving their accuracy to zero. Only after mastering the common data does the model shift its focus to the second phase: memorizing these rare examples that appear only once. Please refer to §E.1 for a more detailed interpretation of grokking dynamics.

## 4 MECHANISTIC INTERPRETATION OF LEARNED MODEL

In this section, we first tackle the interpretability question in a slightly idealized setting, leveraging the trigonometric patterns in Observations 1-3 and, motivated by Observation 4, adopting a quadratic activation for analytical convenience. We show that the trained model effectively approximates an *indicator function* via a *majority-voting* scheme within the Fourier space.

**Single-Neuron Contribution and Majority Voting.** Under the parametrization of (3.1) in Observation 1 and the phase-alignment condition $2\phi_m - \psi_m = 0 \mod 2\pi$ for all $m$ in Observation 2, the contribution of each neuron to the logit at dimension $j \in [p]$ can be expressed as:

$$f^{[m]}(x, y; \xi, \theta)[j] \propto \cos(\omega_{\varphi(m)}(x - y)/2)^2 \cdot \{\underbrace{\cos(\omega_{\varphi(m)}(x + y - j))}_{\text{primary signal}}$$

$$+ 2\cos(\omega_{\varphi(m)}j + 2\phi_m) + \cos(\omega_{\varphi(m)}(x + y + j) + 4\phi_m)\}. \quad (4.1)$$

Here, $\cos(\omega_k(x + y - j))$ provides the primary signal—its value peaks exactly at $j = (x + y) \mod p$—while the remaining terms act as residual noise whose amplitude and sign depend on the chosen frequency $k$, phase $\phi_m$, and input pair $(x, y)$. Similar results have also been reported in Gromov (2023); Zhong et al. (2023); Nanda et al. (2023); Doshi et al. (2023).

Although each neuron's contribution is biased by its own frequency-phase "view", the network as a whole can attain perfect accuracy via a *majority-voting* mechanism: every neuron votes based on its

individual view, the model then aggregates these biased yet diverse votes to distill the correct answer. Despite this intuitive diversification argument, two questions remain unanswered: (a) How should we define "diversification"? (b) To what extent can the residual noise be canceled by aggregating over a diverse set of frequency-phase pairs $(\varphi(m), \phi_m)$?

**Majority-Voting Approximates Indicator via Overparameterization.** Motivated by Observation 3, when $M$ is sufficiently large, the model naturally learns completely diversified neurons: every frequency $k$ is represented, and the phases exhibit uniform symmetry. We formalize this below.

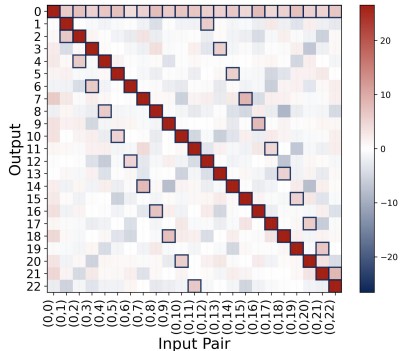

**Definition 4.1.** *The neurons is called fully diversified if the frequency-phase pairs $\{(\varphi(m), \phi_m)\}_{m \in [M]}$ satisfy the following properties: (i) for every frequency $k \in [\frac{p-1}{2}]$, there are exactly $N$ neurons $m$ with $\varphi(m) = k$, (ii) there exists a constant $a > 0$ such that $\alpha_m \beta_m^2 = a$ for all $m \in [M]$, and (iii) for each $k$ and $\iota \in \{2, 4\}$, $\exp\left(i \cdot \iota \sum_{m \in \mathcal{N}_k} \phi_m\right) = 0$.*

Definition 4.1 is primarily a formal restatement of Observation 3. In particular, condition (ii) follows from the homogeneous scaling of magnitudes, and condition (iii) captures the high-order phase symmetry implied by the uniformity

Figure 4: Heatmap of the output logits with quadratic activation.

within the frequency group. Condition (i) assumes an exact frequency balance—an idealization that holds approximately under random initialization. We are now ready to present the interpretation of learned model.

**Proposition 4.2.** *Suppose that the neurons are completely diversified as per Definition 4.1. Under the parametrization in (3.1) and the phase-alignment condition $2\phi_m - \psi_m = 0 \mod 2\pi$ for all $m \in [M]$, the output logit at dimension $j \in [p]$ takes the form:*
$$f(x, y; \xi, \theta)[j] = \frac{aN}{2}\Big\{-1 + \underbrace{\frac{p}{2}\mathbb{1}(x + y \mod p = j)}_{\text{signal term}} + \underbrace{\frac{p}{4}\big(\mathbb{1}(2x \mod p = j) + \mathbb{1}(2y \mod p = j)\big)}_{\text{noise terms}}\Big\}.$$

*For any $\epsilon \in (0, 1)$, by taking $a \gtrsim (Np)^{-1} \cdot \log(p/\epsilon)$, we have $\|\texttt{smax} \circ f(\cdot, \cdot; \xi, \theta) - e_{m_p(\cdot, \cdot)}\|_{1, \infty} \le \epsilon$.*

Please refer to §F.1 for a detailed proof of Proposition 4.2. The proposition states that although each neuron individually implements a trigonometric mechanism as shown in (4.1), the diversified neurons indeed collectively approximate the indicator function $\mathbb{1}(x + y \mod p = j)$. As noted in Zhong et al. (2023), the $\cos(\omega_{\varphi(m)}(x - y)/2)^2$ term in (4.1) is the Achilles' heel of this strategy. We show that even under complete diversification, it would still introduce spurious peaks at $2x \mod p$ and $2y \mod p$. However, from the above equation, we see that the true signal peak exceeds these noise peaks by $aNp/8$. Hence, after the softmax operation, the model's output would concentrate on the correct sum $x + y \mod p$ as long as the magnitude grows large enough during the training.

## 5 TRAINING DYNAMICS FOR FEATURE EMERGENCE

In this section, we provide a theoretical understanding of how features emerge during standard gradient-based training. Unlike previous theoretical works that focused on loss landscape analysis (e.g., Morwani et al., 2023), we offer a more complete view from the perspective of training dynamics. To achieve this, we track the evolution of the model's parameters directly in Fourier space.

### 5.1 A DYNAMICAL PERSPECTIVE ON FEATURE EMERGENCE

In the following, we provide a theoretical understanding of how the features—*single-frequency* and *phase alignment* patterns, i.e, Observation 1 and 2, emerge during training. For theoretical convenience, we adopt the quadratic activation (Arous et al., 2025) and focus on the training over a complete dataset $\mathcal{D}_{\text{full}}$, a familiar setting in prior work (e.g., Morwani et al., 2023; Tian, 2024).

**Gradient Flow.** Consider training a two-layer neural network as defined in (2.1) with one-hot input embeddings, i.e., $h_x = e_x \in \mathbb{R}^p$, parameterized by $\Theta = \{\xi, \theta\}$, and the loss $\ell$ is given by the cross-entropy (CE) loss in (2.2), evaluated over the full dataset $\mathcal{D}_{\text{full}}$. When training the parameter $\Theta$ using the gradient flow, the dynamics are governed by $\partial_t \Theta_t = \nabla \ell(\Theta_t)$. We consider gradient flow under an initialization that satisfies the following conditions.

**Assumption 5.1** (Initialization). *For each neuron $m \in [M]$, the network parameters $(\xi_m, \theta_m)$ are initialized as $\theta_m \sim \kappa_{\text{init}} \cdot \sqrt{p/2} \cdot (\varrho_1[1] \cdot b_{2k} + \varrho_1[2] \cdot b_{2k+1})$ and $\xi_m \sim \kappa_{\text{init}} \cdot \sqrt{p/2} \cdot (\varrho_2[1] \cdot b_{2k} + \varrho_2[2] \cdot b_{2k+1})$ where $\varrho_1, \varrho_2 \overset{i.i.d.}{\sim} \text{Unif}(\mathbb{S}^1)$, $k \sim \text{Unif}([\frac{p-1}{2}])$ and $\kappa_{\text{init}}$ is sufficiently small.*

Assumption 5.1 posits that each neuron is initialized randomly but contains a single-frequency component, all at the same small scale. This specialized initialization is adopted for theoretical convenience, allowing us to sidestep the chaotic frequency competition and study the evolution of one specific frequency. Specifically, the single-frequency pattern is **sufficient to capture the overall behavior** as each frequency component evolves within its own **orthogonal** subspace. In §D.1, we will extend to the case where each neuron is initialized with multiple frequencies.

**Section Roadmap.** With a slight abuse of notation, we let $k^\star$ denote the initial frequency of each neuron (see Assumption 5.1) and use the superscript $\star$ instead of $k^\star$ to simplify the notation further. In the following, we aim to show that (i) the single-frequency pattern, i.e., $g_m[j] = r_m[j] = 0$ for all $j \neq 2k^\star, 2k^\star + 1$, is preserved throughout the gradient flow (see §5.1.1), and (ii) the phases of the first and second layers will align such that $2\phi_m^\star(t) - \psi_m^\star(t) \bmod 2\pi$ converges to 0 (see §5.1.2).

### 5.1.1 PRESERVATION OF SINGLE-FREQUENCY PATTERN

Recall that the dynamics of the parameters are approximately given by ODEs in (A.2a) and (A.2b). Note the constant frequency, i.e., $g_m[1]$ and $r_m[1]$, remains almost 0 due to the centralized essence:

$$\partial_t \theta_m[j](t), \ \partial_t \xi_m[j](t) \in \text{span}(\{b_\tau\}_{\tau=2}^p), \qquad \forall j \in [p]. \tag{5.1}$$

By definition, we can show that $\partial_t g_m[1](t) = \langle b_1, \partial_t \theta_m(t) \rangle$ and $\partial_t r_m[1](t) = \langle b_1, \partial_t \xi_m(t) \rangle$. Given the zero-initialization $g_m[1] = r_m[1] = 0$ (see Assumption 5.1), and utilizing (5.1), it follows that

$$\partial_t g_m[1](t) \approx \partial_t r_m[1](t) \approx 0 \quad \text{s.t.} \quad g_m[1](t) \approx r_m[1](t) \approx 0, \tag{5.2}$$

throughout the first stage. Moreover, to establish frequency preservation, we track the magnitudes of each frequency, i.e., $\{\alpha_m^k\}_{k \in [(p-1)/2]}$ and $\{\beta_m^k\}_{k \in [(p-1)/2]}$. Thanks to the orthogonality of the Fourier basis, by applying the chain rule, for each frequency $k$, it holds that

$$\partial_t \alpha_m^k(t) \approx 2p \cdot \alpha_m^k(t)\beta_m^k(t) \cdot \cos\left(2\phi_m^k(t) - \psi_m^k(t)\right), \ \partial_t \beta_m^k(t) \approx p \cdot \alpha_m^k(t)^2 \cdot \cos\left(2\phi_m^k(t) - \psi_m^k(t)\right),$$

where the evolution of the magnitudes for frequency $k$ only depends on $(\alpha_m^k, \beta_m^k, \phi_m^k, \psi_m^k)$. Given the initial value $\alpha_m^k(0) = \beta_m^k(0) = 0$ for $k \neq k^\star$ (see Assumption 5.1), we have

$$\alpha_m^k(t) \approx \beta_m^k(t) \approx 0, \qquad \forall k \neq k^\star. \tag{5.3}$$

Recall that we define $\alpha_m^k = \sqrt{2/p} \cdot \|g_m^k\|$ and $\beta_m^k = \sqrt{2/p} \cdot \|r_m^k\|$. By combining (5.2) and (5.3), we can establish the preservation of *single-frequency* pattern (see Figure 13 for experimental results):

$$g_m[j](t) \approx r_m[j](t) \approx 0, \qquad \forall j \neq 2k^\star, 2k^\star + 1. \tag{5.4}$$

Based on (5.4), by simple calculations, we have

$$\partial_t \theta_m[j](t) \approx 2p \cdot \alpha_m^\star(t) \cdot \beta_m^\star(t) \cdot \cos(\omega_\star j + \psi_m^\star(t) - \phi_m^\star(t)),$$
$$\partial_t \xi_m[j](t) \approx p \cdot \alpha_m^\star(t)^2 \cdot \cos(\omega_\star j + 2\phi_m^\star(t)). \tag{5.5}$$

For each neuron, its evolution can be approximately characterized by a four-particle dynamical system consisting of magnitudes $\alpha_m^\star(t)$ and $\beta_m^\star(t)$ and phases $\phi_m^\star(t)$ and $\psi_m^\star(t)$. We formalize the result in (5.4) and the approximate arguments above into the following theorem.

**Theorem 5.2** (Informal). *Under the initialization in Assumption 5.1, for a given threshold $C_{\text{end}} > 0$, we define the initial stage as $(0, t_{\text{init}}]$, where $t_{\text{init}} := \inf\{t : \max_{m \in [M]} \|\theta_m(t)\|_\infty \vee \|\xi_m(t)\|_\infty \leq C_{\text{end}}\}$. Suppose that $\log M/M \lesssim c^{-1/2} \cdot (1 + o(1))$, $\kappa_{\text{init}} = o(M^{-1/3})$ and $C_{\text{end}} \asymp \kappa_{\text{init}}$, given sufficiently small $\kappa_{\text{init}}$, we have $\max_{k \neq k^\star} \inf_{t \in (0, t_{\text{init}})} \alpha_m^k(t) \vee \beta_m^k(t) = o(\kappa_{\text{init}})$.*

The formal statement and proof of Theorem 5.2 is provided in §F.4. The theorem states that under a small random initialization, during the initial training stage, the non-feature frequencies, which are initialized at zero, will not grow beyond $o(\kappa_{\text{init}})$.

### 5.1.2 NEURON-WISE PHASE ALIGNMENT

We proceed to investigate the emergence of the phase alignment phenomenon. To build intuition, we first consider a special stationary point $\psi_m^\star = 2\phi_m^\star$. According to the dynamics given by (5.5), it is straightforward to observe the stationarity, as: $\partial_t \theta_m[j](t) \propto \cos(\omega_\star j + \phi_m^\star(t))$ and $\partial_t \xi_m[j](t) \propto \cos(\omega_\star j + 2\phi_m^\star(t))$. This implies that at the double-phase stationary point, $\theta_m[j](t)$ and $\xi_m[j](t)$ evolve in the same direction as themselves and cease to rotate. By applying the chain rule over (5.5),

$$\partial_t \exp(i\phi_m^\star(t)) \approx 2p \cdot \beta_m^\star(t) \cdot \sin\left(2\phi_m^\star(t) - \psi_m^\star(t)\right) \cdot \exp\left(i\{\phi_m^\star(t) - \pi/2\}\right),$$
$$\partial_t \exp(i\psi_m^\star(t)) \approx p \cdot \alpha_m^\star(t)^2/\beta_m^\star(t) \cdot \sin\left(2\phi_m^\star(t) - \psi_m^\star(t)\right) \cdot \exp\left(i\{\psi_m^\star(t) + \pi/2\}\right). \tag{5.6}$$

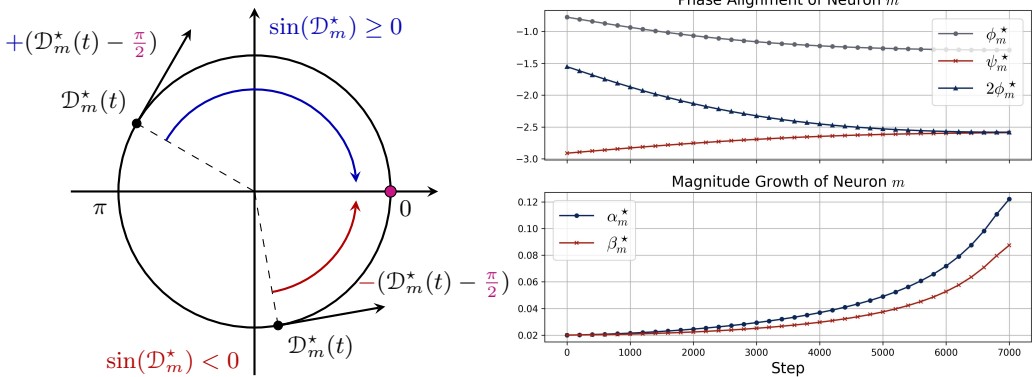

(a) Illustration of Phase Alignment Behavior.  (b) Dynamics of Magnitudes and Phases for Neuron $m$.

Figure 5: Visualizations of the alignment behavior and neuron dynamics. Figure (a) illustrates the dynamics of the normalized phase difference $\mathcal{D}_m^\star(t)$ given by (5.7). Initialized randomly on the unit circle, the gradient flow will always drive $\mathcal{D}_m^\star(t)$ to 0, regardless of the initial half-space. Figure (b) plots the dynamics of magnitudes and phases of the feature frequency for a specific neuron $m$.

Thus, phases $\phi_m^\star$ and $\psi_m^\star$ evolve in the opposite directions, with rotation speed primarily determined by the magnitudes and misalignment level, quantified by $|\sin(2\phi_m^\star(t) - \psi_m^\star(t))|$. This suggests that $2\phi_m^\star$ will eventually "meet" $\psi_m^\star$. To understand the dynamics of the alignment behavior, we track $\mathcal{D}_m^\star(t) = 2\phi_m^\star(t) - \psi_m^\star(t) \mod 2\pi \in [0, 2\pi)$. Using (5.6), the chain rule gives that

$$\partial_t \exp(i\mathcal{D}_m^\star(t)) \approx \left(4\beta_m^\star(t) - \alpha_m^\star(t)^2/\beta_m^\star(t)\right) \cdot p \cdot \sin\left(\mathcal{D}_m^\star(t)\right) \cdot \exp\left(i\{\mathcal{D}_m^\star(t) - \pi/2\}\right). \quad (5.7)$$

Notably, though $\{0, \pi, 2\pi\}$ are all stationary points of (5.7), the evolution of $\mathcal{D}_m^\star(t)$ is consistently directed toward 0. This is due to the sign of $\sin(\mathcal{D}_m^\star(t))$, which adaptively ensures $\partial_t \exp(i\mathcal{D}_m^\star(t))$ converges only to zero (see Figure 5a). Thus, we can establish the *phase alignment* behavior below:

$$2\phi_m^\star(t) - \psi_m^\star(t) \mod 2\pi \to 0 \quad \text{when} \quad t \to \infty.$$

**Magnitude Remains Small after Alignment.** Note the above analysis hinges on the parameter scale being sufficiently small. To complete the argument, it remains to show that $\alpha_m^\star(t)$ and $\beta_m^\star(t)$ remain small even after the phase is well-aligned.

Under the initialization specified in Assumption 5.1, we can establish the following relationship:

$$\sin(\mathcal{D}_m^\star(t)) = \sin(\mathcal{D}_m^\star(0)) \cdot \{\mathcal{R}_m^\star(t) \cdot (2\mathcal{R}_m^\star(t)^2 - 1)\}^{-1}, \quad \text{where} \quad \mathcal{R}_m^\star(t) := \beta_m^\star(t)/\kappa_{\text{init}}.$$

This implies that when misalignment level $\sin(\mathcal{D}_m^\star(t))$ reaches a small threshold $\delta > 0$, the ratio $\mathcal{R}_m^\star(t)$ is bounded by $\{\sin(\mathcal{D}_m^\star(0))/\delta\}^{1/3}$. Thus, since $\alpha_m^\star(t) \asymp \beta_m^\star(t)$, when the neuron is well-aligned, the parameter scales remain on the same order as at initialization. This aligns with experimental results in Figure 5b. We summarize these findings in the theorem below.

**Theorem 5.3** (Informal). *Consider the main flow dynamics under the initialization in Assumption 5.1. For any initial misalignment $\mathcal{D}_m^\star(0) \in [0, 2\pi)$ and small tolerance level $\delta \in (0, 1)$, the minimal time $t_\delta$ required for the phase to align such that $|\mathcal{D}_m^\star(t)| \leq \delta$ satisfies that*

$$t_\delta \asymp (p\kappa_{\text{init}})^{-1} \cdot \left(1 - \{\sin(\mathcal{D}_m^\star(0))/\delta\}^{-1/3} + \max\{\pi/2 - |\mathcal{D}_m^\star(0) - \pi|, 0\}\right),$$

*and the magnitude at this time is given by $\beta_m^\star(t_\delta) \asymp \kappa_{\text{init}} \cdot \{\sin(\mathcal{D}_m^\star(0))/\delta\}^{1/3}$. Moreover, in the mean-field regime $m \to \infty$, let $\rho_t = \text{Law}(\phi_m^\star(t), \psi_m^\star(t))$ for all $t \in \mathbb{R}^+$ and let $\lambda$ denote the uniform law on $(0, 2\pi]$. Then, $\rho_0 = \lambda \otimes \lambda$ and $\rho_\infty = T_\# \lambda$, where $T : \varphi \mapsto (\varphi, 2\varphi) \mod 2\pi$.*

Theorem 5.3 provides two key insights. First, it establishes that the convergence time depends on (i) the initial misalignment level, (ii) the extent to which $\mathcal{D}_m^\star(0)$ deviates from the intermediate stage for $\mathcal{D}_m^\star(0) \in \left(\frac{\pi}{2}, \frac{3\pi}{2}\right)$, and (iii) the initialization scale $\kappa_{\text{init}}$ and modulus $p$. Second, the theorem provides a theoretical justification for phase symmetry (Observation 3) in the mean-field regime. For the formal theorem, a proof sketch, and the complete proof, see Theorem F.7, §F.3.1, and §F.5.

**Theoretical Extensions.** In §D, we extend the results from §5 to two more general scenarios: lottery mechanism under multi-frequency initialization in §D.1 and the dynamics with ReLU activation in §D.2 based on the preliminary result above.

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

CONTENTS

# A  BACKGROUNDS AND SUPPLEMENTARY RESULTS

## A.1  ADDITIONAL FIGURES AND TABLES

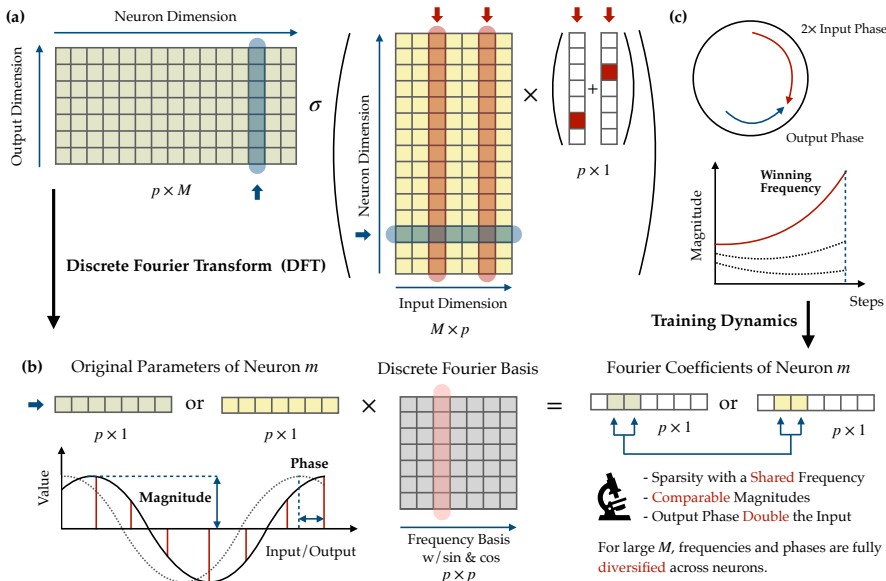

Figure 6: An illustration of the primary analytical technique and results. Discrete Fourier Transform (DFT) is utilized to quantitatively interpret the mechanism of learned models within the feature space, revealing the training dynamics that provably result in consistent feature learning.

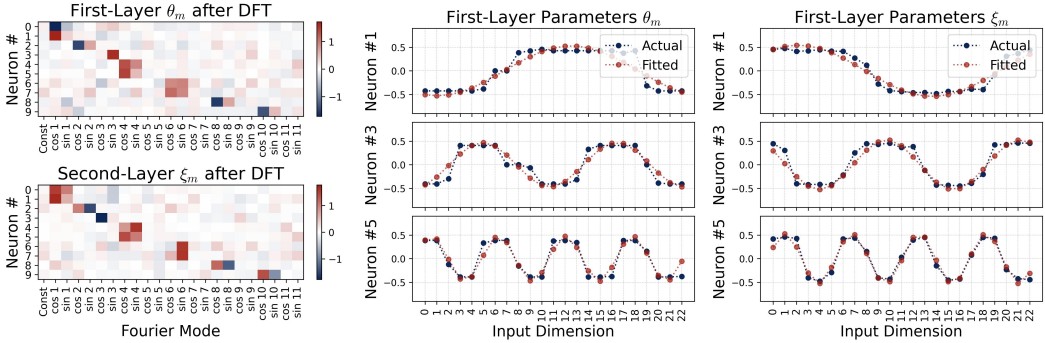

(a) Heatmap of Learned Parameters.      (b) Actual Learned and Fitted Parameters of Each Neuron.

Figure 7: Learned parameters under the full random initialization with $p = 23$ and ReLU activation using AdamW. Figure (a) plots a heatmap of the learned parameters for the top 10 neurons after Discrete Fourier Transform (DFT, see §A.3). Each row in the heatmap corresponds to the Fourier components of a single neuron's parameters. The plot clearly reveals a single-frequency pattern: each neuron exhibits a large, non-zero value focused on only one specific frequency component, confirming a highly sparse and specialized frequency encoding. Figure (b) further examines the periodicity by plotting line plots of the learned parameters for three neurons, each overlaid with a trigonometric curve fitted via DFT. The fitted curve aligns almost perfectly with the actual one.

| $\sigma(x)$ | $\max\{x, 0\}$ | $|x|$ | $x^2$ | $x^4$ | $x^8$ | $\log(1 + e^{2x})$ | $e^x$ | $x$ | $x^3$ | $x^5$ | $x^7$ |
|---|---|---|---|---|---|---|---|---|---|---|---|
| Loss | $1.194 \times 10^{-8}$ | 0.000 | 0.000 | $3.1 \times 10^{-5}$ | 0.051 | $1.2 \times 10^{-3}$ | $6.5 \times 10^{-4}$ | 4.246 | 3.891 | 3.611 | 3.413 |
| Acc. | 1.000 | 1.000 | 1.000 | 1.000 | 1.000 | 1.000 | 1.000 | 0.041 | 0.036 | 0.032 | 0.026 |

Table 1: loss and accuracy of a model trained with ReLU activation, then tested the same architecture with different activation functions replacing ReLU. As shown in the table, activations such as absolute function, even-order polynomials, and exponential function achieve perfect accuracy.

## A.2 Notation and Further Related Works

**Notation.** For $n \in \mathbb{N}^+$, let $[n] = \{i \in \mathbb{Z} : 1 \leq i \leq n\}$. Let $\mathbb{Z}_p$ denote the set of integers modulo $p$. The $\ell_p$-norm is denoted by $\| \cdot \|_p$. For a vector $\nu \in \mathbb{R}^d$, its $i$-th entry is denoted by $\upsilon[i]$ The softmax operator, $\mathtt{smax}(\cdot)$, maps a vector to a probability distribution, where the $i$-th component is given by $\mathtt{smax}(\upsilon)_i = \exp(\upsilon_i)/\sum_j \exp(\upsilon_j)$. For two non-negative functions $f(x)$ and $g(x)$ defined on $x \in \mathbb{R}^+$, we write $f(x) \lesssim g(x)$ or $f(x)$ as $O(g(x))$ if there exists two constants $c > 0$ such that $f(x) \leq c \cdot g(x)$, and write $f(x) \gtrsim g(x)$ or $f(x)$ if there exists two constants $c > 0$ such that $f(x) \geq c \cdot g(x)$. We write $f(x) \asymp g(x)$ or $f(x) = \Theta(g(x))$ if $f(x) \lesssim g(x)$ and $g(x) \lesssim f(x)$.
In the following, we discuss the additional related works in detail, which complements the discussion in §1.1.

**Training Dynamics of Neural Networks.** To understand how neural networks perform feature learning, a significant body of work has analyzed the training dynamics of neural networks under gradient-based optimization. This research typically focuses on settings where the target function exhibits a low-dimensional structure, such as single-index (Ba et al., 2022; Lee et al., 2024; Berthier et al., 2024; Chen et al., 2025) and multi-index models (Damian et al., 2022; Arnaboldi et al., 2024; Ren et al., 2025). Furthermore, Allen-Zhu & Li (2019); Shi et al. (2022; 2023) have considered more general cases, analyzing function classes that encode latent features.

**Theoretical Interpretation of Modular Addition.** Theoretical understanding of this modular addition task, however, remains incomplete. Morwani et al. (2023) characterized the loss landscape under the max-margin framework using a non-standard $\ell_{2,3}$-regularization. Tian (2024) further analyzed the landscape of a modified $\ell_2$-loss within the Fourier space, generalized these results to data with semi-ring structures on Abelian groups, and provided a heuristic derivation for the mean-field dynamics of frequencies. Recently, Wang & Wang (2025) formalized and extended these mean-field results by analyzing the Wasserstein gradient flow under a geometric equivariance constraint. While Tian (2024) and Wang & Wang (2025) provide a characterization of a simpler, mean-field dynamics, a full analytical result explaining the alignment and competition dynamics at the finite, neuron-wise level remains an open problem. A different approach studies grokking modular arithmetic via the average gradient outer product for backpropagation-free models (Mallinar et al., 2024). Another line of research focuses on grokking dynamics and frames it as a two-phase process, transitioning from an initial lazy (kernel) regime to a later rich (feature) regime (Kumar et al., 2024; Lyu et al., 2023; Mohamadi et al., 2024; **?**), which are broadly related to our work.
Furthermore, to compare with the existing results in depth, we further compare with the closely related works at a technical level in §B.

## A.3 Technical Background: Discrete Fourier Transform and Notations

Motivated by empirical observations in §3, it is natural to apply the Fourier transform to model parameters and to track the evolution of the Fourier coefficients throughout the training process. This allows us to investigate how these Fourier features are learned. We begin by defining the Fourier basis matrix over $\mathbb{Z}_p$ by $B_p = [b_1, \ldots, b_p] \in \mathbb{R}^{p \times p}$, where each column is given by

$$b_1 = \frac{\mathbf{1}_p}{\sqrt{p}}, \qquad b_{2k} = \sqrt{\frac{2}{p}} \cdot [\cos(\omega_k), \ldots, \cos(\omega_k p)], \qquad b_{2k+1} = \sqrt{\frac{2}{p}} \cdot [\sin(\omega_k), \ldots, \sin(\omega_k p)],$$

where $w_k = 2k\pi/p$ for all $k \in [\frac{p-1}{2}]$[1]. We then project the model parameters, $\xi_m$'s and $\theta_m$'s, onto this basis. This change of basis is equivalent to applying the Discrete Fourier Transform (DFT, Sundararajan, 2001), yielding the Fourier coefficients:

$$g_m = B_p^\top \theta_m, \qquad r_m = B_p^\top \xi_m, \qquad \forall m \in [M].$$

To better interpret these coefficients, we group the sine and cosine components for each frequency $k$ and reparameterize them by their magnitude and phase. Let $g_m^k = (g_m[2k], g_m[2k+1])$ and $r_m^k = (r_m[2k], r_m[2k+1])$ denote the coefficient vector in correspondence to frequency $k$. Their magnitudes $(\alpha_m^k, \beta_m^k)$ and phases $(\phi_m^k, \psi_m^k)$ are defined as

$$\alpha_m^k = \sqrt{\frac{2}{p}} \cdot \|g_m^k\|, \qquad \phi_m^k = \mathrm{atan}(g_m^k), \qquad \beta_m^k = \sqrt{\frac{2}{p}} \cdot \|r_m^k\|, \qquad \psi_m^k = \mathrm{atan}(r_m^k).$$

Here, $\mathrm{atan}(x) = \mathrm{atan2}(-x[2], x[1])$ where $\mathrm{atan2} : \mathbb{R} \times \mathbb{R} \mapsto (-\pi, \pi]$ is the 2-argument arctangent. This polar representation is intuitive, as it directly relates the coefficients to a phase-shifted cosine, e.g., $g_m[2k] \cdot b_{2k}[j] + g_m[2k+1] \cdot b_{2k+1}[j] = \alpha_m^k \cdot \cos(w_k j + \phi_m^k)$. By setting constant coefficients as $\alpha_m^0 = g_m[1]/\sqrt{p}$ and $\beta_m^0 = r_m[1]/\sqrt{p}$, we can recover the expanded form in (3.2).

---

[1]We choose $p$ as a prime number greater than 2 to simplify the analysis.

## A.4 PROPERTIES AT THE INITIAL STAGE

Given a sufficiently small initialization in Assumption 5.1, a key property at the initial stage is that the parameter magnitudes remain small, resulting in the softmax output being nearly uniform over. Formally, $\|\theta_m\|_\infty$ and $\|\xi_m\|_\infty$ are small such that the following equality holds approximately:

$$\texttt{smax} \circ f(x, y; \xi, \theta) \approx \frac{1}{p} \cdot \mathbf{1}_p. \tag{A.1}$$

While (A.1) suggests that the neural network behaves as a poorly performing uniform predictor at the initial stage due to the small parameter magnitudes, this does not imply that the model learns nothing. Instead, the model can learn the "feature direction" of the data under the guidance of the gradient. In what follows, we examine the key components of the gradient and define the time threshold $t_{\text{init}}$ to ensure all parameters remain within a small scale.

**Neuron Decoupling.** We first show that the neurons are *decoupled* at the initial stage, meaning the evolution of parameters $\theta_m$ and $\xi_m$ depends solely on $(\theta_m, \xi_m)$—the parameters of neuron $m$ itself—by using the approximation in (A.1). To establish this, we compute the gradient and simplify it using periodicity. We derive that the gradient flow for each neuron $m \in [M]$ at the initial stage admits the following simplified form: for each entry $j \in [p]$, we have

$$\partial_t \theta_m[j](t) \approx 2p \cdot \sum_{k=1}^{(p-1)/2} \alpha_m^k(t) \cdot \beta_m^k(t) \cdot \cos(\omega_k j + \psi_m^k(t) - \phi_m^k(t))$$

$$+ 2p \cdot \beta_m^0(t) \cdot \sum_{k=1}^{(p-1)/2} \alpha_m^k \cdot \cos(\omega_k j + \phi_m^k(t)), \tag{A.2a}$$

$$\partial_t \xi_m[j](t) \approx p \cdot \sum_{k=1}^{(p-1)/2} \alpha_m^k(t)^2 \cdot \cos(\omega_k j + 2\phi_m^k(t)). \tag{A.2b}$$

Here, we use the Fourier expansion of parameters $\theta_m(t)$ and $\xi_m(t)$ as given in (3.2). Following this, we can see that the dynamics, i.e., $\partial_t \theta_m(t)$ and $\partial_t \xi_m(t)$, only depends on $\{(\alpha_m^k, \beta_m^k, \phi_m^k, \psi_m^k)\}_{k \in [(p-1)/2]}$ and $r_m[1]$ that corresponds to neuron $m$. This demonstrates a decoupled evolution among neurons. Hence, in the remaining section, we can focus on a fixed neuron $m$. Similar decoupling technique with a similar small output scale is also seen in Lee et al. (2024); Chen et al. (2025) for $\ell_2$-loss.

**Remark A.1** (Equivalence to Margin Maximization under Small Initialization). *Notice that the module task is a multi-class classification problem. To understand the feature emergence, Morwani et al. (2023) considers an average margin maximization problem, where the margin is defined by*

$$\max_{\xi, \theta} \ell_{\textsf{AM}}(\xi, \theta) \ \text{ with } \ \ell_{\textsf{AM}}(\xi, \theta) = \sum_{x \in \mathbb{Z}_p} \sum_{y \in \mathbb{Z}_p} \left\{ f(x, y; \xi, \theta)[(x + y) \bmod p] - \frac{1}{p} \sum_{j \in \mathbb{Z}_p} f(x, y; \xi, \theta)[j] \right\}.$$

*In comparison, given the small scale of parameters during the initial stage, we can show that, similar to the approximation in (A.1), the loss takes the approximate form:*

$$\ell(\xi, \theta) = -\sum_{x \in \mathbb{Z}_p} \sum_{y \in \mathbb{Z}_p} f(x, y; \xi, \theta)[(x + y) \bmod p] + \sum_{x \in \mathbb{Z}_p} \sum_{y \in \mathbb{Z}_p} \log \left( \sum_{j=1}^{p} \exp(f(x, y; \xi, \theta)[j]) \right)$$

$$\approx \underbrace{-\sum_{x \in \mathbb{Z}_p} \sum_{y \in \mathbb{Z}_p} f(x, y; \xi, \theta)[(x + y) \bmod p] + \frac{1}{p} \sum_{x \in \mathbb{Z}_p} \sum_{y \in \mathbb{Z}_p} \sum_{j=1}^{p} f(x, y; \xi, \theta)[j]}_{= -\ell_{\textsf{AM}}(\xi, \theta)} + p^2 \log p,$$

*where we use the first-order approximations $\exp(x) \approx 1 + x$ and $\log(1 + x) \approx x$ for small $x$. Following this, we observe that during the initial stage, minimizing the loss in (2.2) is equivalent to optimizing the average margin. This connection underpins the theoretical insights in Morwani et al. (2023), which links the margin maximization problem to empirical observations.*

## B COMPARISON WITH EXISTING RESULTS

Our work is closely related to that of Tian (2024) and Wang & Wang (2025), who studied a two-layer network for learning group multiplication on an Abelian group, which is a generalization of the standard modular addition task. For theoretical convenience, they adopt a modified $\ell_2$-loss to mitigate noisy interactions induced by the constant frequency. Let $\mathscr{P}_1^\perp = I - \frac{1}{p}\mathbf{1}\mathbf{1}^\top$ denote the mean-zero projection, then the loss is defined as

$$\widetilde{\ell}(\xi, \theta) = -\sum_{x \in \mathbb{Z}_p} \sum_{y \in \mathbb{Z}_p} \left\| \mathscr{P}_1^\perp \left( 1/2p \cdot f(x, y; \xi, \theta) - e_{(x+y) \bmod p} \right) \right\|^2, \tag{B.1}$$

where the output of the network is normalized by $1/2p$ within loss calculation. Unlike (B.1), we show that minimizing a standard CE loss with a small initialization naturally decouples the dynamics of each frequency (see Theorem 5.2), with the constant frequency having a zero gradient throughout training and therefore remaining zero under zero-constant initialization (see Corollary D.1).

**Notation Clarifications.** We begin by explaining the notation used in Tian (2024). In their analysis, the (modified) complex Fourier coefficients of the weights are given by $z_{qkm} \in \mathbb{C}$, where the indices $q \in \{\xi, \theta\}$, $m \in [M]$ and $k \in [p-1] \cup \{0\}$ correspond to the layer, neuron, and frequency, respectively. This complex representation is equivalent to the real-valued cosine-sine pairs used in our DFT definition in §A.3. Specifically, for all $k \leq (p-1)/2$, we can show that

$$z_{\theta km} = \alpha_m^k / \sqrt{2} \cdot \exp(i\phi_m^k), \qquad z_{\xi km} = \beta_m^k / \sqrt{2} \cdot \exp(-i\psi_m^k).$$

By the conjugate symmetry of the DFT coefficients, our single real component at frequency $k$ determines the complex coefficients for both $k$ and $p - k$. Therefore, for the higher frequencies $(p+1)/2 \leq k \leq p$, the relationship is given by

$$z_{\theta km} = \bar{z}_{\theta(p-k)m} = \alpha_m^k / \sqrt{2} \cdot \exp(-i\phi_m^k), \qquad z_{\xi km} = \bar{z}_{\xi(p-k)m} = \beta_m^k / \sqrt{2} \cdot \exp(i\psi_m^k),$$

which completes the one-to-one correspondence between our basis and the one used by Tian (2024).

**Loss Landscape within Fourier Domain.** Tian (2024) expresses the loss $\widetilde{\ell}$ from (B.1) in the Fourier domain using $\{z_{qkm}\}$. In Theorem 1, they show that the loss $\widetilde{\ell}$ decouples into per-frequency terms $\widetilde{\ell} = p^{-1} \cdot \sum_{k \neq 0} \widetilde{\ell}_k + (p-1)/p$, where $\widetilde{\ell}_k$ is a quadratic polynomial whose variables $\{\rho_{k_1 k_2 k}\}_{k_1, k_2 \in [p-1]}$ are third-order monomials of the Fourier coefficients. Formally, we have

$$\widetilde{\ell}_k = \text{poly}\left(\{\rho_{k_1 k_2 k}\}_{k_1, k_2 \in [p-1]}\right), \quad \text{where} \quad \rho_{k_1 k_2 k} = \sum_{m=1}^M z_{\theta k_1 m} z_{\theta k_2 m} z_{\xi k m}. \tag{B.2}$$

**Mean-Field Gradient Dynamics.** Building on their analysis of the loss, Theorem 7 in Tian (2024) presents a heuristic result for the gradient dynamics. By considering a simplified setting with a truncated loss polynomial from (B.2), a symmetric Gaussian initialization, and the mean-field limit $M \to \infty$, they show that

$$\partial_t \rho_{k_1 k_2 k}(t) = 2 \cdot \zeta_{k_1 k_2 k}(t) \cdot \{\mathbb{1}(k_1 = k_2 = k) - \rho_{k_1 k_2 k}(t)\}, \tag{B.3}$$

where $\zeta_{k_1 k_2 k}(t)$ is a term of constant order along the training. The solution to the ODE in (B.3) provides a more high-level theoretical basis for the emergence of the key structural properties we identified in our work. Consider the case $k_1 = k_2 = k$, we have

$$\rho_{kkk}(t) = \sum_{m=1}^M z_{\theta km}^2(t) \cdot z_{\xi km}(t) \propto \sum_{m=1}^M \alpha_m^k(t)^2 \cdot \beta_m^k(t) \cdot \exp(i\{\psi_m^k(t) - 2\phi_m^k(t)\}) \overset{t \to \infty}{\longrightarrow} 1.$$

For this to hold, the imaginary part of $\rho_{kkk}(t)$ should converge to 0:

$$\Im(\rho_{kkk}(t)) \propto \sum_{m=1}^M \alpha_m^k(t)^2 \cdot \beta_m^k(t) \cdot \sin(\psi_m^k(t) - 2\phi_m^k(t)) \overset{t \to \infty}{\longrightarrow} 0.$$

This convergence is a direct consequence of the phase alignment dynamic $(2\phi_m^k(t) - \psi_m^k(t)) \bmod 2\pi \to 0$ as revealed in §5.1.2. Moreover, if we consider $k_1, k_2 \neq k$, then we have

$$\rho_{k_1 k_2 k}(t) \propto \sum_{m=1}^M \alpha_m^{k_1}(t) \cdot \alpha_m^{k_2}(t) \cdot \beta_m^k(t) \cdot \exp(i\{\psi_m^k(t) - \phi_m^{k_1}(t) - \phi_m^{k_2}(t)\}) \overset{t \to \infty}{\longrightarrow} 0.$$

A sufficient condition for this is that the product of amplitudes $\alpha_m^{k_1}(t) \cdot \alpha_m^{k_2}(t) \cdot \beta_m^k(t)$ goes to zero for all $m \in [M]$. This corresponds precisely to the single-frequency sparsity we observed in §D.1. Beyond these, Tian (2024) also discussed data with a general algebraic structure and its relationship with properties of global optimizers. Recently, Wang & Wang (2025) formalized these mean-field dynamics by modeling the network's parameters as a continuous distribution. This approach allows the training process to be rigorously described as a Wasserstein gradient flow on the measure space.

## C    CONCLUSION

In this paper, we provide an end-to-end reverse engineering of how two-layer neural networks learn modular addition, from training dynamics to the final learned model. First, we show that trained networks implement a majority-voting algorithm in the Fourier domain through phase alignment and model symmetry. Second, we explain how these features emerge from a lottery-like mechanism where frequencies compete within each neuron, with the winner determined by initial magnitude and phase misalignment. Third, we characterize grokking as a three-stage process where weight decay prunes non-feature frequencies, transforming a perturbed Fourier representation into a clean, generalizable solution. These findings offer insights into the dynamics of feature learning in neural networks, a mechanism that may extend to more general tasks.

## D    THEORETICAL EXTENSIONS

In this section, we extend the results from §5 to two more general scenarios: lottery mechanism under multi-frequency initialization in §D.1 and the dynamics with ReLU activation in §D.2.

### D.1    THEORETICAL UNDERPINNING OF LOTTERY TICKET MECHANISM

To understand why a single frequency pattern emerges from a random, multi-frequency initialization (Observation 1), we can analyze the training dynamics for each frequency within a specific neuron. The ODEs capture the dynamics of competition in (D.1), which are fully derived in §5.1.

$$\partial_t \alpha_m^0(t) \approx \partial_t \beta_m^0(t) \approx 0,$$

$$\partial_t \alpha_m^k(t) \approx 2p \cdot \alpha_m^k(t) \cdot \beta_m^k(t) \cdot \cos(\mathcal{D}_m^k(t)), \quad \partial_t \beta_m^k(t) \approx p \cdot \alpha_m^k(t)^2 \cdot \cos(\mathcal{D}_m^k(t)), \quad \text{(D.1)}$$

$$\partial_t \mathcal{D}_m^k(t) \approx -\left(4\beta_m^k(t) - \alpha_m^k(t)^2/\beta_m^k(t)\right) \cdot p \cdot \sin(\mathcal{D}_m^k(t)), \qquad \forall k \neq 0.$$

A key insight from these equations is that the dynamics are fully *decoupled*. The evolution of each frequency is *self-contained*, proceeding orthogonally without cross-frequency interaction. This structural independence establishes the competitive environment required for the lottery ticket mechanism. The ODEs also reveal a powerful *reinforcing dynamic*: the growth rate, proportional to the alignment term $\cos(\mathcal{D}_m^k(t))$, is amplified by the magnitudes This creates a "larger-grows-faster" positive feedback loop that drives the winner's dominance.

As introduced in §3.1.1, this process is not chaotic but is instead a predictable competition governed by a "Lottery Ticket Mechanism". Applying an ODE comparison lemma (Smith, 1995), we can compare the evolution of frequency magnitudes based on their initial conditions. This allows us to formally prove that the "lottery ticket" drawn at initialization determines which frequency will ultimately dominate. We formalize the results into the following corollary.

**Corollary D.1** (Informal). *Consider a multi-frequency initialization akin to Assumption 5.1. For a given dominance level $\varepsilon \in (0,1)$ and fixed neuron $m$, let $t_\varepsilon$ be the minimal time required for the winning frequency $k^\star$ to dominate all others, such that $\max_{k \neq k^\star} \beta_m^k(t)/\beta_m^\star(t) \leq \varepsilon$. Then, we have*

$$k^\star = \min_k \widetilde{\mathcal{D}}_m^k(0), \quad t_\varepsilon \lesssim \frac{\pi^2 p^{-(2c+3)}}{\kappa_{\text{init}}} + \frac{(c+1)\log p + \log \frac{1}{1-\varepsilon}}{p\kappa_{\text{init}} \cdot \{1 - 2c^2\pi^2 \cdot (\log p/p)^2\}},$$

*where the bound holds under mild conditions and with a high probability of at least $1 - \widetilde{\Theta}(p^{-c})$.*

The proof is deferred to §G.1. Corollary D.1 formalizes our Lottery Ticket Mechanism in Observation 6. It states that under a multi-frequency random initialization where all frequencies start with identical magnitudes, the frequency with the smallest initial misalignment $\widetilde{\mathcal{D}}_m^\star$ will inevitably dominate. This dominance occurs rapidly, on a timescale of $\widetilde{O}\left(\frac{\log p}{p\kappa_{\text{init}}}\right)$.

### D.2    DYNAMICS BEYOND QUADRATIC ACTIVATION

Thus far, we have focused on quadratic activation for more precise interpretation. However, experimental results indicate that quadratic activation is not *essential* or can be even *problematic*. In

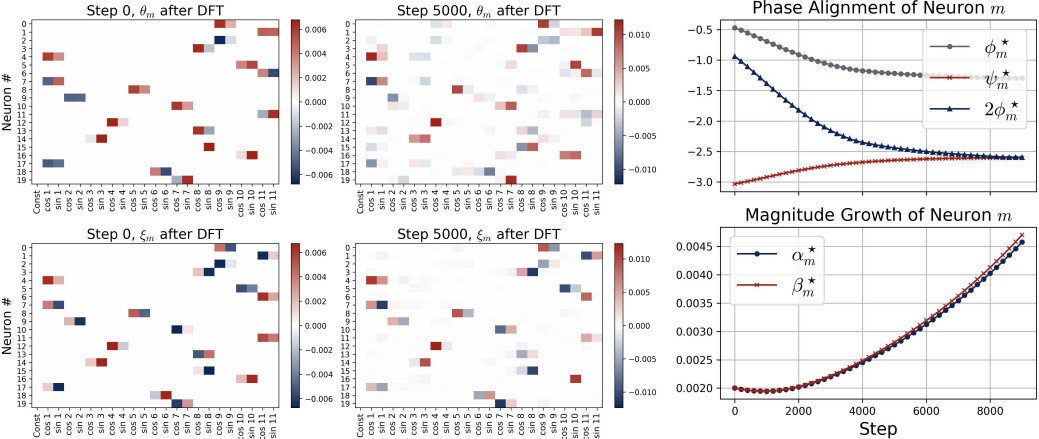

(a) Heatmaps of parameters after discrete Fourier transform for the first 20 neurons with ReLU activation at the intial stage.

(b) Dynamics of magnitude and phase for Neuron $m$ with ReLU activation.

Figure 8: Learned feature and dynamics of parameters initialized at Assumption 5.1 with $p = 23$ and ReLU activation. Figure (a) shows heatmaps of the parameters after DFT at initialization and at the end of the initial stage. Similar to the quadratic activation (see Figure 13), the single-frequency pattern is approximately maintained, with small values emerging at frequencies "$3k^\star$", "$5k^\star$" for $\theta_m$, and "$2k^\star$", "$3k^\star$" for $\xi_m$. Figure (b) plots the dynamics of a specific neuron $m$. Here, the phase quickly aligns, i.e., $\psi_m^\star \approx 2\phi_m^\star$, and the magnitudes $\alpha_m^\star$ and $\beta_m^\star$ grow rapidly and synchronously.

practice, quadratic activation often leads to *unstable training* and *inconsistent feature learning* when training from scratch.[2] In contrast, ReLU activation consistently leads to the emergence of desired features, as shown in §3. In this section, we investigate the training dynamics of ReLU activation.

**Training Dynamics of ReLU Activation.** In parallel, we adopt an experimental setup identical to that of Figure 13 using the single-frequency initialization specified in Assumption 5.1, with the only modification being the replacement of quadratic activation with ReLU activation. The experimental results are shown in Figure 8, and the key observation is summarized below.

> **Observation 7 (ReLU Leakage).** For ReLU activation, although each neuron is initialized with a single frequency $k^\star$, such a pattern is *preserved approximately* with *small leakage*, with small values emerging at other frequencies. For $\theta_m$, the values emerges at frequencies "$3k^\star$", "$5k^\star$" and higher *odd* multiples, with magnitudes decaying gradually. For $\xi_m$, these appear at "$2k^\star$", "$3k^\star$", and others, which also exhibit decay with increasing multiplicative factors.

As shown in Observation 5, ReLU mostly preserves the single-frequency pattern but still exhibits small leakage at other frequencies. For instance, in Figure 8a, Neuron 3 is initialized with dominant frequency 1. After 30,000 training steps, small values emerge at frequencies 3 and 5 in $\theta_m$, and at 2 and 3 in $\xi_m$. In what follows, we first formalize the multiplicative relationship among frequencies.

**Definition D.2.** *Given $k, \tau \in [\frac{p-1}{2}]$, we say frequency $\tau$ is $r$-fold multiple of $k$ under modulo $p$ if $\tau = rk \bmod p$ or $p - \tau = rk \bmod p$ for some $r \in [\frac{p-1}{2}]$, denoted by $\tau \stackrel{p}{=} rk$.*

Now we are ready to present the main proposition for training dynamics of ReLU activation.

**Proposition D.3.** *Consider gradient update with respect to the decoupled loss $\ell_m$ and assume that $(\theta_m, \xi_m)$ satisfying (3.1). Let $\Delta_v^k = \sqrt{\langle \nabla_v \ell_m, b_{2k} \rangle^2 + \langle \nabla_v \ell_m, b_{2k+1} \rangle^2}$ denote the incremental scale for frequency $k \in [\frac{p-1}{2}]$. Under the asymptotic regime where $p \to \infty$, it holds that*

*(i) $\Delta_{\theta_m}^k / \Delta_{\theta_m}^\star = \Theta(r_k^{-2})$ and $\Delta_{\xi_m}^k / \Delta_{\xi_m}^\star = \Theta(r_k^{-2}) \cdot \mathbb{1}(r \text{ is odd})$, where $k \stackrel{p}{=} r_k k^\star$;*

*(ii) $\mathscr{P}_{k^\star}^{\parallel} \nabla_v \ell_m \propto v$ for $v \in \{\theta_m, \xi_m\}$ when $\psi_m = 2\phi_m \bmod p$, $\mathscr{P}_k^{\parallel} = I - \sum_{j \geq 1, j \neq 2k^\star, 2k^\star+1} b_j b_j^\top$.*

See §G.2 for a detailed proof. Proposition D.3 indicates that, starting from a single-frequency point, the dynamics with respect to ReLU dynamics approximately preserve such a pattern. Specifically,

---

[2]The failure of the quadratic activation stems from the *significant disparity in growth rates* among neurons due to the nature of the quadratic function. Specifically, a few neurons with more well-aligned initial phases grow faster in magnitude and come to dominate the output, leaving an insufficient number of neurons to support diversification (see §4). This issue can be mitigated using techniques such as normalized GD (Cortés, 2006).

the gradient components at non-feature frequencies $k$ decay at a rate of $\Theta(r_k^{-2})$ compared with $k^\star$. If we exclude the small gradient components at other frequencies $k \neq k^\star$—by projecting $\nabla_v \ell_m$ onto the subspace spanned by $b_{2k^\star}$, and $b_{2k^\star+1}$—the resulting stationary point of the ReLU dynamic system remains $\psi_m = 2\phi_m \bmod p$, thereby explaining the convergence of aligned phases.

# E ADDITIONAL EXPERIMENTAL DETAILS AND RESULTS

## E.1 DETAILED INTERPRETATION OF GROKKING DYNAMICS IN SECTION 3.2

**Inverse Participation Ratio (IPR).** To quantitatively characterize the concentration of Fourier coefficients at a specific frequency $k$, or equivalently, the sparsity level of the learned parameters in the Fourier domain, we introduce the inverse participation ratio (IPR). This metric, originally used in physics as a localization measure (Kramer & MacKinnon, 1993), was recently adopted in Doshi et al. (2023) as a progress measure to understand the generalization behavior in machine learning. Specifically, given $\nu \in \mathbb{R}^d$, the IPR is defined as $\mathtt{IPR}(\nu) = (\|\nu\|_{2r}/\|\nu\|_2)^{2r}$ for some integer $r > 1$. We calculate the IPR for all $\{\theta_m\}_{m \in [M]}$ and $\{\xi_m\}_{m \in [M]}$, and take the average.

**Definition of Progress Measure.** Here, we provide a formal definition of the progress measure for grokking used in Figure 3, which is defined over the model output and parameters $\theta_m$'s and $\xi_m$'s.

- **Loss** :
$$\ell_{\mathcal{D}} = - \sum_{(x,y) \in \mathcal{D}} \big\langle \log \circ \mathtt{smax} \circ f(x,y;\xi,\theta), e_{(x+y) \bmod p} \big\rangle$$

- **Accuracy** :
$$\mathtt{Acc}_{\mathcal{D}} = \frac{1}{|\mathcal{D}|} \sum_{(x,y) \in \mathcal{D}} \mathbb{1}\big\{\mathrm{argmax}\big(\mathtt{smax} \circ f(x,y;\xi,\theta)\big) = (x+y) \bmod p\big\}$$

- **IPR** :
$$\mathtt{IPR}_{\theta,\xi} = \frac{1}{2M} \sum_{m=1}^{M} \left(\frac{\|B_p^\top \theta_m\|_4}{\|B_p^\top \theta_m\|_2}\right)^4 + \frac{1}{2M} \sum_{m=1}^{M} \left(\frac{\|B_p^\top \xi_m\|_4}{\|B_p^\top \xi_m\|_2}\right)^4$$

- $\ell_2$-**norm** :
$$\ell_2\text{-}\mathtt{norm}_{\theta,\xi} = \frac{1}{2M} \sum_{m=1}^{M} (\|\theta_m\|_2 + \|\xi_m\|_2)$$

**Three-Phase Dynamics of Grokking.** As discussed in §3.2, the grokking process is governed by the interplay between two primary forces: loss minimization and weight decay. The dynamics unfold across three major phases: an initial memorization stage dominated by the loss gradient, followed by two distinct generalization stages where the balance between these forces shifts. Below, we provide a more detailed account of each phase by examining our key progress measures.

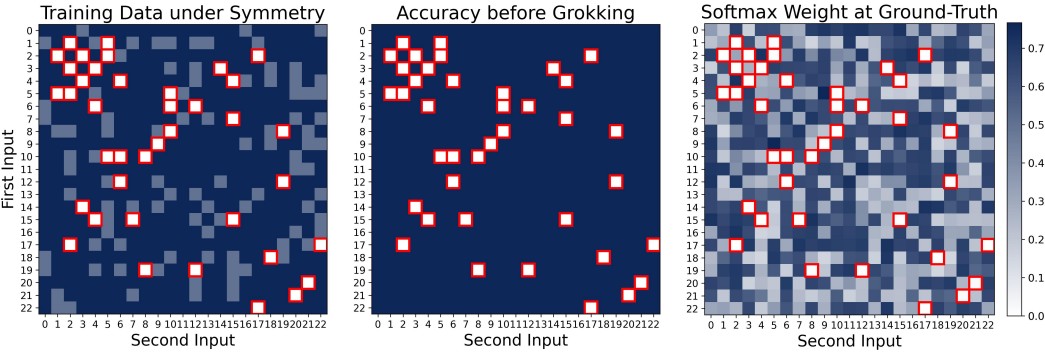

Figure 9: Heatmaps of trained model from Figure 3 at the end of the **memorization stage**. The left panel displays the data distribution: dark blue entries represent training data, light blue entries are test data whose symmetric counterparts are in the training set, and white entries (outlined in red) are the remaining held-out test data. The middle panel shows the model's accuracy, demonstrating that it has perfectly memorized all training data and their symmetric variants but completely fails to generalize to the held-out data. Finally, the right panel visualizes the model's post-softmax output on the correct answer for each data point, further confirming the accuracy results.

- **Phase I: Memorization.** Initially, the network quickly memorizes the training data, reaching 100% accuracy. Test accuracy also improves to around 70%, aided by the model's symmetric

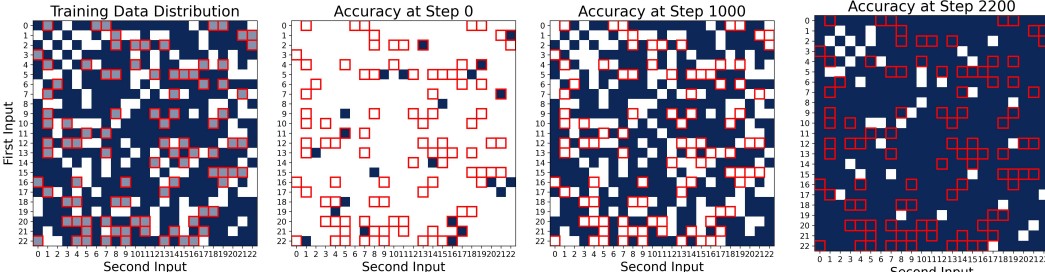

Figure 10: Data distribution during the memorization stage. The first panel illustrates the data partitioning, which, unlike in Figure 9, uses the following scheme: white entries denote test data, dark blue entries represent **common (symmetric)** training data, and light blue entries (outlined in red) denote **rare (asymmetric)** training data. The remaining three plots track the model's accuracy, demonstrating a two-stage memorization scheme. At initialization, the model performs at a low, chance-level accuracy. However, after approximately 1000 steps, it masters the common symmetric training data, but its performance on rare asymmetric data drops to zero, overwriting any initially correct random predictions. By the end of the memorization stage, the model finally memorizes these rare data points, achieving 100% training accuracy

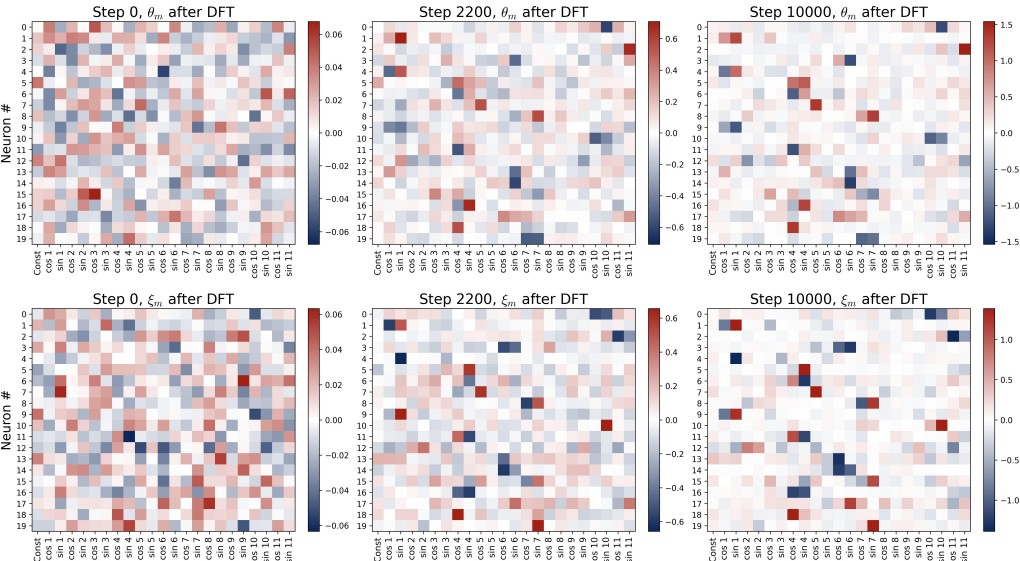

Figure 11: Heatmaps of parameters after applying discrete Fourier transform along training epochs for the first 20 neurons with $p = 23$ under train-test split setup. At the end of the memorization stage (step 2200), a single-frequency pattern has started to emerge, accompanied by noisy perturbations in other frequencies. This initial "perturbed Fourier solution" is subsequently refined, as weight decay prunes the noisy, non-feature frequencies to reveal the final, clean pattern.

architecture. Figure 9 provide clear empirical evidence for this perfect memorization. The model achieves flawless accuracy and high confidence on the training data (dark blue entries) and test data whose symmetric counterparts were part of the training set (light blue entries). Note that the model completely fails on the truly "unseen" held-out test data (white entries outlined in red), confirming it has learned to exploit symmetry rather than achieving true generalization at this stage. During this time, feature frequencies become roughly aligned (see Figure 3c) and their sparsity increases significantly (see Figure 3d). While these dynamics resemble a full-data setup, the incomplete data yields a *perturbed Fourier solution* that overfits the training set.

- **Phase II: Loss-Driven Norm Growth with Rapid Feature Cleanup.** After reaching perfect training accuracy, the model's parameters continue evolving to further reduce the loss. Instead of naively amplifying parameter magnitudes, weight decay actively steers their direction. As shown in Figure 3d, the dynamic is thus a balancing act: the loss gradient pushes to scale up parameters, while weight decay *prunes unnecessary frequencies* to decelerate the growth of norm.

- **Phase III: Slow Cleanup Driven Solely by Weight Decay.** By the end of Phase II, training loss is near-zero and test accuracy approaches 100%. Thus, in the final stage, the diminished loss gradient allows weight decay to dominate, causing the parameter norm to decrease (see Figure 3d). Without the main driving force of the loss, this final "cleanup" phase is extremely slow (see Figure 3b), during which test accuracy gradually converges to 100%.

### E.2 Ablations Studies for Fully-Diversified Parametrization

In this section, we present comprehensive ablation studies investigating the **efficiency** of the fully diversified parametrization as defined in Definition 4.1. We evaluate the models based on the CE loss defined in Equation 2.2 while maintaining a fixed, equivalent computational budget.

| PART I: FREQUENCY DIVERSITY ABLATION. | | | | | |
|---|---|---|---|---|---|
| Loss | 1 Freqs | 2 Freqs | 4 Freqs | 8 Freqs | Full Freqs |
| Avg. | 1.64 | $6.02 \times 10^{-1}$ | $2.88 \times 10^{-2}$ | $2.99 \times 10^{-8}$ | $7.41 \times 10^{-15}$ |
| Std. | $2.01 \times 10^{-2}$ | $8.79 \times 10^{-2}$ | $1.55 \times 10^{-2}$ | $1.07 \times 10^{-7}$ | – |
| PART II: PHASE DIVERSITY ABLATION. | | | | | |
| | $[0, 0.4\pi)$ | $[0, 0.8\pi)$ | $[0, 1.2\pi)$ | $[0, 1.6\pi)$ | $[0, 2\pi)$ |
| Loss | 4.82 | $2.00 \times 10^{-3}$ | $1.19 \times 10^{-9}$ | $3.54 \times 10^{-7}$ | $7.41 \times 10^{-15}$ |

Table 2: Performance of the predictor under different ablation configurations. For the frequency ablation study, the average and standard deviation of the loss are reported across all possible combinations of frequencies of the specified size $|\mathcal{K}|$. The results show that the fully diversified parametrization achieves the lowest CE loss, confirming its maximum efficiency under the fixed constraints of model scale $\alpha_m \beta_m^2 = 1$ and neuron budget $M = 128$.

All predictors share a fixed neuron constraint $M = 128$ and scale $\alpha_m \beta_m^2 = 1$ for all $m \in [M]$. The ablation is performed across two distinct dimensions of the diversification strategy:

- **Ablation of Frequency Diversification.** We examine the impact of restricting the number of learned frequencies. We use only a subset of frequencies $\mathcal{K} \subseteq [\frac{p-1}{2}]$ with $|\mathcal{K}| = \{1, 2, 4, 8\}$. The phases for each selected frequency $k$ are kept uniformly distributed over $[0, 2\pi)$.
- **Ablation of Phase Uniformity.** We investigate the effect of restricting the range of the phase distribution. The model utilizes the full set of frequencies, but the phase for each frequency is uniformly distributed over a restricted interval $[0, \iota\pi)$ with $\iota \in \{0.4, 0.8, 1.2, 1.6\}$.

The ablation study results in Table 2 confirm that full frequency and phase diversification is essential for maximizing parametrization efficiency under fixed constraints. Part I shows that the CE loss decreases rapidly as the number of frequencies increases, dropping from 1.64 at $|\mathcal{K}| = 1$ to $7.41 \times 10^{-15}$ for the full frequency set, underscoring the critical role of spectral richness. Part II reveals that restricting the phase distribution range significantly degrades performance. For instance, the loss is 4.82 for $[0, 0.4\pi)$ but achieves the minimum of $7.41 \times 10^{-15}$ only when the phases span the full $[0, 2\pi)$ interval. These findings collectively validate that the fully diversified parametrization achieves the maximum efficiency. Visually, this maximum efficiency is confirmed in Figure 12, where the fully diversified parametrization generates the highest confidence prediction by creating the largest logit gap between the ground truth label and all incorrect alternatives. Please refer to Figure 12 for visualizations of model outputs under different ablation configurations.

### E.3 Training Dynamics with Quadratic Activation

To under the training dynamics with quadratic activation, we set $p = 23$ and use a two-layer neural network with width $M = 512$. The network is trained using SGD optimizer with step size $\eta = 10^{-4}$, initialized under Assumption 5.1 with initial scale $\kappa_{\text{init}} = 0.02$.

As shown in Figure 13, a single-frequency pattern is preserved throughout the training process. This empirical result aligns with our theoretical findings in Theorem 5.2, which states that under a sufficiently small initialization, the single-frequency structure will remain stable during the initial stage of training. In other words, the neurons are fully decoupled and the main flow dominates.

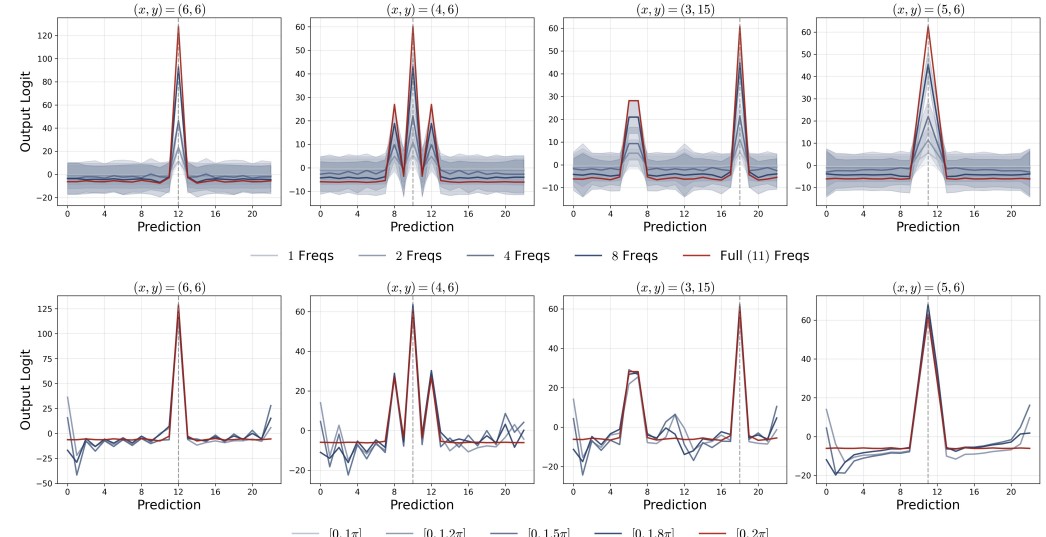

Figure 12: Output logits for the predictor under different ablation configurations, evaluated across four distinct query points $(x, y)$. The true prediction label is indicated by the dashed vertical line in each panel. The fully diversified parametrization yields the largest logit gap between the ground truth and incorrect labels, signifying maximal prediction confidence.

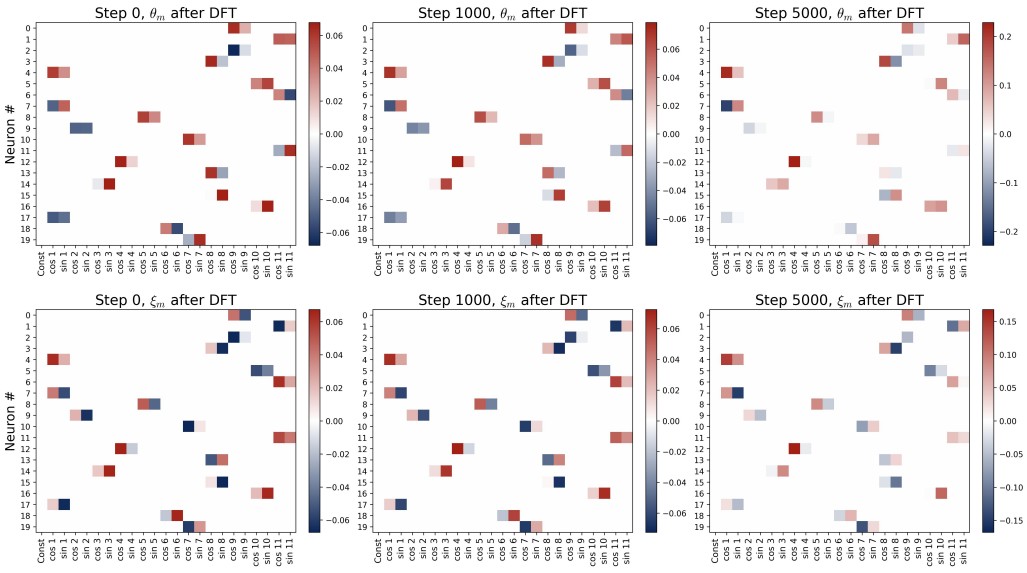

Figure 13: Heatmaps of parameters after applying discrete Fourier transform along training epochs for the first 20 neurons initialized under Assumption 5.1 with $p = 23$ and quadratic activation. At the initial stage, these neurons preserve the single-frequency pattern by evolving only the Fourier coefficients corresponding to the initial frequency $k^\star$, while keeping the others 0 throughout.

# F  PROOF OF RESULTS IN SECTION 4 AND 5

## F.1  PROOF OF PROPOSITION 4.2

We first introduce a useful lemma about the softmax operation.

**Lemma F.1.** *Let* $\nu \in \mathbb{R}^d$. *If* $i^* = \operatorname{argmax}_i \nu_i$ *and* $\nu_{i^*} - \nu_i \geq \tau$ *for all* $i \neq i^*$, *then*

$$\|\mathtt{smax}(\nu) - e_{i^*}\|_1 \leq \frac{d-1}{\exp(\tau) + (d-1)}.$$

*Proof of Lemma F.1.* See Lemma 3.6 in Chen & Li (2024) for a detailed proof. ☐

Now we are ready to present the proof of Proposition 4.2.

*Proof of Proposition 4.2.* Let $f^{[m]}$ be the logit contributed by neuron $m$, and fix $j \in [p]$. Under the parametrization in (3.1) and the phase-alignment condition $2\phi_m - \psi_m = 0 \mod 2\pi$, we have

$$f^{[m]}(x, y; \xi, \theta)[j]$$

$$= \alpha_m \beta_m^2 \cdot \cos(\omega_{\varphi(m)} j + 2\phi_m) \cdot \left( \cos(\omega_{\varphi(m)} x + \phi_m) + \cos(\omega_{\varphi(m)} y + \phi_m) \right)^2$$

$$= 2a \cdot \cos(\omega_{\varphi(m)}(x-y)/2)^2 \cdot \cos(\omega_{\varphi(m)} j + 2\phi_m) \cdot \{1 + \cos(\omega_{\varphi(m)}(x+y) + 2\phi_m)\}$$

$$= a \cdot \cos(\omega_{\varphi(m)}(x-y)/2)^2 \cdot \{2\cos(\omega_{\varphi(m)} j + 2\phi_m)$$

$$+ \cos(\omega_{\varphi(m)}(x+y-j)) + \cos(\omega_{\varphi(m)}(x+y+j) + 4\phi_m)\},$$

where the second equality uses the homogeneous scaling, i.e., condition (ii) in Definition 4.1. Next, summing over all neurons in the frequency-group $\mathcal{N}_k$, gives

$$\sum_{m \in \mathcal{N}_k} f^{[m]}(x, y; \xi, \theta)[j]$$

$$= a \cdot \cos(\omega_k(x-y)/2)^2 \cdot \underbrace{N \cdot \cos(\omega_k(x+y-j))}_{\text{condition (i): } |\mathcal{N}_k| = N}$$

$$+ a \cdot \cos(\omega_k(x-y)/2)^2 \cdot \underbrace{\sum_{m \in \mathcal{N}_k} \{2\cos(\omega_k j + 2\phi_m) + \cos(\omega_k(x+y+j) + 4\phi_m)\}}_{= 0 \text{ due to condition (iii)}}$$

$$= aN/2 \cdot \cos(\omega_k(x+y-j)) + aN/4 \cdot \{\cos(\omega_k(2x-j)) + \cos(\omega_k(2y-j))\}, \quad \text{(F.1)}$$

where the second equality follows from the balanced-frequency and the high-order phase-symmetry conditions (i) and (iii) in Definition 4.1. Summing (F.1) over all frequency $k$ yields

$$f(x, y; \xi, \theta)[j] = \sum_{k=1}^{(p-1)/2} \sum_{m \in \mathcal{N}_k} f^{[m]}(x, y; \xi, \theta)[j]$$

$$= aN/2 \cdot \sum_{k=1}^{(p-1)/2} \cos(\omega_k(x+y-j))$$

$$+ aN/4 \cdot \left\{ \sum_{k=1}^{(p-1)/2} \cos(\omega_k(2x-j)) + \sum_{k=1}^{(p-1)/2} \cos(\omega_k(2y-j)) \right\}. \quad \text{(F.2)}$$

By symmetry, for any fixed $z \in \mathbb{N}$, $\sum_{k=1}^{(p-1)/2} \cos(\omega_k z) = (p-1)/2$ if $z = 0 \mod p$ else $-1/2$. Then,

$$\sum_{k=1}^{(p-1)/2} \cos(\omega_k z) = -\frac{1}{2} + \frac{p}{2} \cdot \mathbb{1}(z \mod p = 0). \quad \text{(F.3)}$$

Thus, by combining (F.2) and (F.3), we can conclude that

$$f(x, y; \xi, \theta)[j] = aN/2 \cdot \left\{ -1 + p/2 \cdot \mathbb{1}(x+y \mod p = j) + p/4 \cdot \sum_{z \in \{x,y\}} \mathbb{1}(2z \mod p = j) \right\}, \quad \forall j \in [p].$$

Note that when $x \neq y$, the true-signal logit at $j = (x+y) \mod p$ exceeds all others by $aNp/8$, and when $x = y$, the margin is even larger. Applying Lemma F.1 yields

$$\|\text{smax} \circ f(x, y; \xi, \theta) - e_{(x+y) \mod p}\|_1 \leq \frac{p-1}{\exp(aNp/8) + p - 1} \leq p \cdot \exp(-aNp/8).$$

Hence, to achieve error $\epsilon$, it suffices to choose $a \gtrsim (Np)^{-1} \cdot \log(p/\epsilon)$, which completes the proof. $\square$

## F.2 PRELIMINARY: GRADIENT COMPUTATION

Recall the logit of the two-layer neural network in (2.1) takes the form:

$$f(x, y) := f(x, y; \xi, \theta) = \sum_{m=1}^{M} \xi_m \cdot \sigma(\langle e_x + e_y, \theta_m \rangle) \in \mathbb{R}^p. \quad \text{(F.4)}$$

For theoretical analysis, we consider the training dynamics over the full dataset $\mathcal{D}_{\text{full}} = \{(x, y, z) \mid x, y \in \mathbb{Z}_p, z = (x + y) \bmod p\}$ and the corresponding CE loss, defined in (2.2), can be written as

$$\ell := \ell(\xi, \theta; \mathcal{D}_*) = - \sum_{x \in \mathbb{Z}_p} \sum_{y \in \mathbb{Z}_p} \big\langle \log \circ \mathtt{smax} \circ f(x, y; \xi, \theta), e_{(x+y) \bmod p} \big\rangle$$

$$= - \sum_{x \in \mathbb{Z}_p} \sum_{y \in \mathbb{Z}_p} \log \left( \frac{\exp(f(x, y)[(x + y) \bmod p])}{\sum_{j=1}^{p} \exp(f(x, y)[j])} \right)$$

$$= \underbrace{- \sum_{x \in \mathbb{Z}_p} \sum_{y \in \mathbb{Z}_p} f(x, y)[(x + y) \bmod p]}_{:= \widetilde{\ell}} + \underbrace{\sum_{x \in \mathbb{Z}_p} \sum_{y \in \mathbb{Z}_p} \log \left( \sum_{j=1}^{p} \exp(f(x, y)[j]) \right)}_{:= \bar{\ell}}.$$

(F.5)

Following the loss decomposition in (F.5), we compute the gradients of these two parts respectively. Recall that the two-layer neural network is parametrized by $\xi = \{\xi_m\}_{m \in [M]}$ and $\theta = \{\theta_m\}_{m \in [M]}$ with $\xi_m, \theta_m \in \mathbb{R}^p$. By substituting the form of $f$ in (F.4) into $\widetilde{\ell}$ and $\bar{\ell}$, we have

$$\widetilde{\ell} = - \sum_{x \in \mathbb{Z}_p} \sum_{y \in \mathbb{Z}_p} \sum_{m=1}^{M} \xi_m[(x + y) \bmod p] \cdot \sigma(\langle e_x + e_y, \theta_m \rangle),$$

$$\bar{\ell} = \sum_{x \in \mathbb{Z}_p} \sum_{y \in \mathbb{Z}_p} \log \left( \sum_{j=1}^{p} \exp \left( \sum_{m=1}^{M} \xi_m[j] \cdot \sigma(\langle e_x + e_y, \theta_m \rangle) \right) \right).$$

Fix a neuron $m \in [M]$. First, we calculate the gradients for $\widetilde{\ell}$. By direct calculation, we have

$$\nabla_{\xi_m} \widetilde{\ell} = - \sum_{x \in \mathbb{Z}_p} \sum_{y \in \mathbb{Z}_p} e_{(x+y) \bmod p} \cdot \sigma(\langle e_x + e_y, \theta_m \rangle).$$

Following this, the entry-wise derivative with respect to $\xi_m[j]$ satisfies that

$$\frac{\partial \widetilde{\ell}}{\partial \xi_m[j]} = - \sum_{x, y \in \mathbb{Z}_p : (x+y) \bmod p = j} \sigma(\langle e_x + e_y, \theta_m \rangle) := - \sum_{(x, y) \in \mathcal{S}_j^p} \sigma(\langle e_x + e_y, \theta_m \rangle). \qquad \text{(F.6)}$$

Here, we define $\mathcal{S}_j^p = \{x, y \in \mathbb{Z}_p : (x + y) \bmod p = j\}$ for notational simplicity. Similarly, we can compute the gradient with respect to $\theta_m$, following that

$$\nabla_{\theta_m} \widetilde{\ell} = - \sum_{x \in \mathbb{Z}_p} \sum_{y \in \mathbb{Z}_p} \xi_m[(x + y) \bmod p] \cdot (e_x + e_y) \cdot \sigma'(\langle e_x + e_y, \theta_m \rangle)$$

$$= -2 \sum_{x \in \mathbb{Z}_p} e_x \cdot \sum_{y \in \mathbb{Z}_p} \xi_m[(x + y) \bmod p] \cdot \sigma'(\langle e_x + e_y, \theta_m \rangle),$$

where the last equality uses the symmetry of $x$ and $y$. Hence, the entry-wise derivative follows

$$\frac{\partial \widetilde{\ell}}{\partial \theta_m[j]} = -2 \sum_{x \in \mathbb{Z}_p} \xi_m[m_p(x, j)] \cdot \sigma'(\langle e_x + e_j, \theta_m \rangle), \qquad \text{(F.7)}$$

where we re-index $x = j$ and $y \to x$ to simplify the form. Next, we compute the gradients for $\bar{\ell}$. Following a similar argument in (F.7) and (F.6), based on the chain rule, it holds that

$$\frac{\partial \bar{\ell}}{\partial \xi_m[j]} = \sum_{x \in \mathbb{Z}_p} \sum_{y \in \mathbb{Z}_p} \frac{\exp \left( \sum_{m=1}^{M} \xi_m[j] \cdot \sigma(\langle e_x + e_y, \theta_m \rangle) \right)}{\sum_{i=1}^{p} \exp \left( \sum_{m=1}^{M} \xi_m[i] \cdot \sigma(\langle e_x + e_y, \theta_m \rangle) \right)} \cdot \sigma(\langle e_x + e_y, \theta_m \rangle). \qquad \text{(F.8)}$$

In addition, by direct calculation, we can obtain that

$$\frac{\partial \bar{\ell}}{\partial \theta_m[j]} = 2 \sum_{x \in \mathbb{Z}_p} \sum_{\tau=1}^{p} \frac{\exp \left( \sum_{m=1}^{M} \xi_m[\tau] \cdot \sigma(\langle e_x + e_j, \theta_m \rangle) \right)}{\sum_{i=1}^{p} \exp \left( \sum_{m=1}^{M} \xi_m[i] \cdot \sigma(\langle e_x + e_j, \theta_m \rangle) \right)} \cdot \xi_m[\tau]$$

$$\cdot \sigma'(\langle e_x + e_j, \theta_m \rangle), \qquad \text{(F.9)}$$

where the last equality results from re-indexing $x = j$, $y \to x$, and $j \to i$. Throughout the section, we consider quadratic activation $\sigma(x) = x^2$ for theoretical convenience.

### F.3 MAIN FLOW APPROXIMATION UNDER SMALL PARAMETER SCALING

The key property used in Stage I is that the scale of parameters is relatively small due to the small initialization and sufficiently small constant $a$. Following this, we have the approximation below:

$$\big(\texttt{smax}\circ f(x,y;\xi,\theta)\big)[j] = \frac{\exp\big(\sum_{m=1}^{M}\xi_m[j]\cdot\langle e_x+e_y,\theta_m\rangle^2\big)}{\sum_{i=1}^{p}\exp\big(\sum_{m=1}^{M}\xi_m[i]\cdot\langle e_x+e_y,\theta_m\rangle^2\big)} \approx \frac{1}{p}, \qquad \forall j\in[p]. \quad \text{(F.10)}$$

To formalize the approximation above, we introduce the following approximation error terms:

$$\mathsf{Err}_{m,j}^{(1)} = \sum_{x\in\mathbb{Z}_p}\sum_{y\in\mathbb{Z}_p}\left(\frac{\exp\big(\sum_{m=1}^{M}\xi_m[j]\cdot\langle e_x+e_y,\theta_m\rangle^2\big)}{\sum_{i=1}^{p}\exp\big(\sum_{m=1}^{M}\xi_m[i]\cdot\langle e_x+e_y,\theta_m\rangle^2\big)}-\frac{1}{p}\right)\cdot\langle e_x+e_y,\theta_m\rangle^2,$$

$$\mathsf{Err}_{m,j}^{(2)} = 2\sum_{x\in\mathbb{Z}_p}\sum_{\tau=1}^{p}\left(\frac{\exp\big(\sum_{m=1}^{M}\xi_m[\tau]\cdot\langle e_x+e_j,\theta_m\rangle^2\big)}{\sum_{i=1}^{p}\exp\big(\sum_{m=1}^{M}\xi_m[i]\cdot\langle e_x+e_j,\theta_m\rangle^2\big)}-\frac{1}{p}\right)\cdot\xi_m[\tau]\cdot\langle e_x+e_y,\theta_m\rangle,$$

for all $(j,m)\in[p]\times[M]$. The approximation result is formalized in the following lemma.

**Lemma F.2.** *Denote $\|\theta\|_\infty = \max_m\|\theta_m\|_\infty$ and $\|\xi\|_\infty = \max_{m]}\|\xi_m\|_\infty$. For all $(j,m)\in[p]\times[M]$, the approximation error is upper bounded by*

$$|\mathsf{Err}_{m,j}^{(1)}|\vee|\mathsf{Err}_{m,j}^{(2)}| \le 8p\cdot\|\theta_m\|_\infty\cdot\max\{\|\xi_m\|_\infty,\|\theta_m\|_\infty\}\cdot(\exp(8M\cdot\|\xi\|_\infty\cdot\|\theta\|_\infty^2)-1).$$

*Proof of Lemma F.2.* Let $s_j(x,y) = \sum_{m=1}^{M}\xi_m[j]\cdot\langle e_x+e_y,\theta_m\rangle^2$ denote the score given by the neural network for the $j$-th entry. Then, for fixed $(x,y)$, the softmax vector for $j$-th entry is given by $p(x,y)[j] = \exp(s(x,y)[j])/\sum_{i=1}^{p}\exp(s(x,y)[i])$. Note that, for any $(m,j)\in[M]\times[p]$, we have

$$|\mathsf{Err}_{m,j}^{(1)}| = \sum_{x\in\mathbb{Z}_p}\sum_{y\in\mathbb{Z}_p}\left(p(x,y)[j]-\frac{1}{p}\right)\cdot\langle e_x+e_y,\theta_m\rangle^2$$

$$\le 4p^2\cdot\max_{(x,y)\in\mathbb{Z}_p^2}\left|p(x,y)[j]-\frac{1}{p}\right|\cdot\|\theta_m\|_\infty^2. \quad \text{(F.11)}$$

Let $\Delta_{x,y} = \max_{j\in[p]}s(x,y)[j]-\min_{j\in[p]}s(x,y)[j]>0$ for any $(x,y)\in\mathbb{Z}_p^2$. It is straightforward to see that $\Delta_{x,y}$ can be effectively bounded by the scales of $\theta_m$'s and $\xi_m$'s, following that

$$\Delta_{x,y}\le 2\left\|\sum_{m=1}^{M}\xi_m[j]\cdot\langle e_x+e_y,\theta_m\rangle^2\right\|_\infty \le 8M\cdot\|\xi\|_\infty\cdot\|\theta\|_\infty^2, \qquad \forall(x,y)\in\mathbb{Z}_p^2. \quad \text{(F.12)}$$

Following this, we upper bound the difference between the softmax-induced distribution and the uniform distribution using the small-scale score vector. By simple algebra, we can show that

$$\max_{j\in[p]}\left|p(x,y)[j]-\frac{1}{p}\right| \le \max_{j\in[p]}\left|p(x,y)[j]-\frac{1}{p}\right|\bigvee\min_{j\in[p]}\left|p(x,y)[j]-\frac{1}{p}\right|$$

$$\le \left|\frac{1}{1+(p-1)\cdot\exp(-\Delta_{x,y})}-\frac{1}{p}\right|\bigvee\left|\frac{1}{1+(p-1)\cdot\exp(\Delta_{x,y})}-\frac{1}{p}\right|$$

$$= \frac{p-1}{p}\cdot\left\{\frac{\exp(\Delta_{x,y})-1}{\exp(\Delta_{x,y})+p-1}\bigvee\frac{1-\exp(-\Delta_{x,y})}{\exp(-\Delta_{x,y})+p-1}\right\}$$

$$\le \frac{1}{p}\cdot(\exp(\Delta_{x,y})-1)\cdot\max\{\exp(-\Delta_{x,y}),1\}$$

$$\le \frac{1}{p}\cdot(\exp(\Delta_{x,y})-1). \quad \text{(F.13)}$$

By combining (F.11), (F.12) and (F.13), we can reach the conclusion that

$$|\mathsf{Err}_{m,j}^{(1)}| \le 4p\cdot\|\theta_m\|_\infty^2\cdot(\exp(8M\cdot\|\xi\|_\infty\cdot\|\theta\|_\infty^2)-1).$$

Building upon a similar argument, it holds that

$$|\mathsf{Err}_{m,j}^{(2)}| = 2\sum_{x\in\mathbb{Z}_p}\sum_{\tau=1}^{p}\left|p(x,j)[\tau]-\frac{1}{p}\right|\cdot\xi_m[\tau]\cdot\langle e_x+e_y,\theta_m\rangle$$

$$\le 8p\cdot\|\theta_m\|_\infty\cdot\|\xi_m\|_\infty\cdot(\exp(8M\cdot\|\xi\|_\infty\cdot\|\theta\|_\infty^2)-1).$$

Hence, we complete the proof of bounded approximation error. □

Lemma F.2 formalizes a key technical tool for analyzing the dynamics during the initial stage: given small-scale parameters $\theta_m$'s and $\xi_m$'s, and a specified small constant $a \in \mathbb{R}$ (introduced for technical convenience), the softmax components in the gradient can be effectively approximated by a uniform vector, with a controllable and small approximation error.

In the following sections, we denote $\mathsf{Err}_m^{(1)} = (\mathsf{Err}_{m,j}^{(1)})_{j \in [p]} \in \mathbb{R}^p$ and $\mathsf{Err}_m^{(2)} = (\mathsf{Err}_{m,j}^{(2)})_{j \in [p]} \in \mathbb{R}^p$ for notational simplicity and we remark that the error vectors would vary along the grdient flow.

### F.3.1 Proof Overview: Simplified Dynamics under Approximation

Before delving into the technical details, we provide a brief summary of the approximate dynamics of parameters and their transformations along gradient flow in Table 3. This overview characterizes the training during the initial phase, when parameter magnitudes are small. We use $\approx$ to highlight the central flow, omitting the perturbations introduced by approximation errors as defined in (F.10). The simplification of the approximate dynamics leverages two key features that arise under the specialized initialization in Assumption 5.1: *neuron-wise decoupled* loss landscape—meaning the evolution of each neuron depends only on itself—and preservation of a *single-frequency* structure—i.e., the parameters exhibit only one frequency component in the Fourier domain. These properties hold during the early stage of training. Refer to §5.1 for a detailed illustration and proof sketch. With slight abuse of notation, we let $k^\star$ denote the initial frequency of each neuron and we use the superscript $\star$ instead of $k^\star$ to simplify the notation in Table 3.

**Roadmap.** In Part I, we present the dynamics of parameters—$\{\theta_m\}_{m \in [M]}$ and $\{\xi_m\}_{m \in [M]}$ with calculation details provided in §F.3.2 and §F.4. In Part II, building on the results from §F.4, we shift focus to the dynamics of the discrete Fourier coefficients, defined in §A.3, to better understand the evolution of parameters in the Fourier domain. Finally, based on the results in Part I and Part II, we analyze the dynamics of the magnitudes and phases of the Fourier signals (see §A.3 for definitions), to interpret the alignment behavior between $\theta_m$ and $\xi_m$, and the detailed derivations are provided in §F.5. The auxiliary equalities naturally arise from the definition of discrete Fourier coefficients and their transformations.

---

**PART I: DYNAMICS OF ORIGINAL PARAMETERS.**

| | |
|---|---|
| $\theta_m[j](t)$ | $\partial_t \theta_m[j](t) \approx 2p \cdot \alpha_m^\star(t) \cdot \beta_m^\star(t) \cdot \cos(\omega_k j + \psi_m^\star(t) - \phi_m^\star(t))$ |
| $\xi_m[j](t)$ | $\partial_t \xi_m[j](t) \approx p \cdot \alpha_m^\star(t)^2 \cdot \cos(\omega_{k^\star} j + 2\phi_m^\star(t))$ |

**PART II: DYNAMICS OF DISCRETE FOURIER COEFFICIENTS.**

| | |
|---|---|
| $g_m[2k^\star](t)$ | $\partial_t g_m[2k^\star](t) \approx \sqrt{2} \cdot p^{3/2} \cdot \alpha_m^\star(t) \cdot \beta_m^\star(t) \cdot \cos\left(\psi_m^\star(t) - \phi_m^\star(t)\right)$ |
| $g_m[2k^\star + 1](t)$ | $\partial_t g_m[2k^\star + 1](t) \approx -\sqrt{2} \cdot p^{3/2} \cdot \alpha_m^\star(t) \cdot \beta_m^\star(t) \cdot \sin\left(\psi_m^\star(t) - \phi_m^\star(t)\right)$ |
| $r_m[2k^\star](t)$ | $\partial_t r_m[2k^\star](t) \approx p^{3/2}/\sqrt{2} \cdot \alpha_m^\star(t)^2 \cdot \cos\left(2\phi_m^\star(t)\right)$ |
| $r_m[2k^\star + 1](t)$ | $\partial_t r_m[2k^\star + 1](t) \approx -p^{3/2}/\sqrt{2} \cdot \alpha_m^\star(t)^2 \cdot \sin\left(2\phi_m^\star(t)\right)$ |

**PART III: DYNAMICS OF MAGNITUDES AND PHASES.**

| | |
|---|---|
| $\alpha_m^\star(t)$ | $\partial_t \alpha_m^\star(t) \approx 2p \cdot \alpha_m^\star(t) \cdot \beta_m^\star(t) \cdot \cos\left(2\phi_m^\star(t) - \psi_m^\star(t)\right)$ |
| $\beta_m^\star(t)$ | $\partial_t \beta_m^\star(t) \approx p \cdot \alpha_m^\star(t)^2 \cdot \cos\left(2\phi_m^\star(t) - \psi_m^\star(t)\right)$ |
| $\phi_m^\star(t)$ | $\partial_t \exp(i\phi_m^\star(t)) \approx 2p \cdot \beta_m^\star(t) \cdot \sin\left(2\phi_m^\star(t) - \psi_m^\star(t)\right) \cdot \exp\left(i\left\{\phi_m^\star(t) - \pi/2\right\}\right)$ |
| $\psi_m^\star(t)$ | $\partial_t \exp(i\psi_m^\star(t)) \approx p \cdot \frac{\alpha_m^\star(t)^2}{\beta_m^\star(t)} \cdot \sin\left(2\phi_m^\star(t) - \psi_m^\star(t)\right) \cdot \exp\left(i\left\{\psi_m^\star(t) + \pi/2\right\}\right)$ |
| $\mathcal{D}_m^\star(t)$[1] | $\partial_t \exp(i\mathcal{D}_m^\star(t)) \approx p \cdot \left(4\beta_m^\star(t) + \frac{\alpha_m^\star(t)^2}{\beta_m^\star(t)}\right) \cdot \sin(\mathcal{D}_m^\star(t)) \cdot \exp\left(i\left\{\mathcal{D}_m^\star(t) - \pi/2\right\}\right)$ |

**PART IV: AUXILIARY EQUALITIES.**

$$\cos(\phi_m^\star(t)) = \sqrt{2/p} \cdot g_m[2k^\star](t)/\alpha_m^\star(t), \qquad \sin(\phi_m^\star(t)) = -\sqrt{2/p} \cdot g_m[2k^\star + 1](t)/\alpha_m^\star(t),$$
$$\cos(\psi_m^\star(t)) = \sqrt{2/p} \cdot r_m[2k^\star](t)/\beta_m^\star(t), \qquad \sin(\psi_m^\star(t)) = -\sqrt{2/p} \cdot r_m[2k^\star + 1](t)/\beta_m^\star(t).$$

---

[1] We use $\mathcal{D}_m^\star(t)$ denote the phase misalignment level defined as $\mathcal{D}_m^\star(t) = 2\phi_m^\star(t) - \psi_m^\star(t) \mod 2\pi$.

Table 3: Summarization of the approximate dynamics during the initial stage. Please refer to §F.3.2, §F.4 and §F.5 for formalized arguments and detailed derivations.

### F.3.2 PROOF OF LEMMA F.3: MAIN FLOW OF DECOUPLED NEURONS

**Lemma F.3** (Main Flow). *Consider the discrete Fourier coefficients, as well as the signal magnitudes and phases, defined over $\{\theta_m\}_{m \in [M]}$ and $\{\xi_m\}_{m \in [M]}$ (see §A.3 for definitions). Then, at each time $t \in \mathbb{R}^+$ and $m \in [M]$, the gradient dynamics takes the following form:*

$$\partial_t \xi_m[j](t) = p \cdot \sum_{k=1}^{(p-1)/2} \alpha_m^k(t)^2 \cdot \cos(\omega_k j + 2\phi_m^k(t)) - \mathsf{Err}_{m,j}^{(1)}(t),$$

$$\partial_t \theta_m[j](t) = 2p \cdot \sum_{k=1}^{(p-1)/2} \alpha_m^k(t) \cdot \beta_m^k(t) \cdot \cos(\omega_k j + \psi_m^k(t) - \phi_m^k(t))$$

$$+ 2p \cdot \beta_m^0(t) \cdot \sum_{k=1}^{(p-1)/2} \alpha_m^k \cdot \cos(\omega_k j + \phi_m^k(t)) - \mathsf{Err}_{m,j}^{(2)}(t),$$

*where the approximation errors $\mathsf{Err}_{m,j}^{(1)}(t)$ and $\mathsf{Err}_{m,j}^{(2)}(t)$ are defined in §F.3*

Lemma F.3 indicates that the dynamics of $\theta_m(t)$'s and $\xi_m(t)$'s only depend on $\theta_m(t)$ and $\xi_m(t)$ such that the neurons are almost fully decoupled with small approximation errors.

*Proof of Lemma F.3.* Consider a fixed neuron $m$. By combining the gradient computations in (F.6) and (F.8), we can write the complete form of derivative of loss $\ell$ with respect to $\xi_m[j]$ as

$$\frac{\partial \ell}{\partial \xi_m[j]} = \frac{\partial \widetilde{\ell}}{\partial \xi_m[j]} + \frac{\partial \bar{\ell}}{\partial \xi_m[j]}$$

$$= -\sum_{(x,y) \in \mathcal{S}_j^p} \sigma(\langle e_x + e_y, \theta_m \rangle) + \frac{1}{p} \cdot \sum_{x \in \mathbb{Z}_p} \sum_{y \in \mathbb{Z}_p} \sigma(\langle e_x + e_y, \theta_m \rangle) + \mathsf{Err}_{m,j}^{(1)}. \qquad \text{(F.14)}$$

Similarly, by combining (F.7) and (F.9), we have the derivative of $\ell$ with respect to $\theta_m[j]$:

$$\frac{\partial \ell}{\partial \theta_m[j]} = \frac{\partial \widetilde{\ell}}{\partial \theta_m[j]} + \frac{\partial \bar{\ell}}{\partial \theta_m[j]} = -2 \sum_{x \in \mathbb{Z}_p} \xi_m[m_p(x,j)] \cdot \sigma'(\langle e_x + e_j, \theta_m \rangle)$$

$$+ \frac{2}{p} \cdot \sum_{x \in \mathbb{Z}_p} \sum_{\tau=1}^{p} \xi_m[\tau] \cdot \sigma'(\langle e_x + e_j, \theta_m \rangle) + \mathsf{Err}_{m,j}^{(2)}. \qquad \text{(F.15)}$$

Motivated by Lemma F.3, we focus on the dominant terms of the gradient and carefully manage the error terms to characterize the central flow that determines the main dynamics in the initial stage.

**Step 1: Deriving Gradient of $\xi_m$.** By switching from the standard canonical basis to the Fourier basis, we can write $\theta_m$ using a form of discrete Fourier expansion, as shown in (3.2). Then, we have

$$\sum_{(x,y) \in \mathcal{S}_j^p} \sigma(\langle e_x + e_y, \theta_m \rangle) = \sum_{(x,y) \in \mathcal{S}_j^p} \left( 2\alpha_m^0 + \sum_{k=1}^{(p-1)/2} \alpha_m^k \sum_{z \in \{x,y\}} \cos(\omega_k z + \phi_m^k) \right)^2$$

$$= 4p \cdot (\alpha_m^0)^2 + \sum_{k=1}^{(p-1)/2} (\alpha_m^k)^2 \cdot \mathbf{(i)} + \sum_{1 \le k \ne \tau \le (p-1)/2} \alpha_m^k \alpha_m^\tau \cdot \mathbf{(ii)}$$

$$+ 2\alpha_m^0 \cdot \sum_{k=1}^{(p-1)/2} \alpha_m^k \cdot \mathbf{(iii)}. \qquad \text{(F.16)}$$

where we denote each term as

$$\mathbf{(i)} = \sum_{(x,y) \in \mathcal{S}_j^p} \left( \sum_{z \in \{x,y\}} \cos(\omega_k z + \phi_m^k) \right)^2,$$

$$\mathbf{(ii)} = \sum_{(x,y) \in \mathcal{S}_j^p} \sum_{z \in \{x,y\}} \cos(\omega_k z + \phi_m^k) \cdot \sum_{z \in \{x,y\}} \cos(\omega_\tau z + \phi_m^\tau),$$

$$\mathbf{(iii)} = \sum_{(x,y) \in \mathcal{S}_j^p} \sum_{z \in \{x,y\}} \cos(\omega_k z + \phi_m^k).$$

In the following, we compute **(i)**, **(ii)** and **(iii)** respectively using trigonometric identities and the periodicity of the module addition task over the full space $\mathbb{Z}_p^2$. First, note that

$$\textbf{(i)} = 2 \sum_{x \in \mathbb{Z}_p} \cos(\omega_k x + \phi_m^k)^2 + 2 \sum_{(x,y) \in \mathcal{S}_j^p} \cos(\omega_k x + \phi_m^k) \cdot \cos(\omega_k y + \phi_m^k)$$

$$= p + \sum_{x \in \mathbb{Z}_p} \cos(\omega_{2k} x + 2\phi_m^k) + \sum_{(x,y) \in \mathcal{S}_j^p} \cos(\omega_k(x+y) + 2\phi_m^k) + \sum_{(x,y) \in \mathcal{S}_j^p} \cos(\omega_k(x-y))$$

$$= p \cdot (1 + \cos(\omega_k j + 2\phi_m^k)), \tag{F.17}$$

where the last equality uses the fact that $\sum_{(x,y) \in \mathcal{S}_j^p} \cos(\omega_k(x-y)) = \sum_{x \in \mathbb{Z}_p} \cos(\omega_k x) = 0$ and $\cos(\omega_k(x+y) + 2\phi_m^k) = \cos(\omega_k j + 2\phi_m^k)$ for all $(x,y) \in \mathcal{S}_j^p$. Similarly, we have

$$\textbf{(ii)} = 2 \sum_{x \in \mathbb{Z}_p} \cos(\omega_k x + \phi_m^k) \cdot \cos(\omega_\tau x + \phi_m^\tau) + 2 \sum_{(x,y) \in \mathcal{S}_j^p} \cos(\omega_k x + \phi_m^k) \cdot \cos(\omega_\tau y + \phi_m^\tau)$$

$$= \sum_{(x,y) \in \mathcal{S}_j^p} \cos((\omega_k x + \omega_\tau y) + \phi_m^k + \phi_m^\tau) + \sum_{(x,y) \in \mathcal{S}_j^p} \cos((\omega_k x - \omega_\tau y) + \phi_m^k - \phi_m^\tau)$$

$$+ \sum_{x \in \mathbb{Z}_p} \cos((\omega_k + \omega_\tau)x + \phi_m^k + \phi_m^\tau) + \sum_{x \in \mathbb{Z}_p} \cos((\omega_k - \omega_\tau)x + \phi_m^k - \phi_m^\tau) = 0, \tag{F.18}$$

where we use $\sum_{(x,y) \in \mathcal{S}_j^p} \cos((\omega_k x + \omega_\tau y) + \phi_m^k + \phi_m^\tau) = \sum_{x \in \mathbb{Z}_p} \cos((\omega_k - \omega_\tau) + \omega_\tau j + \phi_m^k + \phi_m^\tau)$ in the last inequality for the first term and a similar arguent for the second one. In addition, it is easy to show that **(iii)** $= 2 \sum_{x \in \mathbb{Z}_p} \cos(\omega_k x + \phi_m^k) = 0$. By combining (F.16), (F.17), (F.18), we have

$$\sum_{(x,y) \in \mathcal{S}_j^p} \sigma(\langle e_x + e_y, \theta_m \rangle) = 4p \cdot (\alpha_m^0)^2 + p \cdot \sum_{k=1}^{(p-1)/2} (\alpha_m^k)^2 \cdot (1 + \cos(\omega_k j + 2\phi_m^k)).$$

Following this, based on (F.14), the simplified derivative of each entry takes the form

$$\frac{\partial \ell}{\partial \xi_m[j]} - \mathsf{Err}_{m,j}^{(1)} = - \sum_{(x,y) \in \mathcal{S}_j^p} \sigma(\langle e_x + e_y, \theta_m \rangle) + \frac{1}{p} \cdot \sum_{j=1}^{p} \sum_{(x,y) \in \mathcal{S}_j^p} \sigma(\langle e_x + e_y, \theta_m \rangle)$$

$$= -p \cdot \sum_{k=1}^{(p-1)/2} (\alpha_m^k)^2 \cdot \cos(\omega_k j + 2\phi_m^k), \qquad \forall j \in [p].$$

**Step 2: Deriving Gradient of $\theta_m$.** Next, we calculate the gradient of $\theta_m$, following a procedure analogous to the one in Step 1. To begin, we consider the expression:

$$\sum_{x \in \mathbb{Z}_p} \xi_m[m_p(x,j)] \cdot \sigma'(\langle e_x + e_j, \theta_m \rangle) = 2 \underbrace{\sum_{x \in \mathbb{Z}_p} \xi_m[m_p(x,j)] \cdot \theta_m[x]}_{\textbf{(iv)}} + 2\, \theta_m[j] \cdot \underbrace{\sum_{x \in \mathbb{Z}_p} \xi_m[x]}_{\textbf{(v)}}. \tag{F.19}$$

Term **(iv)** can be decomposed and simplified using the fourier expansions of $\xi_m$ and $\theta_m$ in (3.2). By carefully applying cosine product identities and rearranging the terms, we have

$$\textbf{(iv)} = \sum_{x \in \mathbb{Z}_p} \left( \beta_m^0 + \sum_{k=1}^{(p-1)/2} \beta_m^k \cdot \cos(\omega_k \cdot m_p(x,j) + \psi_m^k) \right) \cdot \left( \alpha_m^0 + \sum_{k=1}^{(p-1)/2} \alpha_m^k \cdot \cos(\omega_k x + \phi_m^k) \right)$$

$$= p \cdot \alpha_m^0 \cdot \beta_m^0 + \sum_{k=1}^{(p-1)/2} \alpha_m^k \beta_m^k \cdot \textbf{(iv.1)} + \alpha_m^0 \cdot \sum_{k=1}^{(p-1)/2} \beta_m^k \cdot \textbf{(iv.2)}$$

$$+ \sum_{1 \le k \ne \tau \le (p-1)/2} \alpha_m^k \beta_m^\tau \cdot \textbf{(iv.3)} + \beta_m^0 \cdot \sum_{k=1}^{(p-1)/2} \alpha_m^k \cdot \textbf{(iv.4)}.$$

where we denote each term as

$$\textbf{(iv.1)} = \sum_{x \in \mathbb{Z}_p} \cos(\omega_k \cdot m_p(x,j) + \psi_m^k) \cdot \cos(\omega_k x + \phi_m^k), \quad \textbf{(iv.2)} = \sum_{x \in \mathbb{Z}_p} \cos(\omega_k \cdot m_p(x,j) + \psi_m^k),$$

$$\textbf{(iv.3)} = \sum_{x \in \mathbb{Z}_p} \cos(\omega_\tau \cdot m_p(x,j) + \phi_m^\tau) \cdot \cos(\omega_k x + \psi_m^k), \quad \textbf{(iv.4)} = \sum_{x \in \mathbb{Z}_p} \cos(\omega_k \cdot m_p(x,j) + \phi_m^k).$$

Analogous to (F.17) and (F.18), using the trigonometric identities and periodicity of the module addition task, we have **(iv.2)** = **(iv.3)** = **(iv.4)** = $0$, and for the first term we can show that

$$\textbf{(iv.1)} = \sum_{x \in \mathbb{Z}_p} \cos(\omega_k \cdot m_p(x,j) + \psi_m^k) \cdot \cos(\omega_k x + \phi_m^k) = p \cdot \cos(\omega_k j + \psi_m^k - \phi_m^k).$$

By combining the arguments above, we can conclude that

$$\textbf{(iv)} = p \cdot \alpha_m^0 \cdot \beta_m^0 + p \sum_{k=1}^{(p-1)/2} \alpha_m^k \beta_m^k \cdot \cos(\omega_k j + \psi_m^k - \phi_m^k). \tag{F.20}$$

Besides, by substituting the fourier expansions of $\xi_m$ into **(v)**, it holds that

$$\textbf{(v)} = \theta_m[j] \cdot \sum_{x \in \mathbb{Z}_p} \left( \beta_m^0 + \sum_{k=1}^{(p-1)/2} \beta_m^k \cdot \cos(\omega_k x + \psi_m^k) \right)$$

$$= p \cdot \theta_m[j] \cdot \beta_m^0 = p \cdot \beta_m^0 \cdot \left( \alpha_m^0 + \sum_{k=1}^{(p-1)/2} \alpha_m^k \cdot \cos(\omega_k j + \phi_m^k) \right). \tag{F.21}$$

By combining (F.19), (F.20), (F.21) and substituting them back into (F.15), by simple calculation, we can show that constant frequencies are canceled and we have

$$\frac{\partial \ell}{\partial \theta_m[j]} - \mathsf{Err}_{m,j}^{(2)} = -2 \sum_{x \in \mathbb{Z}_p} \xi_m[m_p(x,j)] \cdot \sigma'(\langle e_x + e_j, \theta_m \rangle) + \frac{2}{p} \cdot \sum_{x \in \mathbb{Z}_p} \sum_{\tau=1}^{p} \xi_m[\tau] \cdot \sigma'(\langle e_x + e_j, \theta_m \rangle)$$

$$= -2p \cdot \sum_{k=1}^{(p-1)/2} \alpha_m^k \beta_m^k \cdot \cos(\omega_k j + \psi_m^k - \phi_m^k)$$

$$- 2p \cdot \beta_m^0 \cdot \sum_{k=1}^{(p-1)/2} \alpha_m^k \cdot \cos(\omega_k j + \phi_m^k), \qquad \forall j \in [p].$$

Recall that the gradient flow is defined as $\partial_t \Theta(t) = -\nabla \ell(\Theta(t))$. Following this, we have $\partial_t \theta_m(t) = -\nabla_{\theta_m} \ell$ and $\partial_t \xi_m(t) = -\nabla_{\xi_m} \ell$ for all $m \in [M]$. Then, by combining Step 1 and Step 2 and using the definition of gradient flow, we complete the proof. $\qquad \square$

### F.4 PROOF OF THEOREM 5.2: SINGLE-FREQUENCY PRESERVATION

**Theorem F.4** (Formal Statement of Theorem 5.2). *Let the model be initialized according to Assumption 5.1 with a scale $\kappa_{\mathsf{init}} > 0$. For a given threshold $C_{\mathsf{end}} > 0$, we define the initial stage as the time interval $(0, t_{\mathsf{init}}]$, where $t_{\mathsf{init}}$ is the first hit time:*

$$t_{\mathsf{init}} := \inf\{t \in \mathbb{R}^+ : \max_{m \in [M]} \|\theta_m(t)\|_\infty \vee \|\xi_m(t)\|_\infty \leq C_{\mathsf{end}}\}. \tag{F.22}$$

*Suppose the following conditions hold: (i) $\log M / M \lesssim c^{-1/2} \cdot (1 + o(1))$, $\kappa_{\mathsf{init}} = o(M^{-1/3})$ and $C_{\mathsf{end}} = \Theta(\kappa_{\mathsf{init}})$, and (ii) scale $\kappa_{\mathsf{init}}$ is sufficiently small such that the event $\mathcal{E}_{\mathsf{phase}} = \{\exists m \in [M] \text{ s.t. } \cos(2\phi_m^\star(t) - \psi_m^\star(t)) \geq 1 - c \cdot (M^{-1} \log M)^2, \forall t \in (0, t_{\mathsf{init}}]\}$ holds with probability greater than $1 - M^{-c}$ for some constant $c > 0$. Then, we have $\max_{k \neq k^\star} \inf_{t \in (0, t_{\mathsf{init}}]} \alpha_m^k(t) \vee \beta_m^k(t) = o(\kappa_{\mathsf{init}})$.*

In Theorem F.4, the initial time interval $(0, t_{\mathsf{init}})$ is defined by imposing that the parameters remain substantially small, upper bounded by $C_{\mathsf{end}}$ as stated in (F.22). $\mathcal{E}_{\mathsf{phase}}$ assumes during the initial stage, there exists at least one well-aligned neuron whose phase difference $2\phi_m^\star(t) - \psi_m^\star(t)$ has a uniformly lower-bounded cosine value. This should hold with high probability under the random initialization in Assumption 5.1, jointly resulting from the concentration (see Lemma F.6) and the

consistent decrease of phase difference for well-initialized neurons when $\kappa_{\text{init}} \mapsto 0$ (see Lemma F.9). Since the difference between the real dynamics for $\theta_m(t), \xi_m(t)$ and the central flow can be bounded by some error uniformly over $t \in (0, t_{\text{init}}]$, where the error is a monotone function with respect to $\kappa_{\text{init}}$, and the real dynamics for $\phi_m^\star(t), \psi_m^\star(t)$ is a continuous function of the real dynamics for $\theta_m(t), \xi_m(t)$, this claim holds.

*Proof of Theorem F.4.* Based on Lemma F.2 and (F.22), throughout the training, we can uniformly upper bound the approximation errors by

$$
\sup_{t \in (0, t_{\text{init}})} \max_{m,j} |\text{Err}_{m,j}^{(1)}(t)| \vee |\text{Err}_{m,j}^{(2)}(t)|
$$

$$
\leq 8p \cdot \sup_{t \in (0, t_{\text{init}})} \max_m \|\theta_m(t)\|_\infty \cdot \max\{\|\xi_m(t)\|_\infty, \|\theta_m(t)\|_\infty\} \cdot (\exp(8M \cdot \|\xi(t)\|_\infty \cdot \|\theta(t)\|_\infty^2) - 1)
$$

$$
\lesssim Mp \cdot C_{\text{end}}^5, \tag{F.23}
$$

where the last inequality uses $\exp(x) - 1 \lesssim x$ for $x \in [0, 1]$ and (F.22) implies $8M \cdot \|\xi(t)\|_\infty \cdot \|\theta(t)\|_\infty^2 \leq 1$ for all $t \in (0, t_{\text{init}})$ under the scaling that $MC_{\text{end}}^3 \asymp M\kappa_{\text{init}}^3 \ll 1$. In the following, we show that the evolution of non-feature frequencies is governed by the bounded error terms, and the feature coefficient can grow rapidly even when perturbed by noise.

**Step 1: Derive the Dynamics with Approximation Errors.** Consider a fixed neuron $m$. By applying the chain rule, we have $\partial_t g_m(t) = B_p^\top \partial_t \theta_m(t)$ and $\partial_t r_m(t) = B_p^\top \partial_t \xi_m(t)$ such that

$$
\partial_t g_m[j](t) = \langle b_j, \partial_t \theta_m(t) \rangle, \qquad \partial_t r_m[j](t) = \langle b_j, \partial_t \xi_m(t) \rangle, \qquad \forall j \in [p].
$$

Hence, the time derivatives of constant frequency, based on Lemma F.3, satisfy that

$$
\partial_t r_m[1](t) = -\langle \text{Err}_m^{(1)}(t), b_1 \rangle, \qquad \partial_t g_m[1](t) = -\langle \text{Err}_m^{(2)}(t), b_1 \rangle, \tag{F.24}
$$

where the the RHS of (F.24) can be controlled by

$$
|\langle \text{Err}_m^{(1)}(t), b_1 \rangle| \leq \|\text{Err}_m^{(1)}(t)\|_2 \cdot \|b_1\|_2 \leq \sqrt{p} \cdot \|\text{Err}_m^{(1)}(t)\|_\infty, \quad |\langle \text{Err}_m^{(2)}(t), b_1 \rangle| \leq \sqrt{p} \cdot \|\text{Err}_m^{(2)}(t)\|_\infty.
$$

Based on Lemma F.3 and the orthogonality of the Fourier basis, by simple calculation, it holds that

$$
\partial_t r_m[2k](t) = p \cdot \sum_{j=1}^p \sqrt{\frac{2}{p}} \cdot \cos(\omega_k j) \cdot \sum_{k=1}^{(p-1)/2} \alpha_m^k(t)^2 \cdot \cos(\omega_k j + 2\phi_m^k(t)) - \sum_{j=1}^p b_{2k}[j] \cdot \text{Err}_{m,j}^{(1)}(t)
$$

$$
= \sqrt{2p} \cdot \alpha_m^k(t)^2 \cdot \sum_{j=1}^p \cdot \cos(\omega_k j) \cdot \cos(\omega_k j + 2\phi_m^k(t)) - \langle \text{Err}_m^{(1)}(t), b_{2k} \rangle
$$

$$
= \frac{p^{3/2}}{\sqrt{2}} \cdot \alpha_m^k(t)^2 \cdot \cos(2\phi_m^\star(t)) - \langle \text{Err}_m^{(1)}(t), b_{2k} \rangle,
$$

and similarly, we have

$$
\partial_t r_m[2k+1](t) = -\frac{p^{3/2}}{\sqrt{2}} \cdot \alpha_m^k(t)^2 \cdot \sin(2\phi_m^k(t)) - \langle \text{Err}_m^{(1)}(t), b_{2k+1} \rangle.
$$

Following this, by applying the chain rule, we have

$$
\partial_t \beta_m^k(t) = \sqrt{\frac{2}{p}} \cdot \partial_t \sqrt{r_m[2k](t)^2 + r_m[2k+1](t)^2}
$$

$$
= \frac{2}{p} \cdot \left\{ \frac{r_m[2k](t)}{\beta_m^k(t)} \cdot \partial_t r_m[2k](t) + \frac{r_m[2k+1](t)}{\beta_m^k(t)} \cdot \partial_t r_m[2k+1](t) \right\}
$$

$$
= \frac{2}{p} \cdot \frac{p^{3/2}}{\sqrt{2}} \cdot \sqrt{\frac{p}{2}} \cdot \alpha_m^k(t)^2 \cdot \left\{ \cos(\psi_m^k(t)) \cdot \cos(2\phi_m^k(t)) + \sin(\psi_m^k(t)) \cdot \sin(2\phi_m^k(t)) \right\} + \widetilde{\text{Err}}_m^{(1)}(t)
$$

$$
= p \cdot \alpha_m^k(t)^2 \cdot \cos(2\phi_m^k(t) - \psi_m^k(t)) + \widetilde{\text{Err}}_m^{(1)}(t), \tag{F.25}
$$

where we define the approximation-induced error term as:

$$
\widetilde{\text{Err}}_m^{(1)}(t) := -\frac{2}{p} \cdot \left\{ \frac{r_m[2k](t)}{\beta_m^k(t)} \cdot \langle \text{Err}_m^{(1)}(t), b_{2k} \rangle - \frac{r_m[2k+1](t)}{\beta_m^k(t)} \cdot \langle \text{Err}_m^{(1)}(t), b_{2k+1} \rangle \right\}.
$$

Here, notice that the error terms can be upper bounded by

$$|\widetilde{\mathsf{Err}}_m^{(1)}(t)| \leq \sqrt{\frac{2}{p}} \cdot \sqrt{\langle \mathsf{Err}_m^{(1)}(t), b_{2k}\rangle^2 + \langle \mathsf{Err}_m^{(1)}(t), b_{2k+1}\rangle^2}$$

$$\leq \sqrt{\frac{2}{p}} \cdot \|\mathsf{Err}_m^{(1)}(t)\|_2 \cdot \sqrt{\|b_{2k}\|_2^2 + \|b_{2k+1}\|_2^2} \leq 2\|\mathsf{Err}_m^{(1)}(t)\|_\infty,$$

where the first inequality uses the Cauchy-Schwarz inequality and the fact that $r_m[2k](t)^2 + r_m[2k+1](t)^2 = p/2 \cdot \beta_m^k(t)^2$ by definition. Moreover, following a similar argument above, we have

$$\partial_t g_m[2k](t) = \sqrt{2}p^{3/2} \cdot \alpha_m^k(t) \cdot \beta_m^k(t) \cdot \cos(\psi_m^k(t) - \phi_m^k(t))$$

$$+ \sqrt{2}p^{3/2} \cdot \beta_m^0(t) \cdot \alpha_m^k(t) \cdot \cos(\phi_m^k(t)) + \langle \mathsf{Err}_m^{(2)}(t), b_{2k}\rangle,$$

and also

$$\partial_t g_m[2k+1](t) = -\sqrt{2}p^{3/2} \cdot \alpha_m^k(t) \cdot \beta_m^k(t) \cdot \sin(\psi_m^k(t) - \phi_m^k(t))$$

$$- \sqrt{2}p^{3/2} \cdot \beta_m^0(t) \cdot \alpha_m^k(t) \cdot \sin(\phi_m^k(t)) + \langle \mathsf{Err}_m^{(2)}(t), b_{2k+1}\rangle.$$

Thus, by applying the chain rule, we can reach that

$$\partial_t \alpha_m^k(t) = \sqrt{\frac{2}{p}} \cdot \partial_t \sqrt{g_m[2k](t)^2 + g_m[2k+1](t)^2}$$

$$= 2p \cdot \alpha_m^k(t) \cdot \beta_m^k(t) \cdot \cos\left(2\phi_m^k(t) - \psi_m^k(t)\right) + 2p \cdot \beta_m^0(t) \cdot \alpha_m^k(t) + \widetilde{\mathsf{Err}}_m^{(2)}(t), \quad \text{(F.26)}$$

where the approximation error satisfies that

$$|\widetilde{\mathsf{Err}}_m^{(2)}(t)| = \frac{2}{p} \cdot \left| \frac{g_m[2k](t)}{\alpha_m^k(t)} \cdot \langle \mathsf{Err}_m^{(2)}(t), b_{2k}\rangle - \frac{g_m[2k+1](t)}{\alpha_m^k(t)} \cdot \langle \mathsf{Err}_m^{(2)}(t), b_{2k+1}\rangle \right| \leq 2\|\mathsf{Err}_m^{(2)}(t)\|_\infty.$$

**Step 2.1: Bound the Growth of Non-feature Frequency.** By combining (F.24), (F.25) and (F.26), since $\cos\left(2\phi_m^k(t) - \psi_m^k(t)\right)$, we can upper bound the growth of non-feature frequencies as

$$\partial_t \alpha_m^k(t) \leq 2p \cdot \alpha_m^k(t) \cdot \beta_m^k(t) + 2p \cdot \beta_m^0(t) \cdot \alpha_m^k(t) + \widetilde{\mathsf{Err}}_m^{(2)}(t), \quad \text{(F.27a)}$$

$$\partial_t \beta_m^k(t) \leq p \cdot \alpha_m^k(t)^2 + \widetilde{\mathsf{Err}}_m^{(1)}(t), \quad \text{(F.27b)}$$

$$\partial_t r_m[1](t) \leq \sqrt{p} \cdot \|\mathsf{Err}_m^{(1)}(t)\|_\infty, \quad \partial_t g_m[1](t) \leq \sqrt{p} \cdot \|\mathsf{Err}_m^{(2)}(t)\|_\infty, \quad \text{(F.27c)}$$

$$|\widetilde{\mathsf{Err}}_m^{(i)}(t)| \lesssim \|\mathsf{Err}_m^{(i)}(t)\|_\infty, \quad \forall i \in \{0, 1\}. \quad \text{(F.27d)}$$

for all $k \neq k^\star$ and $m \in [M]$. For the growth of constant coefficients, (F.27c) indicates that

$$|\alpha_m^0(t)| \vee |\beta_m^0(t)| = 1/\sqrt{p} \cdot |g_m[1](t)| \vee |r_m[1](t)|$$

$$\leq \max_{t \in (0, t_{\mathsf{init}}]} \|\mathsf{Err}_m^{(1)}(t)\|_\infty \vee \|\mathsf{Err}_m^{(2)}(t)\|_\infty \cdot t \lesssim Mp \cdot C_{\mathsf{end}}^5 \cdot t, \quad \text{(F.28)}$$

where the inequality results from (F.23). Following this, by combining (F.27a), (F.27b), (F.27c) and (F.27d), it holds that

$$\partial_t \{\alpha_m^k(t)/\sqrt{2} + \beta_m^k(t)\} \leq p \cdot \alpha_m^k(t) \cdot \{\alpha_m^k(t) + \sqrt{2}\beta_m^k(t)\}$$

$$+ \sqrt{2}p \cdot \alpha_m^k(t) \cdot \beta_m^0(t) + \widetilde{\mathsf{Err}}_m^{(1)}(t) + \widetilde{\mathsf{Err}}_m^{(2)}(t)/\sqrt{2}$$

$$\leq \sqrt{2}p \cdot C_{\mathsf{end}} \cdot \{\alpha_m^k(t) + \sqrt{2}\beta_m^k(t)\}$$

$$+ 2p \cdot C_{\mathsf{end}} \cdot |\beta_m^0(t)| + \widetilde{\mathsf{Err}}_m^{(1)}(t) + \widetilde{\mathsf{Err}}_m^{(2)}(t)/\sqrt{2},$$

where the last inequality uses (F.22) and $\|\theta_m(t)\|_2^2 = p \cdot \alpha_m^0(t)^2 + \frac{p}{2} \cdot \sum_{k=1}^{(p-1)/2} \alpha_m^k(t)^2$ such that

$$\alpha_m^k(t) \leq \sqrt{2/p} \cdot \|\theta_m(t)\|_2 \leq \sqrt{2} \cdot \|\theta_m(t)\|_\infty \leq \sqrt{2} \cdot C_{\mathsf{end}}, \quad \forall t \in (0, t_{\mathsf{init}}), \quad \text{(F.29)}$$

for all frequency $k$ and similarly we have $\beta_m^k(t) \leq \sqrt{2} \cdot C_{\mathsf{end}}$. For $k \neq k^\star$, Lemma F.5 shows that

$$\alpha_m^k(t)/\sqrt{2} + \beta_m^k(t) \leq \{\alpha_m^k(0)/\sqrt{2} + \beta_m^k(0)\} \cdot \exp(\sqrt{2}p \cdot C_{\mathsf{end}} \cdot t)$$

$$+ \underbrace{2p \cdot C_{\mathsf{end}} \cdot \int_0^t |\beta_m^0(s)| \cdot \exp(\sqrt{2}p \cdot C_{\mathsf{end}} \cdot (t-s))\mathrm{d}s}_{\text{(i)}}$$

$$+ \underbrace{\int_0^t \{\widetilde{\mathsf{Err}}_m^{(1)}(s) + \widetilde{\mathsf{Err}}_m^{(2)}(s)/\sqrt{2}\} \cdot \exp(\sqrt{2}p \cdot C_{\mathsf{end}} \cdot (t-s))\mathrm{d}s}_{\text{(ii)}}, \quad \text{(F.30)}$$

where the first term can be eliminated due to the zero initialization for non-feature frequencies as specified in Assumption 5.1. To upper bound (F.30), based on (F.28) and (F.29), we can show that

$$\textbf{(i)} \lesssim Mp^2 \cdot C_{\mathsf{end}}^6 \cdot \int_0^t s \cdot \exp(\sqrt{2}p \cdot C_{\mathsf{end}} \cdot (t-s)) \mathrm{d}s$$

$$\lesssim C_{\mathsf{end}}^4 \cdot \{\exp(\sqrt{2}p \cdot C_{\mathsf{end}} \cdot t) - \sqrt{2}p \cdot C_{\mathsf{end}} \cdot t - 1\} \lesssim Mp^2 \cdot C_{\mathsf{end}}^6 \cdot t^2, \qquad \text{(F.31)}$$

for time $t \le (\sqrt{2}p \cdot C_{\mathsf{end}})^{-1} \wedge t_{\mathsf{init}}$ using $\exp(x) - x - 1 \le x^2$ for $x \in (0,1)$. Similarly, it holds that

$$\textbf{(ii)} \le \int_0^t \{2\|\mathsf{Err}_m^{(1)}(s)\|_\infty + \sqrt{2}\|\mathsf{Err}_m^{(1)}(s)\|_\infty\} \cdot \exp(\sqrt{2}p \cdot C_{\mathsf{end}} \cdot (t-s)) \mathrm{d}s$$

$$\le 4 \sup_{t \in (0, t_{\mathsf{init}})} \max_m \|\mathsf{Err}_m^{(1)}(t)\|_\infty \vee \|\mathsf{Err}_m^{(2)}(t)\|_\infty \cdot \int_0^t \exp(\sqrt{2}p \cdot C_{\mathsf{end}} \cdot (t-s)) \mathrm{d}s$$

$$\lesssim Mp \cdot C_{\mathsf{end}}^5 \cdot \int_0^t \exp(\sqrt{2}p \cdot C_{\mathsf{end}} \cdot (t-s)) \mathrm{d}s \lesssim Mp \cdot C_{\mathsf{end}}^5 \cdot t, \qquad \text{(F.32)}$$

where the first inequality follows (F.27d) and the last inequality results from $\exp(x) - 1 \le 2x$ for $x \in (0,1)$. By combining (F.30), (F.31) and (F.32), we can conclude that

$$\alpha_m^k(t) \vee \beta_m^k(t) \lesssim Mp \cdot C_{\mathsf{end}}^5 \cdot t \cdot \max\{p \cdot C_{\mathsf{end}} \cdot t, 1\} \le Mp \cdot C_{\mathsf{end}}^5 \cdot t, \qquad \text{(F.33)}$$

for all non-feature frequencies $k \ne k^\star$ if we consider time $t \le (\sqrt{2}p \cdot C_{\mathsf{end}})^{-1} \wedge t_{\mathsf{init}}$. For the remainder of this analysis, we will adhere to this interval, and we will later show $t_{\mathsf{init}} \lesssim (p \cdot C_{\mathsf{end}})^{-1}$.

**Step 2.2: Bound the Time of Initial Stage.** Based on (F.25) and (F.26), we first show that during the initial stage, the change in the quantity $\alpha_m^\star(t)^2 - 2\beta_m^\star(t)^2$ remains small. Note that

$$\partial_t\{\alpha_m^\star(t)^2 - 2\beta_m^\star(t)^2\} = 2\alpha_m^\star(t) \cdot \partial_t \alpha_m^\star(t) - 4\beta_m^\star(t) \cdot \partial_t \beta_m^\star(t)$$

$$= 4p \cdot \alpha_m^\star(t)^2 \cdot \beta_m^\star(t) \cdot \cos\left(2\phi_m^\star(t) - \psi_m^\star(t)\right)$$

$$+ 4p \cdot \beta_m^0(t) \cdot \alpha_m^\star(t)^2 + 2\alpha_m^\star(t) \cdot \widetilde{\mathsf{Err}}_m^{(2)}(t)$$

$$- 4p \cdot \alpha_m^\star(t)^2 \cdot \beta_m^\star(t) \cdot \cos\left(2\phi_m^\star(t) - \psi_m^\star(t)\right) - 4\beta_m^\star(t) \cdot \widetilde{\mathsf{Err}}_m^{(1)}(t)$$

$$= 4p \cdot \beta_m^0(t) \cdot \alpha_m^\star(t)^2 + 2\alpha_m^\star(t) \cdot \widetilde{\mathsf{Err}}_m^{(2)}(t) - 4\beta_m^\star(t) \cdot \widetilde{\mathsf{Err}}_m^{(1)}(t).$$

Following this, by integrating on both sides, we can show that

$$\alpha_m^\star(t)^2 - 2\beta_m^\star(t)^2$$

$$\ge \alpha_m^\star(0)^2 - 2\beta_m^\star(0)^2 - \int_0^t |\partial_t\{\alpha_m^\star(s)^2 - 2\beta_m^\star(s)^2\}| \mathrm{d}s$$

$$\ge -\kappa_{\mathsf{init}}^2 - 8p \cdot C_{\mathsf{end}}^2 \cdot \int_0^t |\beta_m^0(s)| \mathrm{d}s - 6\sqrt{2} \cdot C_{\mathsf{end}} \cdot \sup_{t \in (0, t_{\mathsf{init}})} |\widetilde{\mathsf{Err}}_m^{(1)}(t)| \vee |\widetilde{\mathsf{Err}}_m^{(2)}(t)| \cdot t$$

$$\ge -\kappa_{\mathsf{init}}^2 - O(Mp \cdot C_{\mathsf{end}}^6) \cdot t, \qquad \text{(F.34)}$$

where the second inequality uses (F.29). Recall that we choose a sufficiently small $\kappa_{\mathsf{init}}$ such that $\mathcal{E}_{\mathsf{phase}}$ holds. Thus, there exists a neuron $m$ such that $\inf_{t \in (0, t_{\mathsf{init}})} \cos(2\phi_m^\star(t) - \psi_m^\star(t)) \ge C_D$. Leveraging this result along with (F.25) and (F.34), it follows that:

$$\partial_t \beta_m^\star(t) \ge p \cdot C_D \cdot \alpha_m^\star(t)^2 + \widetilde{\mathsf{Err}}_m^{(1)}(t)$$

$$= 2p \cdot C_D \cdot \beta_m^\star(t)^2 + p \cdot C_D \cdot \{\alpha_m^\star(t)^2 - 2\beta_m^\star(t)^2\} + \widetilde{\mathsf{Err}}_m^{(1)}(t)$$

$$\ge 2p \cdot C_D \cdot \beta_m^\star(t)^2 - p \cdot C_D \cdot \kappa_{\mathsf{init}}^2 - O(Mp^2 \cdot C_{\mathsf{end}}^6) \cdot C_D \cdot t - O(Mp \cdot C_{\mathsf{end}}^5)$$

$$\ge 2p \cdot C_D \cdot \beta_m^\star(t)^2 - p \cdot \{\kappa_{\mathsf{init}}^2 + O(M \cdot C_{\mathsf{end}}^5)\}$$

$$\ge 2p \cdot C_D \cdot \beta_m^\star(t)^2 - p \cdot (1 + o(1)) \cdot \kappa_{\mathsf{init}}^2, \qquad \text{(F.35)}$$

where the second inequality results from (F.27d), the third is guaranteed by the time interval constraint $t \le (\sqrt{2}p \cdot C_{\mathsf{end}})^{-1} \wedge t_{\mathsf{init}}$, and the last one uses $M\kappa_{\mathsf{init}}^3 = o(1)$ and $C_{\mathsf{end}} = \Theta(\kappa_{\mathsf{init}})$.

Given the Riccati ODE in (F.35) and the initialization $\beta_m^\star(0) = \kappa_{\text{init}}$, $\beta_m^\star(t)$ is monotone increasing as long as $2C_D \geq 1 + o(1)$, which can be guaranteed by choosing a sufficiently large $M$ such that $\log M/M \lesssim c^{-1/2} \cdot (1 + o(1))$. Following this, we can further show that

$$\partial_t \beta_m^\star(t) \geq 2p\kappa_{\text{init}} \cdot C_D \cdot \beta_m^\star(t) - p \cdot (1 + o(1)) \cdot \kappa_{\text{init}}^2, \qquad \forall t \leq (\sqrt{2}p \cdot C_{\text{end}})^{-1} \wedge t_{\text{init}}. \quad \text{(F.36)}$$

By combining (F.36) and Lemma F.5, we can get

$$\beta_m^\star(t) \geq \kappa_{\text{init}} \cdot \exp(2p\kappa_{\text{init}} \cdot C_D \cdot t) - (1 + o(1)) \cdot \kappa_{\text{init}}/(2C_D) \cdot \{\exp(2p\kappa_{\text{init}} \cdot C_D \cdot t) - 1\}.$$

By definition $\beta_m^\star(t_{\text{init}}) \lesssim C_{\text{end}} \asymp \kappa_{\text{init}}$. Thus, we can upper bound the hitting time $t_{\text{init}}$ by

$$t_{\text{init}} \lesssim \frac{1}{2p\kappa_{\text{init}} \cdot C_D} \cdot \log\left(\frac{C_{\text{end}}/\kappa_{\text{init}} - (1 + o(1))/(2C_D)}{1 - (1 + o(1))/(2C_D)}\right) \lesssim (p\kappa_{\text{init}})^{-1}. \quad \text{(F.37)}$$

**Step 3: Conclude the Proof.** Based on (F.28), (F.33) and (F.37), it holds that

$$\max_{k \neq k^\star} \inf_{t \in (0, t_{\text{init}}]} \alpha_m^k(t) \vee \beta_m^k(t) \lesssim Mp \cdot C_{\text{end}}^5 \cdot t_{\text{init}} \leq o(\kappa_{\text{init}}),$$

which completes the proof.

$\square$

### F.4.1 PROOF OF AUXILIARY LEMMA F.5

**Lemma F.5.** *Let $\iota \neq 0$ denote a non-zero constant and $\zeta : [0, \infty) \mapsto \mathbb{R}^n$ denote a continuous function. For any initial $x(0) \in \mathbb{R}^n$, the unique solution of $\partial_t x(t) = \iota x(t) + \zeta(t)$ is given by*

$$x(t) = x(0) \cdot \exp(\iota t) + \int_0^t \zeta(s) \cdot \exp(\iota(t - s))\mathrm{d}s.$$

*In particular, if $\zeta(t) \equiv \zeta \in \mathbb{R}$ is constant, then $x(t) = x(0) \cdot \exp(\iota t) + \zeta/\iota \cdot (\exp(\iota t) - 1)$.*

*Proof of Lemma F.5.* Note that, by chain rule, we have

$$\partial_t \{x_t \cdot \exp(-\iota t)\} = -\iota x(t) \cdot \exp(-\iota t) + \partial_t x(t) \cdot \exp(-\iota t) = \zeta(t) \cdot \exp(-\iota t).$$

By integrating both sides from $0$ to $t$, we can obtain the desired result. $\square$

**Lemma F.6.** *Under the initialization in Assumption 5.1, with probability greater that $1 - M^{-c}$, it holds that $\max_{m \in [M]} \cos(\mathcal{D}_m^\star) > 1 - c^2\pi^2 \cdot M^{-2}(\log M)^2$, where $c > 0$ is a constant.*

*Proof of Lemma F.6.* Throughout the proof, we drop the initial time $(0)$ for simplicity. Recall that, as specified in Assumption 5.1, the parameters are initialized as below

$$\theta_m \sim \kappa_{\text{init}} \cdot \sqrt{p/2} \cdot (\varrho_1[1] \cdot b_{2k^\star} + \varrho_1[2] \cdot b_{2k^\star+1}), \quad \xi_m \sim \kappa_{\text{init}} \cdot \sqrt{p/2} \cdot (\varrho_2[1] \cdot b_{2k^\star} + \varrho_2[2] \cdot b_{2k^\star+1}).$$

By definition, we have $\cos(\phi_m^\star) = \varrho_1[1]$ and $\sin(\phi_m^\star) = -\varrho_1[2]$. Thus, it holds that

$$(\cos(\phi_m^\star), \sin(\phi_m^\star)) = (\varrho_1[1], -\varrho_1[2]) \overset{d}{=} (\varrho_1[1], \varrho_1[2]),$$

following the symmetry of the uniform distribution on the unit circle. Hence, $\phi_m^\star(0) \sim \text{Unif}(-\pi, \pi)$. Similarly, we have $\psi_m^\star \sim \text{Unif}(-\pi, \pi)$ such that $\mathcal{D}_m^\star = 2\phi_m^\star - \psi_m^\star \mod 2\pi \sim \text{Unif}(0, 2\pi)$. Following this, the tail probability takes the form:

$$\mathbb{P}\left(\max_{m \in [M]} \cos(\mathcal{D}_m^\star) > 1 - c^2\pi^2 \cdot M^{-2}(\log M)^2\right)$$

$$= 1 - \mathbb{P}\left(\forall m \in [M], \cos(\mathcal{D}_m^\star) \leq 1 - c^2\pi^2 \cdot M^{-2}(\log M)^2\right)$$

$$= 1 - \left(1 - \arccos\left(1 - c^2\pi^2 \cdot M^{-2}(\log M)^2\right)/\pi\right)^M. \quad \text{(F.38)}$$

Suppose $M > c\pi \log M$ such that $c\pi \cdot M^{-1} \log M \in (0, 1)$, then we have

$$\arccos\left(1 - c^2\pi^2 \cdot M^{-2}(\log M)^2)\right) \geq \arccos\left(\cos(c\pi \cdot M^{-1}\log M)\right) = c\pi \cdot M^{-1}\log M, \quad \text{(F.39)}$$

where the inequality follows from $\cos(x) \geq 1 - x^2$ for all $x \in \mathbb{R}$ and fact that $\arccos(\cdot)$ is monotonely decreasing on $[-1, 1]$. By combining (F.38) and (F.39), we obtain

$$\mathbb{P}\left(\max_{m \in [M]} \cos(\mathcal{D}_m^\star) > 1 - c^2\pi^2 \cdot M^{-2}(\log M)^2\right)$$

$$\geq 1 - \left(1 - c \cdot M^{-1}\log M\right)^M \geq 1 - \exp(-c\log M) = 1 - M^{-c}.$$

Here, we use $(1 - x)^M \leq \exp(-xM)$ for all $x \in [0, 1]$ and then complete the proof. $\square$

F.5 PROOF OF THEOREM 5.3: PHASE ALIGNMENT

In this section, due to the inherent difficulty of tracking a multi-particle dynamical system with error terms—even when the approximation errors are provably small—we focus on the central flow dynamics presented in Lemma F.3, directly omitting the error terms caused by unpredictable drift. In summary, the resulting dynamical system can be described by the following ODEs:

$$\partial_t \theta_m[j](t) = -2p \cdot \sum_{k=1}^{(p-1)/2} \alpha_m^k(t) \cdot \beta_m^k(t) \cdot \cos(\omega_k j + \psi_m^k(t) - \phi_m^k(t))$$

$$- 2p \cdot \beta_m[1](t) \cdot \sum_{k=1}^{(p-1)/2} \alpha_m^k \cdot \cos(\omega_k j + \phi_m^k(t)), \tag{F.40a}$$

$$\partial_t \xi_m[j](t) = p \cdot \sum_{k=1}^{(p-1)/2} \alpha_m^k(t)^2 \cdot \cos(\omega_k j + 2\phi_m^k(t)), \tag{F.40b}$$

for a fixed neuron $m$ and all $j \in [p]$. We formalize the phase alignment in the following theorem.

**Theorem F.7** (Formal Statement of Theorem 5.3). *Consider the main flow dynamics defined in (F.40a) and (F.40b), under the initialization in Assumption 5.1. Let $\delta = o(1)$ be a sufficiently small tolerance. For any $\mathcal{D}_m^\star(0) \in (0, 2\pi]$, define the convergence time $t_\delta = \inf\{t \in \mathbb{R}^+ : |\mathcal{D}_m^\star(t)| \leq \delta\}$. Then, $t_\delta$ satisfies*

$$t_\delta \asymp (p\kappa_{\mathsf{init}})^{-1} \cdot \left\{ 1 - (\sin(\mathcal{D}_m^\star(0))/\delta)^{-1/3} + \max\{\pi/2 - |\mathcal{D}_m^\star(0) - \pi|, 0\} \right),$$

*Furthermore, the magnitude at this time is given by $\beta_m^\star(t_\delta) \asymp \kappa_{\mathsf{init}} \cdot \{\sin(\mathcal{D}_m^\star(0))/\delta\}^{1/3}$. Moreover, in the mean-field regime $m \to \infty$, let $\rho_t = \mathrm{Law}\big(\phi_m^\star(t), \psi_m^\star(t)\big)$ for all $t \in \mathbb{R}^+$ and let $\lambda$ denote the uniform law on $(-\pi, \pi]$. Then, $\rho_0 = \lambda \otimes \lambda$ and $\rho_\infty = T_\# \lambda$, where $T : \varphi \mapsto (\varphi, 2\varphi) \mod 2\pi$.*

Before presenting the proof of Theorem F.7, we first introduce several key intermediate results that help elucidate the dynamics. We begin with a lemma that characterizes the simplified dynamics of the system, leveraging the Fourier domain and the single-frequency initialization.

**Lemma F.8** (Main Flow under Fourier Domain). *Under the initialization in Assumption 5.1, let $k^\star$ denote the initial frequency of each neuron, and we use the superscript $\star$ for notational simplicity. We define $\mathcal{D}_m^\star(t) = 2\phi_m^\star(t) - \psi_m^\star(t) \mod 2\pi$, then the main flow can be equivalently described as*

$$\partial_t \alpha_m^\star(t) = 2p \cdot \alpha_m^\star(t) \cdot \beta_m^\star(t) \cdot \cos(\mathcal{D}_m^\star(t)), \quad \partial_t \beta_m^\star(t) = p \cdot \alpha_m^\star(t)^2 \cdot \cos(\mathcal{D}_m^\star(t)),$$

$$\partial_t \exp(i\mathcal{D}_m^\star(t)) = p \cdot \left( 4\beta_m^\star(t) + \frac{\alpha_m^\star(t)^2}{\beta_m^\star(t)} \right) \cdot \sin\big(\mathcal{D}_m^\star(t)\big) \cdot \exp\big(i\{\mathcal{D}_m^\star(t) - \pi/2\}\big). \tag{F.41}$$

This lemma allows us to largely simplify the analysis, reducing it from tracking a $2p$-dimensional system to a three-particle dynamical system of $\alpha_m^\star(t)$, $\beta_m^\star(t)$ and $\mathcal{D}_m^\star(t)$. Building on this, the next two lemmas further show that the dynamics is indeed one-dimensional, and the trajectory exhibits a symmetry property that aids in understanding the evolutions under different initializations.

**Lemma F.9.** *Consider the ODE in (F.41), the following quantities remain constant:*

$$\alpha_m^\star(t)^2 - 2\beta_m^\star(t)^2 = C_{\mathsf{diff}}, \qquad \sin(\mathcal{D}_m^\star(t)) \cdot \beta_m^\star(t) \cdot \alpha_m^\star(t)^2 = C_{\mathsf{prod}}, \qquad \forall t \in \mathbb{R}^+.$$

*Building upon this, we can further simplify the dynamics of $\mathcal{D}_m^\star(t)$) in as*

$$\partial_t \mathcal{D}_m^\star(t) = -p \cdot \big(4\beta_m^\star(t) + \alpha_m^\star(t)^2/\beta_m^\star(t)\big) \cdot \sin\big(\mathcal{D}_m^\star(t)\big), \tag{F.42}$$

*due to its well-regularized behavior ensured by the constant relationship.*

We highlight that (F.42) is not a direct corollary from (F.41) due to the potential jump from $0$ to $2\pi$ in the discontinuous definition of $\mod 2\pi$. However, thanks to the constant relationship revealed in Lemma F.9, we can show that $\mathcal{D}_m^\star(t)$ is "well-behaved" by staying in the half-space where it is initialized, and consistently approaching zero throughout the gradient flow.

**Lemma F.10.** *Consider the ODE given in (F.41) with initial condition $\mathcal{D}_m^\star(0) \in (\pi/2, \pi)$. Let $t_{\pi/2}$ denote the hit time that $\mathcal{D}_m^\star(t_{\pi/2}) = \pi/2$, then for any $\Delta t \in (0, t_{\pi/2})$, we have*

$$\beta_m^\star(t_{\pi/2} - \Delta t) = \beta_m^\star(t_{\pi/2} + \Delta t), \qquad \mathcal{D}_m^\star(t_{\pi/2} - \Delta t) + \mathcal{D}_m^\star(t_{\pi/2} + \Delta t) = \pi.$$

*Proof of Lemma F.8, F.9 and F.10.* Please refer to §F.5.1 for a detailed proof. □

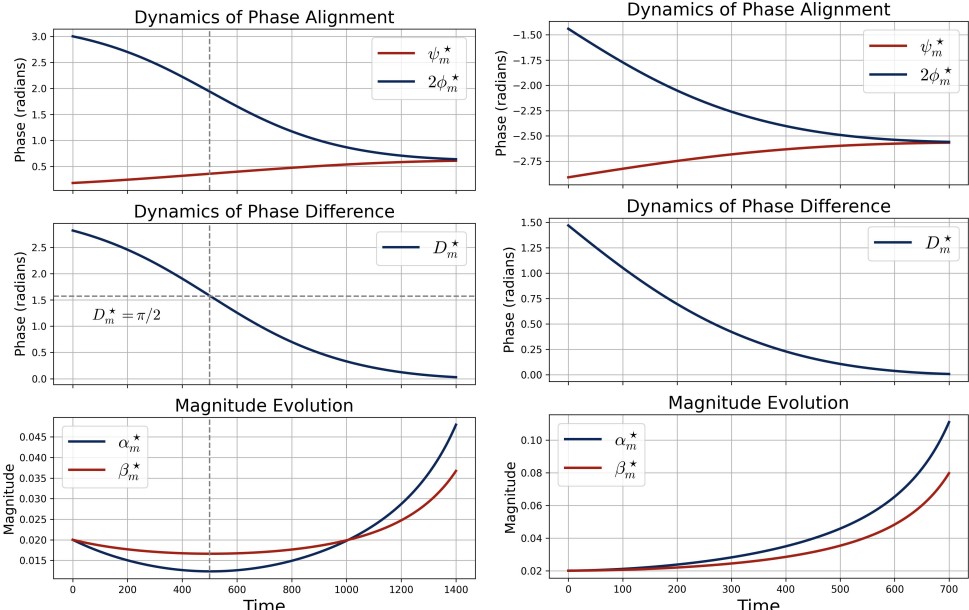

(a) Simplified Dynamics with $\mathcal{D}_m^\star(0) \in (\pi/2, \pi)$. (b) Simplified Dynamics with $\mathcal{D}_m^\star(0) \in (0, \pi/2)$.

Figure 14: Training dynamics of a specific decoupled neuron characterized by (F.48a) and (F.48b) with identical initial scales $\alpha_m^\star(0) = \beta_m^\star(0)$ and different phase difference $\mathcal{D}_m^\star(0)$. Figure (a) plots the dynamics of phases, phase difference, and the magnitudes with $\mathcal{D}_m^\star(0) \in (\pi/2, \pi)$, whose behavior is detailedly characterized in Theorem F.7. The difference decreases monotonically to 0, while the magnitudes first decay slightly when $\mathcal{D}_m^\star(t) \in (\pi/2, \pi)$ and then increase rapidly when $\mathcal{D}_m^\star(t)$ falls below $\pi/2$. Figure (b) plots the dynamics under $\mathcal{D}_m^\star(0) \in (0, \pi/2)$ where $\mathcal{D}_m^\star$ is initialized closer to the convergence point, resulting in a shorter convergence time compared to the case in Figure (a). Moreover, the simplified dynamics shown in Figure (b) align well with the full dynamics in Figure 5a with the same initialization, indicating the effectiveness of the approximation.

Now we are ready to present the proof of Theorem F.7.

*Proof of Theorem F.7.* Without loss of generality, we focus on the case where $\mathcal{D}_m^\star(0) \in (0, \pi)$. The case $\mathcal{D}_m^\star(0) \in (-\pi, 0)$ can be extended identically owing to the symmetry of dynamics in (F.41) as established in Lemmas F.8 and F.9. Specifically, the trajectories of $\alpha_m^\star(t)$ and $\beta_m^\star(t)$ are invariant under a sign flip of $\mathcal{D}_m^\star(t)$ such that the entire dynamics evolves symmetrically, with $\mathcal{D}_m^\star(t)$ mirrored from $(0, \pi)$ to $(-\pi, 0)$ at each time $t$.

**Roadmap.** In the following, we establish the convergence time by further dividing into two cases—$\mathcal{D}_m^\star(0) \in (0, \pi/2)$ and $\mathcal{D}_m^\star(0) \in (\pi/2, \pi)$. Notably, thanks to the symmetry established in Lemma F.10, we only need to characterize two time intervals (i) the travelling time from $\mathcal{D}_m^\star(0)$ to $\pi/2$ for any $\mathcal{D}_m^\star(0) \in (\pi/2, \pi)$, denoted by $\Delta t_{\pi/2}^{\rightarrow}$, both initialized at $\beta_m^\star(0) = \kappa_{\text{init}}$, (ii) the convergence time from $\mathcal{D}_m^\star(0)$ to 0 for an arbitrary initial phase $\mathcal{D}_m^\star(0) \in (0, \pi/2)$, denoted by $\Delta t_\delta^{\rightarrow}$ This is because,

- For $\mathcal{D}_m^\star(0) \in (0, \pi/2)$, the convergence time can be captured by $\Delta t_\delta^{\rightarrow}$
- For $\mathcal{D}_m^\star(0) \in (\pi/2, \pi)$, the time is given by $2\Delta t_{\pi/2}^{\rightarrow} + \Delta t_\delta^{\rightarrow}$, where with slight abuse of notation we let $\Delta t_\delta^{\rightarrow}$ denote the time travelling from $\pi - \mathcal{D}_m^\star(0)$ to 0. Such argument is supported by Lemma F.10, as it takes equal time for $\mathcal{D}_m^\star(t)$ to travel from $\pi - \mathcal{D}_m^\star(0)$ to $\pi/2$ and from $\pi/2$ to $\pi - \mathcal{D}_m^\star(0)$. Also, when $\mathcal{D}_m^\star(t)$ reaches $\pi - \mathcal{D}_m^\star(0)$, we have $\beta_m^\star(t) = \kappa_{\text{init}}$ due to the symmetry, such that the remaining convergence time is equal to $\Delta t_\delta^{\rightarrow}$.

Below are some useful properties. Under the initialization in Assumption 5.1, Lemma F.9 ensures

$$\alpha_m^\star(t)^2 = 2\beta_m^\star(t)^2 - \kappa_{\text{init}}^2, \qquad \forall t \in \mathbb{R}^+. \tag{F.43}$$

Following this, we can characterize the dynamics as follows:

$$\partial_t \beta_m^\star(t) = p \cdot (2\beta_m^\star(t)^2 - \kappa_{\mathsf{init}}^2) \cdot \cos(\mathcal{D}_m^\star(t)), \tag{F.44a}$$

$$\partial_t \mathcal{D}_m^\star(t) = -p \cdot (6\beta_m^\star(t) - \kappa_{\mathsf{init}}^2/\beta_m^\star(t)) \cdot \sin(\mathcal{D}_m^\star(t)). \tag{F.44b}$$

Hence, we have $\mathcal{D}_m^\star(t)$ is monotonely decreasing, and $\beta_m^\star(t)$ first decreases when $\mathcal{D}_m^\star(t) \in (\pi/2, \pi)$ and increases thereafter. Besides, it follows from (F.43) that $\beta_m^\star(t) \geq \kappa_{\mathsf{init}}/\sqrt{2}$ for all $t \in \mathbb{R}^+$.

**Part I: Travelling time from $\mathcal{D}_m^\star(0)$ to $\pi/2$ with $\mathcal{D}_m^\star(0) \in (\pi/2, \pi)$.** We consider $t \in (0, \Delta t_{\pi/2}^\rightarrow]$ where we define $\Delta t_{\pi/2}^\rightarrow = \min\{t \in \mathbb{R}^+ : \mathcal{D}_m^\star(t) \leq \pi/2\}$. Based on (F.44b), by definition, we have

$$\partial_t \mathcal{D}_m^\star(t) \geq -p \cdot (6\beta_m^\star(t) - \kappa_{\mathsf{init}}^2/\beta_m^\star(t)) \geq -5p \cdot \kappa_{\mathsf{init}},$$

where the last inequality uses $6\beta_m^\star(t) - \kappa_{\mathsf{init}}^2/\beta_m^\star(t)$ is monotonically increasing on $\mathbb{R}^+$ and $\beta_m^\star(t) \in [\kappa_{\mathsf{init}}/\sqrt{2}, \kappa_{\mathsf{init}}]$ since $\beta_m^\star(t)$ is monotonically decreasing throughout the stage. Following this, we can lower bound $\mathcal{D}_m^\star(t)$ by $\mathcal{D}_m^\star(t) \geq \mathcal{D}_m^\star(0) - 5p \cdot \kappa_{\mathsf{init}} \cdot t$ for all $t \leq t_1^\epsilon$. Thus, we have

$$\Delta t_{\pi/2}^\rightarrow \geq \frac{\mathcal{D}_m^\star(0) - \mathcal{D}_m^\star(\Delta t_{\pi/2}^\rightarrow)}{5p \cdot \kappa_{\mathsf{init}}} = \frac{\mathcal{D}_m^\star(0) - \pi/2}{5p \cdot \kappa_{\mathsf{init}}},$$

On the other side, (F.44b) implies that $\partial_t \mathcal{D}_m^\star(t) \leq 0$ such that $\mathcal{D}_m^\star(t) \leq \mathcal{D}_m^\star(0)$. Then, we have

$$\partial_t \mathcal{D}_m^\star(t) \leq -p \cdot (6\beta_m^\star(t) - \kappa_{\mathsf{init}}^2/\beta_m^\star(t)) \cdot \sin(\mathcal{D}_m^\star(0)) \leq -2\sqrt{2}p \cdot \kappa_{\mathsf{init}} \cdot \sin(\mathcal{D}_m^\star(0)).$$

Similarly, we can upper bound $\Delta t_{\pi/2}^\rightarrow$. By combining the arguments above, we can reach the conclusion that

$$\Delta t_{\pi/2}^\rightarrow \asymp (p \cdot \kappa_{\mathsf{init}})^{-1} \cdot \{\mathcal{D}_m^\star(0) - \pi/2\}.$$

**Part II: Convergence time from $\mathcal{D}_m^\star(0)$ to $0$ with $\mathcal{D}_m^\star(0) \in (0, \pi/2)$.** Consider a small error level $\delta > 0$, and the convergence time is formalized as $\Delta t_\delta^\rightarrow = \min\{t \in \mathbb{R}^+ : \sin(\mathcal{D}_m^\star(t)) \leq \delta\}$. Note that $\mathcal{D}_m^\star(t)$ is monotonically decreasing and $\beta_m^\star(t)$ is monotonically increasing in this stage. Also,

$$\sin(\mathcal{D}_m^\star(t)) \cdot \beta_m^\star(t) \cdot \alpha_m^\star(t)^2 = \sin(\mathcal{D}_m^\star(t)) \cdot \beta_m^\star(t) \cdot (2\beta_m^\star(t)^2 - \kappa_{\mathsf{init}}^2) = \sin(\mathcal{D}_m^\star(0)) \cdot \kappa_{\mathsf{init}}^3,$$

following (F.43), Lemma F.9 and $\beta_m^\star(0) = \kappa_{\mathsf{init}}$ as specified in Assumption 5.1. By definition,

$$\sin(\mathcal{D}_m^\star(0))/\delta \cdot \kappa_{\mathsf{init}}^3 = \beta_m^\star(\Delta t_\delta^\rightarrow) \cdot (2\beta_m^\star(\Delta t_\delta^\rightarrow)^2 - \kappa_{\mathsf{init}}^2) \asymp \beta_m^\star(\Delta t_\delta^\rightarrow)^3.$$

Hence, we have $\beta_m^\star(\Delta t_\delta^\rightarrow)/\kappa_{\mathsf{init}} \asymp \sqrt[3]{\sin(\mathcal{D}_m^\star(0))/\delta}$. Following (F.44a), it holds that

$$\partial_t \log\left(\frac{\beta_m^\star(t) - \kappa_{\mathsf{init}}/\sqrt{2}}{\beta_m^\star(t) + \kappa_{\mathsf{init}}/\sqrt{2}}\right) = \frac{2\sqrt{2} \cdot \kappa_{\mathsf{init}} \cdot \partial_t \beta_m^\star(t)}{2\beta_m^\star(t)^2 - \kappa_{\mathsf{init}}^2} = 2\sqrt{2} \cdot \kappa_{\mathsf{init}} \cdot p \cdot \cos(\mathcal{D}_m^\star(t)) \asymp \kappa_{\mathsf{init}} \cdot p,$$

since $\cos(\mathcal{D}_m^\star(t)) \in [\cos(\mathcal{D}_m^\star(0)), 1]$. Hence, by integrating over time $(0, \Delta t_\delta^\rightarrow]$, we can show that

$$\log\left(\frac{\beta_m^\star(\Delta t_\delta^\rightarrow) - \kappa_{\mathsf{init}}/\sqrt{2}}{\beta_m^\star(\Delta t_\delta^\rightarrow) + \kappa_{\mathsf{init}}/\sqrt{2}}\right) + \log(3 + 2\sqrt{2}) \asymp \kappa_{\mathsf{init}} \cdot p \cdot \Delta t_\delta^\rightarrow. \tag{F.45}$$

Next, we bound the scale of the term within the logarithm. For a small tolerance $\delta = o(1)$, we have

$$\frac{\beta_m^\star(\Delta t_\delta^\rightarrow) - \kappa_{\mathsf{init}}/\sqrt{2}}{\beta_m^\star(\Delta t_\delta^\rightarrow) + \kappa_{\mathsf{init}}/\sqrt{2}} = 1 - 2(\sqrt{2} \cdot \beta_m^\star(\Delta t_\delta^\rightarrow)/\kappa_{\mathsf{init}} + 1)^{-1} = 1 - \Theta(\sqrt[3]{\delta/\sin(\mathcal{D}_m^\star(0))}). \tag{F.46}$$

Thus, by combing the arguments in (F.45) and (F.46), we can conclude that

$$\Delta t_\delta^\rightarrow \asymp (p \cdot \kappa_{\mathsf{init}})^{-1} \cdot \{1 - \sqrt[3]{\delta/\sin(\mathcal{D}_m^\star(0))}\},$$

where we use the fact that $\log(1 - x) \asymp x$ for small $x > 0$.

Based on the results in Part I and Part II, for any initial phase difference $\mathcal{D}_m^\star(0) \in (0, \pi)$ and sufficiently small error tolerance $\delta \in (0, 1)$, by symmetry, the convergence time is of level

$$t_\delta \asymp (p\kappa_{\mathsf{init}})^{-1} \cdot \{1 - (\sin(\mathcal{D}_m^\star(0))/\delta)^{-1/3} + \max\{\pi/2 - |\mathcal{D}_m^\star(0) - \pi|, 0\}\},$$

where we let $(x)_+ = \max\{x, 0\}$ denote the ReLU function.

**Part III: Preservation of Uniform Phase Distribution and Double-Phase Convergence.** Recall that Lemma F.9 gives there exists constant $C_{\text{prod}} \in \mathbb{R}$ such that

$$\sin(\mathcal{D}_m^\star(t)) \cdot \beta_m^\star(t) \cdot \alpha_m^\star(t)^2 = C_{\text{prod}}.$$

Following this, we can write the dynamics of $\phi_m^\star(t)$ and $\psi_m^\star(t)$ as

$$\partial_t \exp(i\phi_m^\star(t)) = 2p \cdot C_{\text{prod}}/\alpha_m^\star(t)^2 \cdot \exp\left(i\left\{\phi_m^\star(t) - \pi/2\right\}\right),$$
$$\partial_t \exp(i\psi_m^\star(t)) = p \cdot C_{\text{prod}}/\beta_m^\star(t)^2 \cdot \exp\left(i\left\{\psi_m^\star(t) + \pi/2\right\}\right).$$
(F.47)

As established previously, the magnitudes of the learned parameters, $\alpha_m^\star(t)$ and $\beta_m^\star(t)$, tend to infinity as $t \to \infty$. This divergence drives the convergence of the corresponding phases to fixed values, $\phi_m^\star(\infty)$ and $\psi_m^\star(\infty)$, which are determined by the initialization. Furthermore, Theorem 5.3 proves that the misalignment term $\mathcal{D}_m^\star(t)$ converges to zero. This directly implies that the limiting phases must satisfy the phase alignment condition: $2\phi_m^\star(\infty) = \psi_m^\star(\infty)$.

Denote $\exp(i\phi_m^\star(t)) = z(t)$, then by (F.47), $z(t)$ is continuously differentiable with respect to $t$. Consider

$$\Phi_m^\star(t) = \phi_m^\star(0) + \int_0^t \Im(\bar{z}(s)z'(s))\mathrm{d}s,$$

then we can check that $\Phi_m^\star(t)$ is continuously differentiable, and by differentiating both sides, we can also check that it satisfies $\exp(i\Phi_m^\star(t)) = z(t)$. Therefore, $\phi_m^\star(t) = \Phi_m^\star(t) \bmod 2\pi$. By direct calculation,

$$\partial_t \Phi_m^\star(t) = -2p \cdot C_{\text{prod}} \cdot \alpha_m^\star(t)^{-2},$$

which shows

$$\Phi_m^\star(t) = \phi_m^\star(0) - 2p \int_0^t C_{\text{prod}} \cdot \alpha_m^\star(s)^{-2}\mathrm{d}s.$$

Since $\alpha_m^\star(t)$ only depends on (F.43),(F.44a) and (F.44b), so once given $(\alpha_m^\star(0), \beta_m^\star(0), \mathcal{D}_m^\star(0))$, $\alpha_m^\star(t)$ is independent of $\phi_m^\star(0)$ for all $t$. In this case, conditional on $(\alpha_m^\star(0), \beta_m^\star(0), \mathcal{D}_m^\star(0))$, $\phi_m^\star(t)$ equals to $\phi_m^\star(0)$ plus some deterministic function up to $\bmod 2\pi$. Since the map

$$(\phi_m^\star(0), \psi_m^\star(0)) \to (\phi_m^\star(0), \mathcal{D}_m^\star(0) = 2\phi_m^\star(0) - \psi_m^\star(0))$$

has determinant $-1$, $\phi_m^\star(0)$ and $\mathcal{D}_m^\star(0)$ are i.i.d. Uniformly distributed on $(-\pi, \pi]$. Combining the above two arguments, we establish that $\phi_m^\star(t)$ is uniformly distributed on $(-\pi, \pi]$.

For $\psi_m^\star(t)$, we can establish the proof in an almost identical way that $\psi_m^\star(t)$ is uniformly distributed on $[0, 2\pi)$ for all $t$; due to the space limit, we omit the full proof here. As a result, $\phi_m^\star(\infty)$ and $\psi_m^\star(\infty)$ are both uniformly distributed over $[0, 2\pi)$.

Combining with the fact that $2\phi_m^\star(\infty) = \psi_m^\star(\infty)$ for any given initialization, we know the joint measure of $(\phi_m^\star(\infty), \psi_m^\star(\infty))$ is degenerated on the (periodic) line $2\phi = \psi$ inside the region $(\phi, \psi) \in (-\pi, \pi]^2$. Since the marginals of them are both uniform, the joint limiting measure is then given by $\rho_\infty = T_\# \lambda$, where $T : \varphi \mapsto (\varphi, 2\varphi) \bmod 2\pi$. Summarizing all the above, we finish the proof.

□

### F.5.1 PROOF OF AUXILIARY LEMMA F.8, F.9 AND F.10

*Proof of Lemma F.8.* Following the same argument in the proof of Theorem 5.2, by pushing the approximation error to 0, we can show an exact single-frequency pattern:

$$\alpha_m^k(t) = \beta_m^k(t) \equiv 0, \qquad \forall t \in \mathbb{R}^+, k \neq k^\star.$$

Formally, this result holds under the initialization in Assumption 5.1, which can be justified using a matrix ODE argument over $u_m^k(t) = (\alpha_m^k(t), \beta_m^k(t))^\top$ with zero initial value. Then, the dynamics of the original parameter can be simplified to a coefficient only related to $k^\star$. For all $j \in [p]$, we have

$$\partial_t \theta_m[j](t) = 2p \cdot \alpha_m^\star(t) \cdot \beta_m^\star(t) \cdot \cos(\omega_k j + \psi_m^\star(t) - \phi_m^\star(t)),$$
(F.48a)
$$\partial_t \xi_m[j](t) = p \cdot \alpha_m^\star(t)^2 \cdot \cos(\omega_{k^\star} j + 2\phi_m^\star(t)).$$
(F.48b)

Recall $\partial_t g_m[j](t) = \langle b_j, \partial_t \theta_m(t) \rangle$, by simple calculation, it holds that

$$\partial_t g_m[2k^\star](t) = \sqrt{2} \cdot p^{3/2} \cdot \alpha_m^\star(t) \cdot \beta_m^\star(t) \cdot \cos\left(\psi_m^\star(t) - \phi_m^\star(t)\right),$$
$$\partial_t g_m[2k^\star + 1](t) = -\sqrt{2} \cdot p^{3/2} \cdot \alpha_m^\star(t) \cdot \beta_m^\star(t) \cdot \sin\left(\psi_m^\star(t) - \phi_m^\star(t)\right),$$

and similarly, by using $\partial_t r_m[j](t) = \langle b_j, \partial_t \xi_m(t)\rangle$, we can obtain that

$$\partial_t r_m[2k^\star](t) = p^{3/2}/\sqrt{2} \cdot \alpha_m^\star(t)^2 \cdot \cos\left(2\phi_m^\star(t)\right),$$

$$\partial_t r_m[2k^\star + 1](t) = -p^{3/2}/\sqrt{2} \cdot \alpha_m^\star(t)^2 \cdot \sin\left(2\phi_m^\star(t)\right),$$

where the additional $\sqrt{2/p}$ arises from the normalization factor in $b_j$'s (see §A.3). Since the magnitudes follows $\alpha_m^\star = \sqrt{2/p} \cdot \|g_m^\star\|$ and $\beta_m^\star = \sqrt{2/p} \cdot \|r_m^\star\|$, by applying the chain rule, then

$$\partial_t \alpha_m^\star(t) = 2p \cdot \alpha_m^\star(t) \cdot \beta_m^\star(t) \cdot \cos\left(2\phi_m^\star(t) - \psi_m^\star(t)\right), \tag{F.49a}$$

$$\partial_t \beta_m^\star(t) = p \cdot \alpha_m^\star(t)^2 \cdot \cos\left(2\phi_m^\star(t) - \psi_m^\star(t)\right). \tag{F.49b}$$

Next, we understand the evolution of phases by tracking the dynamics of $\exp(i\phi_m^\star(t))$ and $\exp(i\psi_m^\star(t))$ via Euler's formula. Note that $\phi_m^\star(t)$ and $\psi_m^\star(t)$ cannot be directly tracked via ODEs due to abrupt jumps from $-\pi$ to $\pi$, which arise from the use of $\mathrm{atan2}(\cdot)$ function in definitions (see §A.3). By definition and the chain rule, it follows that

$$\partial_t \cos(\phi_m^\star(t)) = \sqrt{\frac{2}{p}} \cdot \partial_t \left(\frac{g_m[2k^\star](t)}{\alpha_m^*(t)}\right)$$

$$= \sqrt{\frac{2}{p}} \cdot \left\{\frac{\partial_t g_m[2k^\star](t)}{\alpha_m^*(t)} - \frac{\partial_t \alpha_m^*(t)}{\alpha_m^*(t)} \cdot \frac{g_m[2k^\star](t)}{\alpha_m^*(t)}\right\}$$

$$= 2p \cdot \beta_m^\star(t) \cdot \cos\left(\psi_m^\star(t) - \phi_m^\star(t)\right)$$

$$\quad - 2p \cdot \beta_m^\star(t) \cdot \cos(\phi_m^\star(t)) \cdot \cos\left(2\phi_m^\star(t) - \psi_m^\star(t)\right)$$

$$= 2p \cdot \beta_m^\star(t) \cdot \sin(\phi_m^\star(t)) \cdot \sin\left(2\phi_m^\star(t) - \psi_m^\star(t)\right),$$

where the second equality uses $\cos(\phi_m^\star(t)) = \sqrt{2/p} \cdot g_m[2k^\star](t)/\alpha_m^*(t)$ and the last one results from the trigonometric indentity. Similarly, we have

$$\partial_t \sin(\phi_m^\star(t)) = -2p \cdot \beta_m^\star(t) \cdot \cos(\phi_m^\star(t)) \cdot \sin\left(2\phi_m^\star(t) - \psi_m^\star(t)\right),$$

which gives that

$$\partial_t \exp(i\phi_m^\star(t)) = 2p \cdot \beta_m^\star(t) \cdot \sin\left(2\phi_m^\star(t) - \psi_m^\star(t)\right) \cdot \exp\left(i\left\{\phi_m^\star(t) - \pi/2\right\}\right). \tag{F.50}$$

Following a similar argument, we can show that

$$\partial_t \exp(i\psi_m^\star(t)) = p \cdot \frac{\alpha_m^\star(t)^2}{\beta_m^\star(t)} \cdot \sin\left(2\phi_m^\star(t) - \psi_m^\star(t)\right) \cdot \exp\left(i\left\{\psi_m^\star(t) + \pi/2\right\}\right). \tag{F.51}$$

Thanks to the initialization and preservation of the single-frequency, the $2p$-dimensional dynamical system can be tracked via a four-particle system with $\alpha_m^\star$, $\beta_m^\star$, $\phi_m^\star$, and $\psi_m^\star$, whose dynamics are given by (F.49a), (F.49b), (F.50) and (F.51). Furthermore, note that

$$\partial_t \exp(2i\phi_m^\star(t)) = 2\exp(i\phi_m^\star(t)) \cdot \partial_t \exp(i\phi_m^\star(t))$$

$$= 4p \cdot \beta_m^\star(t) \cdot \sin\left(2\phi_m^\star(t) - \psi_m^\star(t)\right) \cdot \exp\left(i\left\{2\phi_m^\star(t) - \pi/2\right\}\right). \tag{F.52}$$

Based on (F.51) and (F.52), by denoting $\mathcal{D}_m^\star(t) = 2\phi_m^\star(t) - \psi_m^\star(t) \bmod 2\pi$, we obtain that

$$\partial_t \exp(i\mathcal{D}_m^\star(t)) = \frac{\partial_t \exp(2i\phi_m^\star(t))}{\exp(i\psi_m^\star(t))} - \frac{\exp(2i\phi_m^\star(t)) \cdot \partial_t \exp(i\psi_m^\star(t))}{\exp(2i\psi_m^\star(t))}$$

$$= 4p \cdot \beta_m^\star(t) \cdot \sin\left(\mathcal{D}_m^\star(t)\right) \cdot \exp\left(i\left\{\mathcal{D}_m^\star(t) - \pi/2\right\}\right)$$

$$\quad - p \cdot \frac{\alpha_m^\star(t)^2}{\beta_m^\star(t)} \cdot \sin\left(\mathcal{D}_m^\star(t)\right) \cdot \exp\left(i\left\{\mathcal{D}_m^\star(t) + \pi/2\right\}\right)$$

$$= p \cdot \left(4\beta_m^\star(t) + \frac{\alpha_m^\star(t)^2}{\beta_m^\star(t)}\right) \cdot \sin\left(\mathcal{D}_m^\star(t)\right) \cdot \exp\left(i\{\mathcal{D}_m^\star(t) - \pi/2\}\right). \tag{F.53}$$

By combining (F.49a), (F.49b) and (F.53), we complete the proof. □

*Proof of Lemma F.9.* Following the simplified main flow in the Fourier domain (see Lemma F.8), it is easy to show that $\alpha_m^\star(t)^2 - 2\beta_m^\star(t)$ is a constant throughout the gradient flow since

$$\partial_t\{\alpha_m^\star(t)^2 - 2\beta_m^\star(t)^2\} = 2\alpha_m^\star(t) \cdot \partial_t \alpha_m^\star(t) - 4\beta_m^\star(t) \cdot \partial_t \beta_m^\star(t) = 0.$$

Hence, there exists an initialization-dependent constant $C_{\mathsf{diff}}$ such that

$$\alpha_m^\star(t)^2 = 2\beta_m^\star(t)^2 + C_{\mathsf{diff}}, \qquad \forall t \in \mathbb{R}^+.$$

Moreover, by applying the chain rule, we can deduce that

$$\begin{aligned}
\partial_t\{\alpha_m^\star(t)^2 \cdot \beta_m^\star(t)\} &= \alpha_m^\star(t)^2 \cdot \partial_t \beta_m^\star(t) + \partial_t\{\alpha_m^\star(t)^2\} \cdot \beta_m^\star(t) \\
&= \alpha_m^\star(t)^2 \cdot \partial_t \beta_m^\star(t) + 2\partial_t\{\beta_m^\star(t)^2\} \cdot \beta_m^\star(t) \\
&= p \cdot \alpha_m^\star(t)^4 \cdot \cos(\mathcal{D}_m^\star(t)) + 4\beta_m^\star(t)^2 \cdot p \cdot \alpha_m^\star(t)^2 \cdot \cos(\mathcal{D}_m^\star(t)) \\
&= p \cdot \alpha_m^\star(t)^2 \cdot \{\alpha_m^\star(t)^2 + 4\beta_m^\star(t)^2\} \cdot \cos(\mathcal{D}_m^\star(t)).
\end{aligned}$$

Following this, we can compute the time derivative of $\sin(\mathcal{D}_m^\star(t)) \cdot \beta_m^\star(t) \cdot \alpha_m^\star(t)^2$, following that

$$\begin{aligned}
\partial_t&\{\sin(\mathcal{D}_m^\star(t)) \cdot \beta_m^\star(t) \cdot \alpha_m^\star(t)^2\} \\
&= \partial_t \sin(\mathcal{D}_m^\star(t)) \cdot \beta_m^\star(t) \cdot \alpha_m^\star(t)^2 + \sin(\mathcal{D}_m^\star(t)) \cdot \partial_t\{\alpha_m^\star(t)^2 \cdot \beta_m^\star(t)\} \\
&= -\cos(\mathcal{D}_m^\star(t)) \cdot \alpha_m^\star(t)^2 \cdot p \cdot \left(4\beta_m^\star(t)^2 + \alpha_m^\star(t)^2\right) \cdot \sin\left(\mathcal{D}_m^\star(t)\right) \\
&\quad + \sin(\mathcal{D}_m^\star(t)) \cdot p \cdot \alpha_m^\star(t)^2 \cdot \{\alpha_m^\star(t)^2 + 4\beta_m^\star(t)^2\} \cdot \cos(\mathcal{D}_m^\star(t)) = 0,
\end{aligned}$$

where the second equality uses (F.41) in Lemma F.8. Therefore, there exists constant $C_{\mathsf{prod}}$ such that $\sin(\mathcal{D}_m^\star(t)) \cdot \beta_m^\star(t) \cdot \alpha_m^\star(t)^2 = C_{\mathsf{prod}}$ for all $t \in \mathbb{R}^+$.

Finally, we show that $\mathcal{D}_m^\star(t)$ remains within the half-space where it is initialized, which means $\mathcal{D}_m^\star(t) \in (\iota\pi, (\iota+1)\pi)$ for $\iota \in \{-1, 0\}$ determined by the initial state $\mathcal{D}_m^\star(0) \in (\iota\pi, (\iota+1)\pi)$. By Lemma F.9, we always have $\sin(\mathcal{D}_m^\star(t)) \neq 0$, so $\mathcal{D}_m^\star(t)$ will never reach $\iota\pi$ for any $\iota$. This ensures no jump behavior occurs for $\mathcal{D}_m^\star(t)$, allowing us to directly track its dynamics. Following this, by applying chain rule over (F.53), we can reach that

$$\partial_t \mathcal{D}_m^\star(t) = -p \cdot \left(4\beta_m^\star(t) + \alpha_m^\star(t)^2/\beta_m^\star(t)\right) \cdot \sin\left(\mathcal{D}_m^\star(t)\right),$$

which completes the proof. $\qquad\square$

*Proof of Lemma F.10.* Based on the results in Lemma F.8 and F.9, we reduce the main flow into a one-dimensional dynamical system characterized by $\beta_m^\star(t)$. Specifically, we have

$$\begin{aligned}
\partial_t \beta_m^\star(t) &= p \cdot \alpha_m^\star(t)^2 \cdot \cos(\mathcal{D}_m^\star(t)) \\
&= p \cdot (2\beta_m^\star(t)^2 + C_{\mathsf{diff}}) \cdot \mathrm{sign}\{\cos(\mathcal{D}_m^\star(t))\} \cdot \sqrt{1 - \frac{C_{\mathsf{prod}}^2}{\beta_m^\star(t)^2 \cdot (2\beta_m^\star(t)^2 + C_{\mathsf{diff}})^2}} \\
&:= \varsigma(\beta_m^\star(t)) \cdot \mathrm{sign}\{\cos(\mathcal{D}_m^\star(t))\}.
\end{aligned}$$

As given in (F.42), due to the nonnegativity of the magnitudes, we can show that $\mathcal{D}_m^\star(t)$ is monotonely decreasing if $\mathcal{D}_m^\star(0) \in (\pi/2, \pi)$. We consider $s = t - t_{\pi/2}$ for $t \in [t_{\pi/2}, 2t_{\pi/2})$ and $r = t_{\pi/2} - t$ for $t \in (0, t_{\pi/2}]$, where $t_{\pi/2}$ denote the hit time that $\mathcal{D}_m^\star(t_{\pi/2}) = \pi/2$. Following this, we have

$$\partial_s \beta_m^\star(s) = \partial_t \beta_m^\star(t - t_{\pi/2}) = -\varsigma(\beta_m^\star(s)), \quad \partial_r \beta_m^\star(r) = -\partial_t \beta_m^\star(t_{\pi/2} - t) = -\varsigma(\beta_m^\star(r)).$$

Here, we decompose $\partial_t \beta_m^\star(t)$ within time $[0, 2t_{\pi/2}]$ into a backward process within time $(0, t_{\pi/2}]$ and a forward process within time $[t_{\pi/2}, 2t_{\pi/2}]$ respectively. Starting from time $s = r = 0$, where the initial value is both given by $\beta_m^\star(t_{\pi/2})$, since $\varsigma$ is locally Lipschitz, by the uniqueness of the ODE solution, for $s = r$, we have $\beta_m^\star(s) = \beta_m^\star(r)$, i.e., $\beta_m^\star(t_{\pi/2} + \Delta t) = \beta_m^\star(t_{\pi/2} - \Delta t)$ for all $\Delta t \in [0, t_{\pi/2})$. Furthermore, by combining Lemma F.9, the monotonicity of $\mathcal{D}_m^\star(t)$ and the arguments above, we can show that $\mathcal{D}_m^\star(t_{\pi/2} - \Delta t) + \mathcal{D}_m^\star(t_{\pi/2} + \Delta t) = \pi$, which completes the proof. $\qquad\square$

# G  PROOF OF RESULTS FOR THEORETICAL EXTENSIONS IN SECTION D

## G.1  PROOF OF COROLLARY D.1: PHASE LOTTERY TICKET

We first formalize the random multiple frequency initialization as follows.

**Assumption G.1.** *For each neuron $m \in [M]$, the parameters $(\xi_m, \theta_m)$ are initialized as*

$$\theta_m(0) \sim \kappa_{\text{init}} \cdot \sqrt{p/2} \cdot \sum_{k=1}^{(p-1)/2} \left( \varrho_{1,k}[1] \cdot b_{2k} + \varrho_{1,k}[2] \cdot b_{2k+1} \right),$$

$$\xi_m(0) \sim \kappa_{\text{init}} \cdot \sqrt{p/2} \cdot \sum_{k=1}^{(p-1)/2} \left( \varrho_{2,k}[1] \cdot b_{2k} + \varrho_{2,k}[2] \cdot b_{2k+1} \right),$$

*where $\varrho_{r,k} \overset{i.i.d.}{\sim} \mathrm{Unif}(\mathbb{S}^1)$ for all $k$ and $r \in \{1, 2\}$, and $\kappa_{\text{init}} > 0$ denotes a small initialization scale.*

This is the natural extension of Assumption 5.1 to multiple frequencies, and the arguments in §F, i.e., Lemma F.8, F.9 and F.10, go through with only routine modifications thanks to the neuron decoupling and the orthogonality of frequencies. We first state the formal version of Corollary G.2.

**Corollary G.2** (Formal Statement of Corollary D.1). *Consider a random initialization following Assumption G.1, and let $k^\star$ denote the winning frequency given by $k^\star = \min_k \widetilde{\mathcal{D}}_m^k(0)$. For a given $\varepsilon \in (0, 1)$, define the dominance time $t_\varepsilon$ as*

$$t_\varepsilon := \inf\{t \in \mathbb{R}^+ : \max_{k \neq k^\star} \beta_m^k(t)/\beta_m^\star(t) \leq \varepsilon\}.$$

*Then, with probability at least $1 - \widetilde{\Theta}(p^{-c})$, where $c > 0$ satisfying $p \gtrsim c^4 \pi^2 e^{-2(1-c)}$, it holds that*

$$t_\varepsilon \lesssim \frac{\pi^2 p^{-(2c+3)}}{\kappa_{\text{init}}} + \frac{(c+1)\log p + \log \frac{1}{1-\varepsilon}}{p\kappa_{\text{init}} \cdot \{1 - 2c^2\pi^2 \cdot (\log p/p)^2\}}.$$

Before delving into the proof, we first establish a key property of the decoupled dynamics under this initialization—order preservation—under the initialization specified in G.1.

**Lemma G.3.** *Let $\sigma$ be the permutation that sorts the initial phase differences in non-decreasing order:*

$$\widetilde{\mathcal{D}}_m^{\sigma(1)}(0) \leq \widetilde{\mathcal{D}}_m^{\sigma(2)}(0) \leq \cdots \leq \widetilde{\mathcal{D}}_m^{\sigma\left(\frac{p-1}{2}\right)}(0),$$

*where $\widetilde{\mathcal{D}}_m^k(0) = \min\{\mathcal{D}_m^k(0), 2\pi - \mathcal{D}_m^k(0)\}$ represents the shortest circular distance for the initial phase. Under the initialization in Assumption G.1, the rank-ordering of the corresponding magnitudes $\beta_m^k(t)$ is inverted and preserved for all time $t \geq 0$:*

$$\beta_m^{\sigma(1)}(t) \geq \beta_m^{\sigma(2)}(t) \geq \cdots \geq \beta_m^{\sigma\left(\frac{p-1}{2}\right)}(t).$$

*Proof of Lemma G.3.* Please refer to §G.1 for a detailed proof. $\square$

Lemma G.3 states that, when neurons are decoupled and each frequency is initialized at the same scale $\kappa_{\text{init}} > 0$, the ordering of frequencies by magnitude $\beta_m^k$'s within each neuron remains fixed throughout the gradient flow, with larger magnitudes corresponding to smaller initial phase difference. Now we are ready to present the proof of Corollary G.2.

*Proof of Corollary G.2.* As specified in Assumption G.1, for all $m \in [M]$, we initialize $\varrho_{r,k} \overset{i.i.d.}{\sim} \mathrm{Unif}(\mathbb{S}^1)$ for all $r \in \{1, 2\}$ and $k \in [\frac{p-1}{2}]$. Thanks to the orthogonality among frequencies, each frequency evolves independently, so Lemmas F.8, F.9 and F.10 apply to every frequency $k$, not just the feature frequency $k^\star$. For fixed neuron $m$, by defining $\widetilde{\mathcal{D}}_m^k(0) = \min\{\mathcal{D}_m^k(0), 2\pi - \mathcal{D}_m^k(0)\}$, we have

$$\partial_t \beta_m^k(t) = p \cdot (2\beta_m^k(t)^2 - \kappa_{\text{init}}^2) \cdot \cos(\widetilde{\mathcal{D}}_m^k(t)), \tag{G.1a}$$

$$\partial_t \widetilde{\mathcal{D}}_m^k(t) = -p \cdot \left( 6\beta_m^k(t) - \kappa_{\text{init}}^2/\beta_m^k(t) \right) \cdot \sin\left( \widetilde{\mathcal{D}}_m^k(t) \right). \tag{G.1b}$$

**Step 1: Deriving Winning Frequency and Initial Phase Gap.** By Lemma G.3, the dynamics preserves the ordering of $\mathcal{D}_m^k$'s and $\beta_m^k$'s throughout the gradient flow. Specifically, at any time $t \in \mathbb{R}^+$, the ordering remains unchanged. Thus, the lottery ticket winner, i.e., frequency $k$ such that $\beta_m^k(t) \geq \beta_m^\tau(t)$ for all $\tau \neq k$, is given by $k^\star = \mathrm{argmin}_k \widetilde{\mathcal{D}}_m^k(0)$.

To demystify the dominance phenomenon, it suffices to focus on the growth of the magnitude of the winning frequency $k^\star$ and the second-dominant frequency $k^\sharp = \mathrm{argmin}_{k \neq k^\star} \widetilde{\mathcal{D}}_m^k(0)$. Under the

initialization as specified in Assumption G.1, with probability greater than $1 - \widetilde{\Theta}(p^{-c})$ for some constant $c \in (0, 1)$, we have the following good initialization:

$$
\begin{aligned}
\mathcal{E}_{\text{init}} &= \mathcal{E}_{\text{init}}^1 \cap \mathcal{E}_{\text{init}}^2 \cap \mathcal{E}_{\text{init}}^3 \\
&:= \left\{ \widetilde{\mathcal{D}}_m^\sharp(0) < \pi/2 \right\} \cap \left\{ \cos(\widetilde{\mathcal{D}}_m^\star(0)) \geq \cos(\widetilde{\mathcal{D}}_m^\sharp(0)) + \pi^2 p^{-2(c+1)} \right\} \\
&\quad \cap \left\{ \cos(\widetilde{\mathcal{D}}_m^\star(0)) \leq 1 - 2c^2\pi^2 \cdot (\log p/p)^2 \right\}.
\end{aligned}
\tag{G.2}
$$

This is because $\widetilde{\mathcal{D}}_m^k(0) \overset{\text{i.i.d.}}{\sim} \text{Unif}(0, \pi)$ based on a similar argument in Lemma F.6, and thus $\widetilde{\mathcal{D}}_m^\star(0)$ and $\widetilde{\mathcal{D}}_m^\sharp(0)$ are respectively the first- and the second- order statistics of $\frac{p-1}{2}$ i.i.d copies of $\text{Unif}(0, \pi)$, denoted by $U_{(i)}$'s. Notice that

$$
\mathbb{P}\left(\mathcal{E}_{\text{init}}^{1;c}\right) = \mathbb{P}\left(\forall i, \ U_{(i)} \geq \pi/2\right) + \mathbb{P}\left(\forall i > 1, \ U_{(i)} \geq \pi/2, \ U_{(1)} \leq \pi/2\right) = (p+1) \cdot 2^{-\frac{p+1}{2}} \lesssim p^{-c}.
\tag{G.3}
$$

Furthermore, if $p \gtrsim c^4\pi^2 e^{-2(1-c)}$, it holds that

$$
\begin{aligned}
\mathbb{P}\left(\mathcal{E}_{\text{init}}^{2;c}\right) &\leq \mathbb{P}\left(\left\{\cos(U_{(1)}) \leq \cos(U_{(2)}) + \pi^2 p^{-2(c+1)}\right\} \cap \mathcal{E}_{\text{init}}^1\right) + \mathbb{P}\left(\mathcal{E}_{\text{init}}^{1;c}\right) \\
&\lesssim \mathbb{P}\left(\left\{U_{(2)}^2 - U_{(1)}^2 - U_{(2)}^4/12 \leq 2\pi^2 p^{-2(c+1)}\right\} \cap \mathcal{E}_{\text{init}}^1\right) + p^{-c} \\
&\leq \mathbb{P}\left(\left\{U_{(2)}^2 - U_{(1)}^2 \leq 2\pi^2 p^{-2(c+1)} + 2(c\pi/p \cdot \log p)^4\right\} \cap \mathcal{E}_{\text{init}}^1\right) \\
&\quad + \mathbb{P}\left(\left\{U_{(2)}^4 \geq 24 \cdot (c\pi/p \cdot \log p)^4\right\} \cap \mathcal{E}_{\text{init}}^1\right) + p^{-c} \\
&\leq \mathbb{P}\left(U_{(2)}^2 - U_{(1)}^2 \leq 8\pi^2 p^{-2(c+1)}\right) + \mathbb{P}\left(U_{(2)} \geq 2c\pi/p \cdot \log p\right) + p^{-c},
\end{aligned}
\tag{G.4}
$$

where the second inequality uses $1 - x^2/2 \leq \cos(x) \leq 1 - x^2/2 + x^4/24$ for $x \in (0, \pi/2)$. Moreover, to bound the RHS of (G.4), we can show that

$$
\begin{aligned}
\mathbb{P}(U_{(2)}^2 - U_{(1)}^2 \leq 8\pi^2 p^{-2(c+1)}) &\leq \mathbb{P}(U_{(1)} \cdot (U_{(2)} - U_{(1)}) \leq 4\pi^2 p^{-2(c+1)}) \\
&\leq \mathbb{P}(U_{(1)} \leq 2\pi p^{-(c+1)}) + \mathbb{P}(U_{(2)} - U_{(1)} \leq 2\pi p^{-(c+1)}) \\
&= 2 - 2(1 - 2p^{-(c+1)})^{\frac{p-1}{2}} \lesssim p^{-c},
\end{aligned}
\tag{G.5}
$$

where the second inequality follows $U_{(1)} \overset{d}{=} U_{(2)} - U_{(1)}$. Furthermore, it holds that

$$
\begin{aligned}
\mathbb{P}(U_{(2)} \geq 2c\pi/p \cdot \log p) &= (1 - 2c/p \cdot \log p)^{\frac{p-1}{2}} + c(p-1)/p \cdot \log p \cdot (1 - 2c/p \cdot \log p)^{\frac{p-3}{2}} \\
&\leq (1 + c\log p) \cdot (1 - 2c/p \cdot \log p)^{\frac{p-3}{2}} \lesssim p^{-c} \log p.
\end{aligned}
\tag{G.6}
$$

By combining (G.4), (G.5) and (G.6), we have $\mathbb{P}\left(\mathcal{E}_{\text{init}}^{2;c}\right) \lesssim p^{-c} \log p$. Similarly, we can derive that

$$
\begin{aligned}
\mathbb{P}\left(\mathcal{E}_{\text{init}}^{3;c}\right) &= \mathbb{P}\left(\cos(U_{(1)}) \leq 1 - 2c^2\pi^2 \cdot (\log p/p)^2\right) \\
&\leq \mathbb{P}(U_{(1)} \geq 2c\pi/p \cdot \log p) = (1 - 2c/p \cdot \log p)^{\frac{p-1}{2}} \lesssim p^{-c},
\end{aligned}
\tag{G.7}
$$

where the inequality also uses $\cos(x) \geq 1 - x^2/2$ for $x \in (0, \pi/2)$ Based on (G.3),(G.4) and (G.7), the good initialization event $\mathcal{E}_{\text{init}}$ holds with a probability of at least $1 - \Theta(p^{-c} \log p)$. In the subsequent analysis, we assume that this event occurs.

**Step 2: Growth of Gap between Winning Frequency and Others.** Based on (G.1a), the dynamics for the log-magnitude follows

$$
\partial_t \log \beta_m^k(t) = \frac{\partial_t \beta_m^k(t)}{\beta_m^k(t)} = p \cdot (2\beta_m^k(t) - \kappa_{\text{init}}^2/\beta_m^k(t)) \cdot \cos(\widetilde{\mathcal{D}}_m^k(t)).
$$

To track the relative growth of two magnitudes, $\beta_m^\star(t)$ and $\beta_m^\sharp(t)$, we now examine the dynamics of their log-ratio $\partial_t \log \frac{\beta_m^\star(t)}{\beta_m^\sharp(t)}$, whose evolution is given by:

$$
\begin{aligned}
\partial_t \log \frac{\beta_m^\star(t)}{\beta_m^\sharp(t)} &= p \cdot (2\beta_m^\star(t) - \kappa_{\text{init}}^2/\beta_m^\star(t)) \cdot \cos(\widetilde{\mathcal{D}}_m^\star(t)) - p \cdot (2\beta_m^\sharp(t) - \kappa_{\text{init}}^2/\beta_m^\sharp(t)) \cdot \cos(\widetilde{\mathcal{D}}_m^\sharp(t)) \\
&= p \cdot (\beta_m^\star(t) - \beta_m^\sharp(t)) \cdot \{2 + \kappa_{\text{init}}^2/(\beta_m^\star(t) \cdot \beta_m^\sharp(t))\} \cdot \cos(\widetilde{\mathcal{D}}_m^\star(t)) \\
&\quad + p \cdot (2\beta_m^\sharp(t) - \kappa_{\text{init}}^2/\beta_m^\sharp(t)) \cdot \{\cos(\widetilde{\mathcal{D}}_m^\star(t)) - \cos(\widetilde{\mathcal{D}}_m^\sharp(t))\} \\
&\geq 2p \cdot \cos((\widetilde{\mathcal{D}}_m^\star(0)) \cdot (\beta_m^\star(t) - \beta_m^\sharp(t)) + p \cdot \kappa_{\text{init}} \cdot \{\cos(\widetilde{\mathcal{D}}_m^\star(t)) - \cos(\widetilde{\mathcal{D}}_m^\sharp(t))\}.
\end{aligned}
\tag{G.8}
$$

Here, we use (i) $\beta_m^\star(t) \geq \beta_m^\sharp(t)$ and $\widetilde{\mathcal{D}}_m^\star(t) \leq \widetilde{\mathcal{D}}_m^\sharp(t)$ for all $t \in \mathbb{R}^+$ based on the order preservation property in Lemma G.3, and (ii) under the good initialization $\mathcal{E}_{\mathsf{init}}$ where $\widetilde{\mathcal{D}}_m^\star(0), \widetilde{\mathcal{D}}_m^\sharp(0) \leq \frac{\pi}{2}$, we have $\partial_t \widetilde{\mathcal{D}}_m^\diamond(t) < 0$ and $\partial_t \beta_m^\diamond(t) > 0$ for all $(\diamond, t) \in \{\star, \sharp\} \cup \mathbb{R}^+$. Therefore, we have $\cos(\widetilde{\mathcal{D}}_m^\star(t)) \geq \cos(\widetilde{\mathcal{D}}_m^\star(0))$, $\beta_m^\sharp(t) \geq \beta_m^\sharp(0) = \kappa_{\mathsf{init}}$ and $2\beta_m^\sharp(t) - \kappa_{\mathsf{init}}^2/\beta_m^\sharp(t) \geq 2\beta_m^\sharp(0) - \kappa_{\mathsf{init}}^2/\beta_m^\sharp(0) = \kappa_{\mathsf{init}}$ under the initialization in Assumption G.1. Let $\rho_m(t) = \beta_m^\star(t)/\beta_m^\sharp(t)$. Following (G.8), we have

$$\partial_t \log \rho_m(t) \geq 2p \cdot \kappa_{\mathsf{init}} \cdot \cos(\widetilde{\mathcal{D}}_m^\star(0)) \cdot (\rho_m(t) - 1) \vee p \cdot \kappa_{\mathsf{init}} \cdot \{\cos(\widetilde{\mathcal{D}}_m^\star(t)) - \cos(\widetilde{\mathcal{D}}_m^\sharp(t))\},$$

Based on the first term in the right-hand side, a simple calculation shows that the dynamics satisfy:

$$\partial_t \log\left(\frac{\rho_m(t) - 1}{\rho_m(t)}\right) \geq 2p \cdot \kappa_{\mathsf{init}} \cdot \cos(\widetilde{\mathcal{D}}_m^\star(0)) > 0.$$

Given $\rho_m(0) = 1$, we can integrate this result over any interval $[s, t]$ to obtain a lower bound:

$$\rho_m(t) \geq \{1 + (1/\rho_m(s) - 1) \cdot \exp(2p \cdot \cos(\widetilde{\mathcal{D}}_m^\star(0)) \cdot \kappa_{\mathsf{init}} \cdot (t - s))\}^{-1}, \qquad \forall s \in (0, t]. \quad \text{(G.9)}$$

Following this, once the ratio $\rho_m(t)$ is larger than 1, the ratio $\rho_m(t)$ surpasses 1, it begins to grow super-exponentially, accelerating rapidly towards infinity. Motivated by this dynamics, our analysis proceeds in two stages: first, we show that $\rho_m(t)$ does not get stuck at the initial stationary point $\rho_m(t) \equiv 1$, and second, we quantify its rate of growth using (G.9).

**Step 2.1. Initial Growth of the Ratio Beyond Unity.** Consider a short initial time interval $(0, t_1]$, during which the model parameters remain close to their initial values while the ratio $\rho_m(t)$ quickly exceeds 1. Based on (G.1b), we have

$$|\cos(\widetilde{\mathcal{D}}_m^\star(t)) - \cos(\widetilde{\mathcal{D}}_m^\sharp(t)) - \cos(\widetilde{\mathcal{D}}_m^\star(0)) + \cos(\widetilde{\mathcal{D}}_m^\sharp(0))|$$

$$\leq 2 \max_{\diamond \in \{\star, \sharp\}} |\cos(\widetilde{\mathcal{D}}_m^\diamond(t)) - \cos(\widetilde{\mathcal{D}}_m^\diamond(0))|$$

$$\leq 2 \max_{\diamond \in \{\star, \sharp\}} \cos(\widetilde{\mathcal{D}}_m^\diamond(t)) = 2 \max_{\diamond \in \{\star, \sharp\}} \int_0^t \partial_s \cos(\widetilde{\mathcal{D}}_m^\diamond(s)) \mathrm{d}s$$

$$= 2p \cdot \max_{\diamond \in \{\star, \sharp\}} \int_0^t \left(6\beta_m^\diamond(s) - \kappa_{\mathsf{init}}^2/\beta_m^\diamond(s)\right) \cdot \sin(\widetilde{\mathcal{D}}_m^\diamond(s))^2 \mathrm{d}s$$

$$\leq 6p \cdot \max_{\diamond \in \{\star, \sharp\}} \int_0^t \beta_m^\diamond(s) \mathrm{d}s \leq 6pt \cdot \max_{\diamond \in \{\star, \sharp\}} \max_{0 \leq s \leq t} \beta_m^\diamond(s) = 6pt \cdot \beta_m^\star(t), \quad \text{(G.10)}$$

where the last inequality results from $\beta_m^\star(s) \leq \beta_m^\star(t)$ for all $s \in (0, t]$ and the rank preservation property, i.e., $\beta_m^\diamond(t) \leq \beta_m^\star(t)$ at any time $t$, as shown in Lemma G.3. Following (G.1a), we get

$$\partial_t \beta_m^\star(t) \leq p \cdot (2\beta_m^\star(t)^2 - \kappa_{\mathsf{init}}^2) \implies \beta_m^\star(t) \leq \kappa_{\mathsf{init}}/\sqrt{2} \cdot \coth(-\sqrt{2}p\kappa_{\mathsf{init}} \cdot t - \iota_1), \quad \forall t \in \mathbb{R}^+, \quad \text{(G.11)}$$

where we denote $\iota_1 = \mathrm{arccoth}(\sqrt{2})$. By choosing $c_g \in (0, 1)$, we define

$$t_1 := \inf\left\{s \in (0, t] : 3\sqrt{2}p\kappa_{\mathsf{init}} \cdot s \cdot \coth(-\sqrt{2}p\kappa_{\mathsf{init}} \cdot s - \iota_1) > c_g \cdot \pi^2 p^{-2(c+1)}\right\}.$$

Here, we choose a sufficiently small $c_g$ to ensure that $t_1$ is well-defined and finite before the system explodes. This choice makes $t_1$ correspondingly small and the following asymptotic result holds:

$$\coth(-\sqrt{2}p\kappa_{\mathsf{init}} \cdot t_1 - \iota_1) \asymp \sqrt{2} + p\kappa_{\mathsf{init}} \cdot t_1 \implies t_1 \asymp c_g \cdot \pi^2 p^{-(2c+3)}/\kappa_{\mathsf{init}}. \quad \text{(G.12)}$$

Recall from (G.2) that under the good initialization $\mathcal{E}_{\mathsf{init}}$, the initial cosine gap $\cos(\mathcal{D}_m^\star(0)) - \cos(\mathcal{D}_m^\sharp(0))$ is lower bounded by $\pi^2 p^{-2(c+1)}$. By combining (G.10), (G.11) and definition of $t_1$, we have

$$\cos(\widetilde{\mathcal{D}}_m^\star(t)) - \cos(\widetilde{\mathcal{D}}_m^\sharp(t))$$

$$\geq \cos(\widetilde{\mathcal{D}}_m^\star(0)) - \cos(\widetilde{\mathcal{D}}_m^\sharp(0)) - |\cos(\widetilde{\mathcal{D}}_m^\star(t)) - \cos(\widetilde{\mathcal{D}}_m^\sharp(t)) - \cos(\widetilde{\mathcal{D}}_m^\star(0)) + \cos(\widetilde{\mathcal{D}}_m^\sharp(0))|$$

$$\geq \pi^2 p^{-2(c+1)} - 6p \cdot \sup_{t \in (0, t_1]} t \cdot \beta_m^\star(t) \geq (1 - c_g) \cdot \pi^2 p^{-2(c+1)}, \quad \text{(G.13)}$$

for all $t \in (0, t_1]$. Building upon (G.12) and (G.13), we can show that

$$\log \rho_m(t_1) = \log \rho_m(0) + \int_0^{t_1} \cos(\mathcal{D}_m^\star(s)) - \cos(\mathcal{D}_m^\sharp(s)) \mathrm{d}s$$

$$\gtrsim c_g(1 - c_g) \cdot p\kappa_{\mathsf{init}} \cdot t_1 \cdot \pi^2 p^{-2(c+1)} \asymp \pi^4 p^{-4(c+1)},$$

and thus $\rho_m(t_1) \gtrsim \exp(1 + \pi^4 p^{-4(c+1)}) \asymp 1 + \pi^4 p^{-4(c+1)}$ for sufficiently large $p$.

**Step 2.2. Super-exponential Growth.** Let $\varepsilon > 0$ be the dominance threshold. We now derive the time $t_2$ required for the lower bound of the ratio to exceed this threshold, i.e., $\rho_m(t_2) > 1/\varepsilon$, such that $t_\varepsilon \le t_2 b$. Our starting point is the state at time $t_1$, after which we have $\rho_m(t_1) \asymp 1 + \pi^4 p^{-4(c+1)}$. Following (G.9), we have

$$\rho_m(t)^{-1} \le 1 + (1/\rho_m(t_1) - 1) \cdot \exp(2p \cdot \cos(\widetilde{\mathcal{D}}_m^\star(0)) \cdot \kappa_{\mathsf{init}} \cdot (t - t_1))$$

$$\lesssim 1 - \pi^4 p^{-4(c+1)} \cdot \exp(2p \cdot \{1 - 2c^2\pi^2 \cdot (\log p/p)^2\} \cdot \kappa_{\mathsf{init}} \cdot (t - t_1)),$$

where the last inequality results from $1/\rho_m(t_1) - 1 \asymp 1 - \rho_m(t_1)$ given $\rho_m(t_1)$ is close to 1, and the good initialization $\cos(\widetilde{\mathcal{D}}_m^\star(0)) \ge 1 - 2c^2\pi^2 \cdot (\log p/p)^2$ in (G.2). By choosing

$$t_2 = t_1 + \frac{4(c+1)\log p + \log\frac{1}{1-\varepsilon} - 4\log\pi}{2p\kappa_{\mathsf{init}} \cdot \{1 - 2c^2\pi^2 \cdot (\log p/p)^2\}} \asymp \frac{\pi^2 p^{-(2c+3)}}{\kappa_{\mathsf{init}}} + \frac{(c+1)\log p + \log\frac{1}{1-\varepsilon}}{p\kappa_{\mathsf{init}} \cdot \{1 - 2c^2\pi^2 \cdot (\log p/p)^2\}},$$

we can guarantee that $\rho_m(t_\varepsilon)^{-1} < \varepsilon$, which completes the proof. $\qquad\square$

### G.1.1 PROOF OF AUXILIARY LEMMA G.3

We begin by recalling the foundational results for a celebrated class of dynamical systems–known as cooperative systems–which enjoy a useful rank-preservation property (e.g., Smith, 1995). Before stating this formally, let us give a precise definition.

**Definition G.4** (Cooperative System). *Consider a $p$-convex set $\mathcal{S} \subset \mathbb{R}^d$ such that $tx + (1-t)xy \in \mathcal{S}$ for all $t \in [0,1]$ whenever $x, y \in \mathcal{S}$ and $x_- \le x_+$. Suppose $f : \mathcal{S} \mapsto \mathcal{S}$ is continuously differentiable. The dynamical system, defined by $\partial_t x_t = f(x_t)$, is called cooperative if $\frac{\partial f_i}{\partial x_j}(x) \ge 0$ for all $i \ne j$.*

In other words, a cooperative system's Jacobian has *nonnegative off-diagonal entries*, so increasing any coordinate of the state cannot decrease another in the next iteration. With this definition in hand, we can now state the key *monotonicity* property of cooperative systems.

**Lemma G.5.** *Consider a cooperative system $\partial_t x_t = f(x_t)$, and write $x \le y$ for $x, y \in \mathbb{R}^d$ if $x_i \le y_i$ for all $i \in \mathbb{R}^d$. Given two initial values $x_0^1 \le x_0^2$, then we have $x_t^1 \le x_t^2$ at all times $t \in \mathbb{R}^+$.*

*Proof of Lemma G.5.* Please refer to Kamke (1932); Hirsch (1982) for a detailed proof. $\qquad\square$

In what follows, we prove Lemma G.3, which is a direct application of Lemma G.5.

*Proof of Lemma G.3.* Recall that, by Lemmas F.9 and F.10, together with the orthogonality of the frequency basis, for every $k \in [\frac{p-1}{2}]$, the dynamical system is given by (G.1a) and (G.1b) with initial condition $\beta_m^k(0) = \kappa_{\mathsf{init}}$ for every frequency $k$.

We first show that the evolution of $\mathcal{D}_m^k(t)$ consistently shares the symmetric trajectory at any time $t$ if initialized symmetrically. Let $x(t) = (\beta_m^k(t), \mathcal{D}_m^k(t))$ and denote by $\varsigma(x(t))$ right-hand side of (G.1a), (G.1b), such that $\partial_t x(t) = \varsigma(x(t))$. Define the involution $I(\beta, \mathcal{D}) = (\beta, 2\pi - \mathcal{D})$ with its Jacobian following $\mathrm{d}I \equiv \mathrm{diag}(1, -1)$. A direct calculation shows that

$$\varsigma \circ I(\beta_m^k(t), \mathcal{D}_m^k(t)) - \mathrm{d}I \cdot \varsigma(\beta_m^k(t), \mathcal{D}_m^k(t)) \equiv 0,$$

i.e., the system is equivariant under $I$. By uniqueness of solutions, the solution with initial $x(0) = (\beta_m^k(0), 2\pi - \mathcal{D}_m^k(0))$ satisfies $x(t) = I(\beta_m^k(t), \mathcal{D}_m^k(t))$, so the two trajectories remain symmetric. Hence, it suffices to consider the dynamics with standardized initialization $\min\{\mathcal{D}_m^k(0), 2\pi - \mathcal{D}_m^k(0)\} \in (0, \pi]$. Following a similar argument in Lemma F.9, under the standardized initialization, we have $\mathcal{D}_m^k(t) \in (0, \pi)$ at all time $t$. To verify cooperativeness, we introduce $\widetilde{\beta}_m^k = -\beta_m^k$ and rewrite the dynamics in the new coordinates $(\bar{\beta}_m^k, \mathcal{D}_m^k)$. From (G.1a) and (G.1b) one obtains

$$\partial_t \widetilde{\beta}_m^k(t) = -p \cdot (2\widetilde{\beta}_m^k(t)^2 - \kappa_{\mathsf{init}}^2) \cdot \cos(\mathcal{D}_m^k(t)) := \varsigma_1(\beta_m^k(t), \mathcal{D}_m^k(t)),$$

$$\partial_t \mathcal{D}_m^k(t) = p \cdot (6\widetilde{\beta}_m^k(t) - \kappa_{\mathsf{init}}^2/\widetilde{\beta}_m^k(t)) \cdot \sin(\mathcal{D}_m^k(t)) := \varsigma_2(\beta_m^k(t), \mathcal{D}_m^k(t)),$$

and it is easy to check that the vector field is cooperative by

$$\frac{\partial \varsigma_1}{\partial \mathcal{D}_m^k} = \sin(\mathcal{D}_m^k(t)) > 0, \quad -\frac{\partial \varsigma_2}{\partial \widetilde{\beta}_m^k} = p \cdot (6 + \kappa_{\mathsf{init}}^2/\widetilde{\beta}_m^k(t)^2) > 0.$$

Thus, $(-\beta_m^k, \mathcal{D}_m^k)$ is cooperative, and by Lemma G.5, it preserves the initial ordering. Since $\beta_m^k(0) = \kappa_{\mathsf{init}}$ for all $k$ and phase difference $\mathcal{D}_m^k(0)$'s are distinct, it follows that

$$\mathcal{D}_m^k(0) \le \mathcal{D}_m^\tau(0) \implies \forall t \in \mathbb{R}^+, \ \widetilde{\beta}_m^k(t) \le \widetilde{\beta}_m^\tau(t) \implies \forall t \in \mathbb{R}^+, \ \beta_m^k(t) \ge \beta_m^\tau(t),$$

for every pair $k, \tau \in [\frac{p-1}{2}]$, which completes the proof. $\qquad\square$

### G.2 PROOF OF PROPOSITION D.3: DYNAMICS OF ReLU ACTIVATION

*Proof of Proposition D.3.* We begin by recalling from §F.2 that, for each fixed index $m$, the gradient with respect to the decoupled loss $\ell_m$ takes the form

$$\frac{\partial \ell}{\partial \theta_m[j]} = -2 \sum_{x \in \mathbb{Z}_p} \xi_m[m_p(x, j)] \cdot \mathbb{1}(\langle e_x + e_j, \theta_m \rangle \geq 0)$$

$$+ \frac{2}{p} \sum_{x \in \mathbb{Z}_p} \sum_{\tau=1}^{p} \xi_m[\tau] \cdot \mathbb{1}(\langle e_x + e_j, \theta_m \rangle \geq 0), \tag{G.14a}$$

$$\frac{\partial \ell_m}{\partial \xi_m[j]} = - \sum_{(x,y) \in \mathcal{S}_j^p} \max\{\langle e_x + e_y, \theta_m \rangle, 0\} + \frac{1}{p} \sum_{j=1}^{p} \sum_{(x,y) \in \mathcal{S}_j^p} \max\{\langle e_x + e_y, \theta_m \rangle, 0\}, \tag{G.14b}$$

for all $j \in [p]$. We first evaluate these gradients at the single-frequency $\theta_m[j] = \alpha_m^\star \cdot \cos(\omega_{k^\star} j + \phi_m^\star)$ and $\xi_m[j] = \beta_m^\star \cdot \cos(\omega_{k^\star} j + \psi_m^\star)$ for all $j$, and then to extract the DFT coefficients.

**Step 1: Gradient of $\xi_m$.** First observe that $\max\{x, 0\} = (x + |x|)/2$. Then, we have

$$\sum_{(x,y) \in \mathcal{S}_j^p} \sigma(\langle e_x + e_y, \theta_m \rangle) = \frac{1}{2} \sum_{(x,y) \in \mathcal{S}_j^p} \langle e_x + e_y, \theta_m \rangle + \frac{1}{2} \sum_{(x,y) \in \mathcal{S}_j^p} |\langle e_x + e_y, \theta_m \rangle|$$

$$= \frac{\alpha_m^\star}{2} \sum_{(x,y) \in \mathcal{S}_j^p} |\cos(\omega_{k^\star} x + \phi_m^\star) + \cos(\omega_{k^\star} y + \phi_m^\star)|, \tag{G.15}$$

Moreover, by applying the sum-to-product trigonometric identities, we can show that

$$\frac{1}{2} \sum_{(x,y) \in \mathcal{S}_j^p} |\cos(\omega_{k^\star} x + \phi_m^\star) + \cos(\omega_{k^\star} y + \phi_m^\star)|$$

$$= \sum_{(x,y) \in \mathcal{S}_j^p} |\cos(\omega_k(x + y)/2 + \phi_m^\star)| \cdot |\cos(\omega_k(x - y)/2)|$$

$$= |\cos(\omega_k j/2 + \phi_m^\star)| \cdot \sum_{x \in \mathbb{Z}_p} |\cos(\omega_k x/2)| \stackrel{p \to \infty}{=} \frac{2p}{\pi} \cdot |\cos(\omega_k j/2 + \phi_m^\star)|. \tag{G.16}$$

The last inequality uses the fact that for an odd prime $p$, $\{\omega_k x\}_{x \in \mathbb{Z}_p} = \{2kx\pi/p\}_{x \in \mathbb{Z}_p} = \{2\pi x/p\}_{x \in \mathbb{Z}_p}$, which is a uniform sample of $[0, 1]$. Thus, in the limit $p \to \infty$, we have

$$\frac{1}{p} \sum_{x \in \mathbb{Z}_p} |\cos(\omega_k x/2)| \stackrel{p \to \infty}{=} \int_0^1 |\cos(\pi x)| \mathrm{d}x = \frac{1}{\pi} \int_0^\pi |\cos(u)| \mathrm{d}u = \frac{2}{\pi}.$$

By putting these two asymptotic expressions (G.15) and (G.16) into (G.14b), we obtain that

$$\frac{\partial \ell_m}{\partial \xi_m[j]} = -\frac{p\alpha_m^\star}{\pi} \cdot \left( |\cos(\omega_k j/2 + \phi_m^\star)| - \frac{1}{p} \sum_{i=1}^{p} |\cos(\omega_k i/2 + \phi_m^\star)| \right), \qquad \forall j \in [p].$$

Next, we apply DFT with respect to $\nabla_{\xi_m} \ell_m$ in the asymptotic regime $p \to \infty$. Let $r_k \in [p]$ denote the multiplication factor in Definition D.2, i.e., $r_k k^\star = k \mod p$ for $k, k^\star \in [\frac{p-1}{2}]$. Then, we have

$$\frac{1}{2p} \sum_{j=1}^{p} |\cos(\omega_{k^\star} j/2 + \phi_m^\star)| \cdot \exp(i \cdot \omega_k j) \stackrel{p \to \infty}{=} \underbrace{\frac{(-1)^{r_k+1}}{\pi(4r_k^2 - 1)}}_{:= \varsigma_{r_k}} \cdot \exp(-2r_k \phi_m^\star \cdot i). \tag{G.17}$$

A cosine derivation of (G.17) proceeds as follows:

$$\frac{1}{p} \sum_{j=1}^{p} |\cos(\omega_{k^\star} j/2 + \phi_m^\star)| \cdot \cos(\omega_k j) \stackrel{p \to \infty}{=} \int_0^1 |\cos(\pi k^\star x + \phi_m^\star)| \cdot \cos(2r_k \pi k^\star x) \mathrm{d}x$$

$$= \frac{1}{\pi} \int_0^\pi |\cos(u)| \cdot \cos(2r_k \cdot (u - \phi_m^\star)) \mathrm{d}u = \frac{\cos(2r_k \phi_m^\star)}{\pi} \cdot \int_0^\pi |\cos(u)| \cdot \cos(2r_k u) \mathrm{d}u$$

$$= \frac{\cos(2r_k \phi_m^\star)}{\pi} \cdot \left( \int_0^{\frac{\pi}{2}} \cos((2r_k + 1)u) \mathrm{d}u + \int_0^{\frac{\pi}{2}} \cos((2r_k - 1)u) \mathrm{d}u \right) = \frac{2(-1)^{r_k+1}}{\pi(4r_k^2 - 1)} \cdot \cos(2r_k \phi_m^\star),$$

where the third equality follows from trigonometric identities, evenness of $\sin(2ru)$, and periodicity. A similar calculation applies to the sine, and combining both real and imaginary parts yields (G.17). Therefore, we have

$$\langle \nabla_{\xi_m} \ell_m, b_{2k} \rangle = 2\sqrt{2} \cdot \alpha_m^\star / \pi \cdot p^{3/2} \cdot \varsigma_{r_k} \cdot \cos(2r_k \phi_m^\star),$$

$$\langle \nabla_{\xi_m} \ell_m, b_{2k+1} \rangle = -2\sqrt{2} \cdot \alpha_m^\star / \pi \cdot p^{3/2} \cdot \varsigma_{r_k} \cdot \sin(2r_k \phi_m^\star),$$

and thus $\Delta_{\xi_m}^k / \Delta_{\xi_m}^\star = |\varsigma_{r_k}| / |\varsigma_{r_{k^\star}}| = \Theta(r_k^{-2})$. Moreover, it follows by simple calculation

$$(\mathscr{P}_{k^\star}^\| \nabla_{\xi_m} \ell_m)[j] = \langle \nabla_{\xi_m} \ell_m, b_{2k^\star} \rangle \cdot b_{2k^\star}[j] + \langle \nabla_{\xi_m} \ell_m, b_{2k^\star+1} \rangle \cdot b_{2k^\star+1}[j] \propto \cos(2k^\star j + 2\phi_m^\star),$$

for all $j \in [p]$ such that we have $\mathscr{P}_{k^\star}^\| \nabla_{\xi_m} \ell_m \propto \xi_m$.

**Step 2: Gradient of $\theta_m$.** Following (G.14a), first notice that

$$\sum_{x \in \mathbb{Z}_p} \xi_m[m_p(x,j)] \cdot \mathbb{1}(\langle e_x + e_j, \theta_m \rangle)$$

$$= \beta_m^\star \cdot \sum_{x \in \mathbb{Z}_p} \cos(\omega_{k^\star}(x+j) + \psi_m^\star) \cdot \mathbb{1}(\cos(\omega_{k^\star} x + \phi_m^\star) + \cos(\omega_{k^\star} j + \phi_m^\star) \geq 0)$$

$$\overset{p \to \infty}{=} \frac{p\beta_m^\star}{\pi} \cdot |\sin(\omega_{k^\star} j + \phi_m^\star)| \cdot \cos(\omega_{k^\star} j + \psi_m^\star - \phi_m^\star),$$

where the last equality results from the following calculation under the asymptotic regime:

$$\frac{1}{p} \sum_{x \in \mathbb{Z}_p} \cos(\omega_{k^\star}(x+j) + \psi_m^\star) \cdot \mathbb{1}(\cos(\omega_{k^\star} x + \phi_m^\star) + \cos(\omega_{k^\star} j + \phi_m^\star) \geq 0)$$

$$\overset{p \to \infty}{=} \int_0^1 \cos(2\pi x + \omega_{k^\star} j + \psi_m^\star) \cdot \mathbb{1}(\cos(2\pi x + \phi_m^\star) + \cos(\omega_{k^\star} j + \phi_m^\star)) \mathrm{d}x$$

$$= \frac{1}{2\pi} \int_{\substack{\phi_m^\star \leq u \leq \phi_m^\star + 2\pi \\ \cos(u) \geq -\cos(\omega_{k^\star} j + \phi_m^\star)}} \cos(u + \omega_{k^\star} j + \psi_m^\star - \phi_m^\star) \mathrm{d}u$$

$$= \frac{1}{2\pi} \cdot \cos(\omega_{k^\star} j + \psi_m^\star - \phi_m^\star) \cdot \int_{\substack{0 \leq u \leq 2\pi \\ \cos(u) \geq -\cos(\omega_{k^\star} j + \phi_m^\star)}} \cos(u) \mathrm{d}u$$

$$= \frac{1}{\pi} \cdot \underbrace{\sin(\arccos(-\cos(\omega_{k^\star} j + \phi_m^\star)))}_{= |\sin(\omega_{k^\star} j + \phi_m^\star)|} \cdot \cos(\omega_{k^\star} j + \psi_m^\star - \phi_m^\star).$$

By applying DFT over $\nabla_{\theta_m} \ell_m$ in the asymptotic regime $p \to \infty$, we can show that

$$\frac{1}{p} \sum_{j=1}^p |\sin(\omega_{k^\star} j + \phi_m^\star)| \cdot \cos(\omega_{k^\star} j + \psi_m^\star - \phi_m^\star) \cdot \exp(i \cdot \omega_k j)$$

$$\overset{p \to \infty}{=} -\frac{1}{\pi} \cdot \left\{ \frac{\exp(\{\psi_m^* - (r_k + 2)\phi_m^*\} \cdot i)}{r_k(r_k + 2)} + \frac{\exp(-\{\psi_m^* + (r_k - 2)\phi_m^*\} \cdot i)}{r_k(r_k - 2)} \right\} \cdot \mathbb{1}(r_k \text{ is odd}),$$

$$(\text{G.18})$$

where $r_k k^\star = k \bmod p$. The above results follow the calculation below:

$$\frac{1}{p} \sum_{j=1}^p |\sin(\omega_{k^\star} j + \phi_m^\star)| \cdot \cos(\omega_{k^\star} j + \psi_m^\star - \phi_m^\star) \cdot \cos(\omega_k j)$$

$$\overset{p \to \infty}{=} \int_0^1 |\sin(2\pi k^\star x + \phi_m^\star)| \cdot \cos(2\pi k^\star x + \psi_m^\star - \phi_m^\star) \cdot \cos(2\pi r_k k^\star x) \mathrm{d}x$$

$$= \frac{1}{2\pi} \int_{-\phi_m^\star}^{2\pi - \phi_m^\star} |\sin(u)| \cdot \cos(u + \psi_m^\star - 2\phi_m^\star) \cdot \cos(r_k(u - \phi_m^\star)) \mathrm{d}u$$

$$= \frac{1}{4\pi} \int_0^{2\pi} |\sin(u)| \cdot \cos((r_k + 1)u + \psi_m^\star - (r_k + 2)\phi_m^\star) \mathrm{d}u$$

$$+ \frac{1}{4\pi} \int_0^{2\pi} |\sin(u)| \cdot \cos((r_k - 1)u - \psi_m^\star - (r_k - 2)\phi_m^\star) \mathrm{d}u, \qquad (\text{G.19})$$

where for $h_1 = r_k \pm 1$ and $h_2 = \psi_m^\star - (r_k + 2)\phi_m^\star / - \psi_m^\star - (r_k - 2)\phi_m^\star$, we can further show show

$$\int_0^{2\pi} |\sin(u)| \cdot \cos(h_1 u + h_2) \mathrm{d}u = \cos(h_2) \cdot \int_0^{2\pi} |\sin(u)| \cdot \cos(h_1 u) \mathrm{d}u$$

$$= (1 + (-1)^{h_1}) \cdot \cos(h_2) \cdot \int_0^\pi \sin(u) \cos(h_1 u) \mathrm{d}u = \frac{4}{1 - h_1^2} \cdot \cos(h_2) \cdot \mathbb{1}(h_1 \text{ is even}).$$

$$(\text{G.20})$$

By combining (G.19) and (G.20), and performing a similar calculation for the sine component, we obtain the result in (G.18) This implies that for even $r_k$, we have

$$\langle \nabla_{\theta_m} \ell_m, b_{2k} \rangle = -\sqrt{2} \beta_m^\star / \pi \cdot p^{3/2} \cdot \left\{ \frac{\cos(\psi_m^* - (r_k + 2)\phi_m^*)}{r_k(r_k + 2)} + \frac{\cos(\psi_m^* + (r_k - 2)\phi_m^*)}{r_k(r_k - 2)} \right\},$$

$$\langle \nabla_{\theta_m} \ell_m, b_{2k+1} \rangle = -\sqrt{2} \beta_m^\star / \pi \cdot p^{3/2} \cdot \left\{ \frac{\sin(\psi_m^* - (r_k + 2)\phi_m^*)}{r_k(r_k + 2)} - \frac{\sin(\psi_m^* + (r_k - 2)\phi_m^*)}{r_k(r_k - 2)} \right\},$$

Hence, $\Delta^k(\theta_m)/\Delta^\star(\theta_m) = \Theta(r_k^{-2}) \cdot \mathbb{1}(r_k \text{ is even})$ and for all $j \in [p]$

$$(\mathscr{P}_{k^\star}^\parallel \nabla_{\theta_m} \ell_m)[j] = \langle \nabla_{\theta_m} \ell_m, b_{2k^\star} \rangle \cdot b_{2k^\star}[j] + \langle \nabla_{\theta_m} \ell_m, b_{2k^\star + 1} \rangle \cdot b_{2k^\star + 1}[j] \propto \cos(w_{k^\star} j + \phi_m^\star),$$

which gives that $\mathscr{P}_{k^\star}^\parallel \nabla_{\theta_m} \ell_m \propto \theta_m$ and completes the proof. $\qquad \square$

# H STATEMENTS

## H.1 THE USE OF LARGE LANGUAGE MODELS

We acknowledge the use of a large language model (LLM) primarily to improve the grammar and clarity of this manuscript. The LLM was also used to assist with debugging and generating boiler-plate code snippets, which were reviewed and validated by the authors.

## H.2 REPRODUCIBILITY STATEMENT

To ensure the reproducibility of our findings, we have made comprehensive efforts. The synthetic nature of the modular addition dataset, as detailed in Section 2 (Preliminaries), allows for exact replication of our experimental data. Our training methodology, including the use of PyTorch's default initialization and the Adam optimizer with specified hyperparameters, is thoroughly described in Section 3 (Empirical Findings). For full transparency and ease of reproduction, all code, including model implementations and training scripts, will be publicly released on GitHub upon publication. Furthermore, an interactive demo showcasing our results will be made available on a dedicated website.

# I ADDITIONAL EXPERIMENTAL RESULTS FOR DIFFERENT MODULO $p$

In this section, we replicate the key experimental findings presented in §3, but using an increased modulus $p = 47$ and a network width of $M = 1024$.

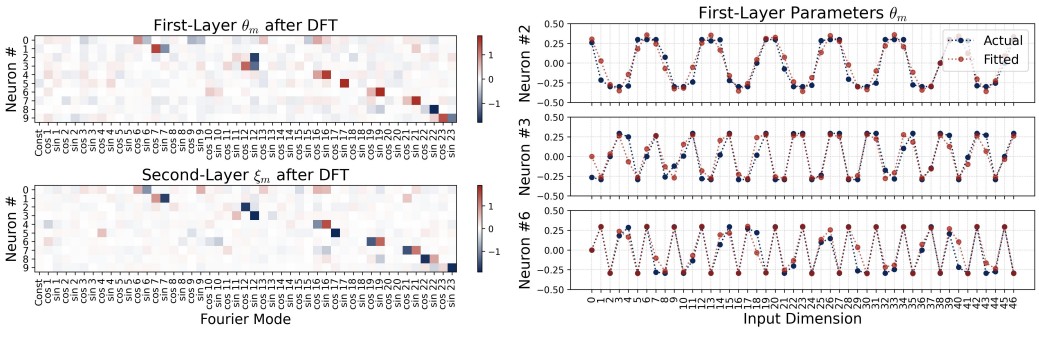

(a) Heatmap of Learned Parameters after DFT.     (b) Actual Learned and Fitted Parameters.

Figure 15: Learned parameters under the full random initialization with $p = 47$ and ReLU activation using AdamW. Figure (a) plots a heatmap of the learned parameters for the top 10 neurons after Discrete Fourier Transform (DFT, see §A.3). Each row in the heatmap corresponds to the Fourier components of a single neuron's parameters. The plot clearly reveals a single-frequency pattern: each neuron exhibits a large, non-zero value focused on only one specific frequency component, confirming a highly sparse and specialized frequency encoding. Figure (b) further examines the periodicity by plotting line plots of the learned parameters for three neurons, each overlaid with a trigonometric curve fitted via DFT. The fitted curve aligns almost perfectly with the actual one.

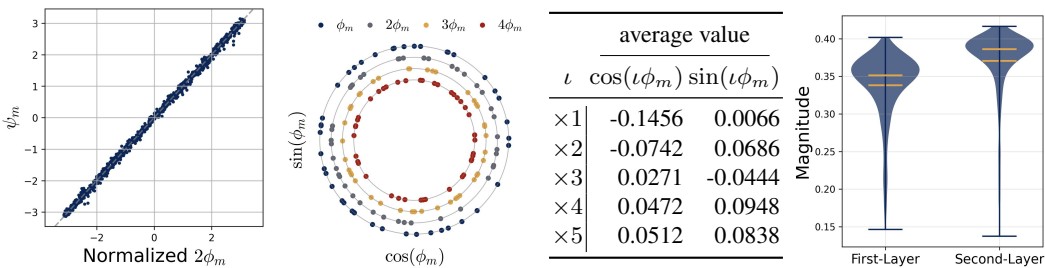

(a) Scatter of $(2\phi_m, \psi_m)$. (b) Phase Symmetry within Frequency Group $\mathcal{N}_k$. (c) Distribution of $\alpha_m, \beta_m$.

Figure 16: Visualizations of learned phases with $p = 47$ and $M = 1024$ neurons. Figure (a) plots the relationship between the normalized $2\phi_m$ and $\psi_m$, with all points lying around $y = x$. Figure (b) shows the uniformity of the learned phases within $\mathcal{N}_k$. The right panel quantifies this symmetry by computing the averages of $\cos(\iota\phi_m)$ and $\sin(\iota\phi_m)$, all of which are close to zero. Figure (c) presents violin plots of the magnitudes $\alpha_m$ and $\beta_m$, suggesting that the neurons learn nearly identical magnitudes.