# OpenReview forum: "On the Mechanism and Dynamics of Modular Addition: Fourier Features, Lottery Ticket, and Grokking"
_ICLR.cc/2026/Conference — Submitted to ICLR 2026_

### Official Review · Reviewer_W5pE · 2025-10-28

**Soundness:** 4
**Presentation:** 4
**Contribution:** 3
**Rating:** 8
**Confidence:** 3

**Summary:**

This paper provides a comprehensive mechanistic and theoretical explanation of how two-layer neural networks learn to perform modular addition and how this process explains grokking.
Using both empirical analysis and formal dynamical proofs, the authors show that during training, each neuron converges to a single-frequency Fourier feature of the form

$$
\theta_m[j] = \alpha_m \cos(\omega_{\phi(m)} j + \phi_m), \quad
\xi_m[j] = \beta_m \cos(\omega_{\phi(m)} j + \psi_m),
$$

where the output phase satisfies a phase-alignment relation

$$
\psi_m \approx 2\phi_m.
$$

Across neurons, phases become uniformly distributed within each frequency group, forming a phase-symmetric ensemble.
The network’s collective behavior can then be interpreted as a majority-voting Fourier circuit that robustly implements the indicator function

$$
1[(x+y)\bmod p = j].
$$

On the dynamics side, the authors identify a “lottery ticket” mechanism in Fourier space:
different frequencies compete within each neuron, and the one with the largest initial magnitude and smallest phase misalignment wins.
This explains the emergence of single-frequency neurons and provides a predictive theory of feature selection.

Finally, the paper analyzes grokking as a three-stage process:
1. Memorization phase dominated by loss minimization
2. First generalization phase where weight decay sparsifies frequencies and sharpens alignment
3. Second generalization phase where weight decay refines the clean Fourier solution

Together, these results offer a unified, end-to-end account of how gradient descent discovers structured Fourier representations and transitions from memorization to generalization.

**Strengths:**

- Comprehensive mechanistic theory.
  This is the most complete explanation so far of modular-addition learning and grokking in shallow networks, connecting empirical phenomena (phase alignment, sparsification) to provable training dynamics.

- Elegant empirical–theoretical correspondence.
  Observations 1–6 are each mirrored by formal results (Proposition 4.2, Theorem 5.2, 5.3). The analytical use of quadratic activations is a well-justified simplification that retains the core dynamics observed with ReLU.

- Novel “Fourier lottery ticket” insight.
  The finding that feature emergence is governed by initial magnitude and phase misalignment provides a simple, predictive explanation of why single-frequency neurons reliably appear.

- Interpretability of grokking.
  The proposed three-stage timeline, validated with metrics like phase alignment and frequency sparsity, makes grokking a measurable and interpretable process.

- Clarity and rigor.
  The exposition is unusually clear for such a technical topic, combining intuitive figures with formal statements.

**Weaknesses:**

- Restricted architectural scope.
  All analyses use two-layer MLPs with one-hot or learned embeddings; while ideal for interpretability, it remains unclear whether the same Fourier-alignment and frequency-competition mechanisms appear in deeper or attention-based models.

- Empirical validation of the voting mechanism.
  Proposition 4.2 predicts that uniform phase diversity cancels noise via majority voting; an ablation that breaks phase uniformity could directly confirm this mechanism.

- Connection to pretrained LLMs not explored.
  The paper convincingly explains why Fourier features emerge in modular tasks, but stops short of relating this to real LLMs—where similar sinusoidal and Fourier-like number encodings have already been observed.

**Questions:**

1. Connection to Fourier features in large models.
   Recent work (arXiv:2502.09741) shows that when numbers are initialized with Fourier features, large pretrained LLMs can learn addition almost instantly, and that existing LLM embeddings already exhibit Fourier structure.
   Can the authors interpret this observation through their phase-alignment dynamics—i.e., are Transformers implicitly performing the same “frequency lottery” at scale?

2. Scaling of grokking time.
   Theoretical results (Theorem 5.3) relate convergence time to initialization scale
   $
   \kappa_{\text{init}}
   $
   and modulus
   $
   p.
   $
   Can this be turned into a quantitative scaling law predicting grokking delay?
   Moreover, can this framework explain the observation from arXiv:2502.09741 that when the magnitude is large for some Fourier component, the model skips the grokking phase and learns addition with less data?

3. Quantifying required neurons.
   Can the authors estimate the minimum number of neurons
   $
   M
   $
   required to achieve
   $
   100\\%
   $
   test accuracy as a function of
   $
   p?
   $


4. Beyond addition.
   Would similar Fourier competition and alignment appear in modular multiplication or other? Extending analysis to those tasks could generalize the theory beyond addition.

---

> ### Comment · Reviewer_W5pE · 2025-11-24
>
> After reading the other reviewers’ comments and going through the related work again, I realized that several of the claims in the paper are actually not novel, even though they’re presented as new “observations.” The authors say Obs.1 and Obs.2 are included for completeness, but other observations also already appeared in earlier papers:
>
> Grokking Modular Polynomials (arXiv:2406.03495) shows that uniformly random phases are already enough for modular addition.
> Razeghi et al. (arXiv:2301.02679) discuss that certain phase patterns cancel noise better than uniform random. This matches what the paper says but isn’t cited or compared. The “lottery ticket” style behavior in Obs.6 is basically the same as what was described in Bridging Lottery Ticket and Grokking and Understanding Grokking from Inner Structure of Networks. These aren’t mentioned at all.
> Because these earlier works already reported most of the “observations,” the empirical novelty is a lot smaller than what the paper suggests. Without the proper citations, it’s honestly kind of misleading for readers. So unless the authors clearly explain what is actually new compared to these previous works, I will have to lower my score.

---

> > ### Author Response · Authors · 2025-11-25
> >
> > Thanks for pointing out the concerns. We address them as below.
> >
> > > "Several of the claims in the paper are actually not novel, even though they’re presented as new 'observations.'"
> >
> > Section 2.1 in (Grokking modular polynomials https://arxiv.org/pdf/2406.03495) **provides a construction** of a two-layer neural network that solves modular addition, where the weights are sinusoidal and the **phases are i.i.d. and uniform**. But this is **merely a construction result** -- it neither guarantees that **gradient training will find this specific solution** nor establishes the **necessary conditions** for such uniformity to actually occur. To narrow this gap, [1] empirically argue the learned network is similar to the constructed one by looking into the loss and prediction accuracy. **But there is no comparison of neural network parameters that substantiate this claim**.
> >
> > In contrast, our work empirically show that, after gradient-based training, symmetric phases (uniformly distributed phases) appear (Observation 2). Motivated by this observation, with **finite network width**, we establish a **much weaker technical condition** than uniformity, i.e., Definition 4.1, finite sum $\sum_{m} \exp(i \cdot \iota \phi_m) = 0$ for harmonics $\iota \in \{2, 4\}$, showing that as long as the $M$ phases are sufficiently diverse, **gradient-based training provably learns a two-layer neural network that solves modular addition** and the noise can be cancelled out.
> >
> > Besides, we prove that gradient-based training **provably learns** a desired network, which uses phase symmetry (cancellation) to build an aggregated predictor that is close to an indicator (modular Kronecker Delta). Moreover, such an indicator approximation argument is based on **both frequency and phase diversification**, whose proof is different from that in [1]. See Proposition 4.2 for details. In comparison, as far as we know, previous results stay in writing the predictor as a **sum of cosine functions**, making the essence of neuron cancellation unknown.
> >
> > > "The “lottery ticket” style behavior in Obs.6 is basically the same as what was described in Bridging Lottery Ticket and Grokking and Understanding Grokking from Inner Structure of Networks. These aren’t mentioned at all."
> >
> > We borrow the well-known "lottery ticket" simply to describe this **initialization-dependent outcome**. To see this similarity, we note that lottery ticket hypothesis states that a dense neural network, after training from random initialization, contains a subnetwork that is as good as as the original network. This is similar to our case, but our lottery ticket is **in terms of the frequencies**. The weight of a given neuron, in initialization, has components in all frequencies. But after training, only one frequency contribute to the final predictor, and this frequency can be viewed as the "winning ticket". The relationship between the learned subnetwork and random full network is analogous to our case, where we have the learned frequencies versus the full spectrum.
> > We would like to highlight that our theory proves that **the winning frequences can be deterministically predicted by the initalization**, which is a novel result.
> >
> >
> > > "Without the proper citations, it’s honestly kind of misleading for readers."
> >
> > Thanks for pointing out the missing related work and we added them in the revised version.
> >
> > Please refer to our response to Reviewer `Ykz3`  for a more detailed elaboration of our novelty.

---

> ### Author Response · Authors · 2025-11-25
>
> We sincerely thank Reviewer W5pE for their exceptionally detailed, insightful, and positive review. Our response is as below.
>
> > Restricted architectural scope. All analyses use two-layer MLPs with one-hot or learned embeddings; while ideal for interpretability, it remains unclear whether the same Fourier-alignment and frequency-competition mechanisms appear in deeper or attention-based models.
>
> We acknowledge the limitation of restricting our analysis to two-layer MLPs. We chose this simplest architecture precisely because it allowed for the **clearest interpretability** of the core mechanisms without the confounding complexity of deeper or attention-based models. While this establishes a foundational understanding, we agree that extending this study to more general DNNs is an important direction for future work to confirm the broader applicability of these dynamics.
>
> > Empirical validation of the voting mechanism. Proposition 4.2 predicts that uniform phase diversity cancels noise via majority voting; an ablation that breaks phase uniformity could directly confirm this mechanism.
>
> We thank the reviewer for the constructive suggestion. We have added ablation study for the majority mechanism, especially the necessity of both **frequency and phase diversification**,  and show that such fully diversified parametrization is most efficient, i.e., achieve smallest loss under neuron and scale constraint.
>
> > Connection to Fourier features in large models. Recent work (arXiv:2502.09741) shows that when numbers are initialized with Fourier features, large pretrained LLMs can learn addition almost instantly, and that existing LLM embeddings already exhibit Fourier structure.
>
> Your question about recent LLM work (arXiv:2502.09741) is highly relevant. Our conjecture is two-fold:
> - **Training Data Curriculum:** We hypothesize that during large-scale LLM training, the model quickly learns **induction-head** curriculum where results in the formation smaller curriculum for simpler tasks, like modular algebra. It abstracts the Next Token Prediction (NTP) task into a specific modular algebra problem (e.g., $a+b \pmod P$).
> - **Implicit Feature Learning:** From the training dynamics perspective presented in our paper, the **phase-alignment dynamics** is the key mechanism for learning addition. The embedding of numbers ($\mathbf{x}_i$) will be naturally trained to form the necessary trigonometric patterns because these features are the most **data-efficient** way to implement addition using a simple two-layer transformation. We suggest that Transformers are implicitly performing a similar **"frequency lottery"** at scale, where the most advantageous Fourier components are quickly selected and reinforced within the embedding layers and low-rank feed-forward networks (FFNs) because they correspond to the ground-truth solution's structure.
>
> > Scaling of grokking time.
>
> Unfortunately, our current theory, which is based on the **full dataset setup**, cannot quantitatively predict the exact grokking delay observed with train-test splits, as the dynamics of generalization with limited data are fundamentally different and more complex. However, our theory _can_ partially explain the observation that **large initial Fourier component magnitudes skip the grokking phase** (arXiv:2502.09741):
> - **Lottery Ticket Mechanism:** As detailed in **Appendix C.1 (Theoretical Underpinning of Lottery Ticket Mechanism)**, a frequency component initialized with a significantly larger scale ($\alpha_m$) grows much faster and quickly dominates the neuron.
> - **Generalization Promotion:** This fast growth results in the neuron learning a **cleaner, well-defined trigonometric solution** much quicker than its peers. Since a clean trigonometric solution is **inherently generalizable**, this neuron promotes the entire network's dynamics toward generalization, effectively **skipping the slow memorization stage** observed in grokking.
>
> > Quantifying required neurons.
>
> Based on the minimal requirement of  structure of the learned solution (Definition 4.1), the minimum number of neurons required per frequency $k$ to satisfy the symmetry conditions $\sum_{m \in N_k} \exp(i \cdot \iota \phi_m) = 0$ for $\iota=2$ and $\iota=4$. This symmetry is necessary to cancel noise and achieve the robust **Majority-Voting** scheme (Proposition 4.2). Since we need $2\phi_m$'s and $4\phi_m$'s to achieve perfect symmetry. Therefore, the minimum number of neurons needed for a fully diversified, phase-symmetric solution is at least:  $4\times \frac{P-1}{2} = 2(P-1)$.

---

> > ### Author Response · Authors · 2025-11-25
> >
> > > Beyond addition. Would similar Fourier competition and alignment appear in modular multiplication or other? Extending analysis to those tasks could generalize the theory beyond addition.
> >
> > We believe our theoretical analysis can be **adapted to develop theory for general data that satisfies the Abelian group** structure considered in related work [1]. The underlying principle—that addition is a superposition of sinusoids—stems from the **Fourier basis** of the cyclic group $(\mathbb{Z}/P\mathbb{Z}, +)$. Modular multiplication operates on a different, more complex group structure, which would require an analysis based on its corresponding **group characters**. We provide initial discussion about this relationship in **Appendix B** and agree that extending the analysis to other operations would be a fruitful and essential direction for future work.
> >
> > [1] Tian, Yuandong. "Composing Global Solutions to Reasoning Tasks via Algebraic Objects in Neural Nets." _The Thirty-ninth Annual Conference on Neural Information Processing Systems_.

---

### Official Review · Reviewer_vTtu · 2025-10-30

**Soundness:** 2
**Presentation:** 2
**Contribution:** 2
**Rating:** 2
**Confidence:** 3

**Summary:**

This paper analyses the phenomenon of delayed generalisation (grokking) on the modulo arithmetic task mod(a+b)23. The work corroborates other findings that, during the grokking process, the two-layer MLP learns Fourier transforms to complete the solution. The paper suggests that grokking occurs in three distinct phases: Memorisation, where the model learns 'common data', then Generalisation Phase 1, where the model begins to minimise loss on 'rare examples' and then finally Generalisation Phase 2, where increased generalisation is governed only by the weight decay term.  Through theoretical and empirical analysis, the authors argue that it is possible to predict the final frequency domain of neurons based on their initial parameterisation via an analysis of Fourier components aligning with their observation of the Lottery Ticket Mechanism. The authors also provide a mechanistic insight into the trained model and identify a majority voting mechanism that cancels out noise and facilitates generalisation. Finally, there is an analysis of how gradient-based training facilitates the representation of features from a training dynamics perspective.

**Strengths:**

1. The paper adds interesting observations about the grokking phenomena in the context of the mod(a+b)23 task, which provide a nice insight into grokking on this dataset.
2. The combination of extensive theoretical results and supporting empirical results strengthens the paper's findings; however, some of the observations provided by the paper are corroborations of previous findings rather than novel insights (see weaknesses below).
3. The notion of the majority voting scheme is an interesting insight; it would be nice to see if, in other grokking tasks, such a dynamic is used to improve generalisation.
4. Predicting the final frequency of a neuron from its initialisation via magnitude and phase misalignment is a neat finding; it would be good to see these findings extended to other modulo arithmetic tasks to demonstrate their generality.

**Weaknesses:**

1. **Lottery Ticket Mechanism**:  In the paper, observation 6 is positioned as a novel observation; however, prior work, namely [1], [2] explicitly mentions the role of internal structure at initialisation, via the Lottery Ticket Hypothesis (LTH) [3], being a primary factor in grokking. Furthermore, [2] even goes on to show that particular 'grokking tickets' reduce the time for generalisation to occur. I think that the 'Lottery Ticket Mechanism' you observe should be positioned as corroborating other findings in the literature, rather than being a novel insight of this paper's analysis.

2. **Fourier Features** Could the authors describe why/if they believe this perspective to be novel? Given that previous literature describing the dynamics of grokking via mechanistic interpretability [1] has shown that neural networks leverage discrete Fourier transforms and trigonometric identities to perform the addition necessary in modular arithmetic tasks.

3. **Narrowness of analysis on mod(a+b) 23 Task**: The grokking task that is analysed in this paper is somewhat non-standard compared to other grokking studies [1] [2], for example, the original paper [4]  that introduces grokking conducts experiments on the mod(a+b)97 task. Is there a rationale behind examining mod(a+b)23? In the mod(a+b)23 case, as shown by Figure 3, there is a slight delay in generalisation; however, this is not as extreme as in the mod(a+b)97 case. Can you show that part of your analysis holds in other modular arithmetic tasks, or are these findings limited to this particular dataset?

4. **Importance of Weight Decay in Grokking**: The three phases of grokking suggested in this paper (Memorisation, Generalisation 1 and Generalisation 2) place a large weighting on the weight decay term in enabling generalisation. While [5] does support this narrative, more recent literature [6] has shown that weight decay is not a causal factor in generalising on algorithmic tasks, as they can mitigate or induce grokking entirely without modifying weight decay.  Additionally, [7] has also shown that grokking can be mitigated and that this mitigation is not solely reliant on the weight decay factor.  In light of these existing results, do you feel that you phases of grokking adequately describe the dynamics of grokking, or that they rely on factors that may correlate with generalisation, but are not causal factors for it?

5. **Common vs Rare Training Examples**: The statements regarding common vs rare training examples resemble conjecture; the idea of common examples where 'symmetric pairs' are memorised is not fully quantified in the paper, nor are 'rare' examples properly explained. To empirically justify statements about 'common' and 'rare' training examples, the authors should conduct an ablation study where the model is trained only on so-called 'rare' examples. This should, under the arguments in the paper, eliminate the memorisation phase and reduce the time to generalise on the test data.

6. **Even-order polynomials and Activation Swapping**: In Appendix Table 1, the activation swapping is shown replacing the RelU function with polynomials, preserving accuracy when the exponent is even and losing accuracy when the exponent is odd. However, there is no even representation of even and odd exponents in this table. Can you add the results for the exponents 5, 7 and 9 to represent odd exponents fully?

7. **Prediction of final frequency**: The paper makes the bold statement that a neuron's final frequency can be predicted entirely from its initial magnitude and frequency alignment. Can the authors please provide an empirical analysis of how many neuron frequencies at the end of training are correctly predicted from this evaluation?

8x. **Lack of clear takeaways**: The paper offers many observations of grokking dynamics; however, it is unclear what the general takeaways of the work should be, given that some of the observations are not entirely novel. The authors should explicitly highlight how their work provides new understandings of grokking that can generalise outside of their specific experimental setup. The paper would benefit from focusing less on the quantity of observations and instead on the clarity of insights. Could the authors please provide a conclusion section to this effect?

References:
[1] Nanda, N., Chan, L., Lieberum, T., Smith, J. and Steinhardt, J., 2023. Progress measures for grokking via mechanistic interpretability. arXiv preprint arXiv:2301.05217.

[2] Furuta, H., Minegishi, G., Iwasawa, Y. and Matsuo, Y., 2024. Towards empirical interpretation of internal circuits and properties in grokked transformers on modular polynomials. Transactions on Machine Learning Research.https://openreview.net/forum?id=MzSf70uXJO.

[3] Frankle, J. and Carbin, M., 2018. The lottery ticket hypothesis: Finding sparse, trainable neural networks. arXiv preprint arXiv:1803.03635.

[4] Power, A., Burda, Y., Edwards, H., Babuschkin, I. and Misra, V., 2022. Grokking: Generalisation beyond overfitting on small algorithmic datasets. arXiv preprint arXiv:2201.02177.

[5] Liu, Z., Michaud, E.J. and Tegmark, M., 2022. Omnigrok: Grokking beyond algorithmic data. arXiv preprint arXiv:2210.01117.

[6] Kumar, T., Bordelon, B., Gershman, S.J. and Pehlevan, C., 2023, September. Grokking as the transition from lazy to rich training dynamics. In The twelfth international conference on learning representations.

[7] Mason-Williams, G. and Mason-Williams, I., Decomposed Learning: An Avenue for Mitigating Grokking. In ICML 2025 Workshop on Methods and Opportunities at Small Scale.

**Questions:**

See weaknesses above.

---

> ### Author Response · Authors · 2025-11-25
>
> We thank Reviewer vTtu for the comments and suggestions.
> We address the weaknesses and questions raised below, clarifying the novelty and scope of our contributions.
>
> > Lottery Ticket Mechanism: In the paper, observation 6 is positioned as a novel observation...
>
> We borrow the well-known "lottery ticket" simply to describe this **initialization-dependent outcome**. To see this similarity, we note that lottery ticket hypothesis states that a dense neural network, after training from random initialization, contains a subnetwork that is as good as as the original network. This is similar to our case, but our lottery ticket is **in terms of the frequencies**. The weight of a given neuron, in initialization, has components in all frequencies. But after training, only one frequency contribute to the final predictor, and this frequency can be viewed as the "winning ticket". The relationship between the learned subnetwork and random full network is analogous to our case, where we have the learned frequencies versus the full spectrum.
> We would like to highlight that our theory proves that **the winning frequences can be deterministically predicted by the initalization**, which is a novel result.
>
> > Fourier Features Could the authors describe why/if they believe this perspective to be novel?
>
> We admit that the emergence of trigonometric patterns has been previously proposed in the literature like [1]. Our paper offers a more comprehensive and unified explanation of the entire process. Our specific novel contributions include:
> * A **neuron-wise, closed-form characterization** of learning dynamics for this architecture, distinct from the mean-field (infinite-neuron) analyses found in prior work (e.g., Wang & Wang, 2025), where taking the limit of infinite neurons simplifies the analysis of majority voting.
> * A rigorous mechanistic explanation of **_how_** the model converges to this full-frequency solution via ODE dynamics in the frequency domain. Our analysis reveals that different frequencies within a single neuron **compete** during training, ultimately leading each neuron to specialize in a single frequency.
> * A detailed elucidation of the "lottery ticket" mechanism, demonstrating how random initialization determines **_which_** specific frequency survives in each neuron to dominate the final representation. We validate this empirically in Figure 2, showing how initial phase mismatch and magnitude influence growth dynamics and final frequency selection.
>
> >  Narrowness of analysis on mod(a+b) 23 Task: The grokking task that is analysed in this paper is somewhat non-standard compared to other grokking studies [1] [2],
>
> Our empirical results hold true **for all prime numbers** $p$; we simply use $p=23$  as a representative example. Due to space constraints, we did not present results for other  values. Moreover, our theory (Section 4) holds for any prime $p$ and our theory supports the empirical observations.  This is strongly supported by our theoretical results (Section 4), which rigorously demonstrate that the derived feature structure is independent of the modulus number $p$, thus we believe it is unnecessary to extensively experiment with various values of $p$. Also, we add  experiments for $p=97$ in Appendix
>
> > Importance of Weight Decay in Grokking: The three phases of grokking suggested in this paper (Memorisation, Generalisation 1 and Generalisation 2) place a large weighting on the weight decay term in enabling generalisation.
>
> We acknowledge that the literature shows other factors can mitigate or induce grokking . While other factors exist, we clearly  **demystify the role played by weight decay** within _our_ specific three-stage timeline (Section 3.2). We show that its  role is to **prune unnecessary frequency components**, effectively **sparsifying the solution**. This **causally leads** to the fact that the final model learns the clean, generalizable Fourier solution instead of the noisy, overfitted solution learned in the initial memorization stage. Weight decay forces the model to select the minimal, most generalizable features. This result complements those in the existing literature.
>
> > Common vs Rare Training Examples: The statements regarding common vs rare training examples resemble conjecture
>
> We provide a more detailed analysis and visualization of this argument in **Appendix D.1** (Figure 11). The distinction is empirical: **Common samples** are those where both the pair $(i, j)$ and its symmetric pair $(j, i)$ are present in the training data, leading to quick initial memorization. **Rare samples** are those where only one unique pair, say $(i, j)$, is present, and the generalization to $(j, i)$ is delayed. We visualize how rare data initially achieves near 0 accuracy while common data achieve perfect accuracy at early memorization stage in Figure 11.

---

> > ### Author Response · Authors · 2025-11-25
> >
> > > Even-order polynomials and Activation Swapping: In Appendix Table 1, the activation swapping is shown replacing the RelU function with polynomials, preserving accuracy when the exponent is even and losing accuracy when the exponent is odd.
> >
> > We have already included the results for odd exponents ($x$ and $x^3$ with $x^5$ and $x^7$ newly added) in **Appendix Table 1** and showed that the resulting test accuracy is very low (close to 0), demonstrating that **odd exponents break the mechanism**.
> >
> > > Prediction of final frequency: The paper makes the bold statement that a neuron's final frequency can be predicted entirely from its initial magnitude and frequency alignment.
> >
> > As shown in "Lottery Ticket Mechanism" (Observation 6), we can **deterministically** predict **what frequency will be learned by a specific neuron** based solely on its individual initial magnitude and phase misalignment. This is also empirically validated in Figure 2.
> >
> >
> > > Lack of clear takeaways: The paper offers many observations of grokking dynamics; however, it is unclear what the general takeaways of the work should be, given that some of the observations are not entirely novel.
> >
> > We acknowledge that the paper is technically dense and contains many observations and theoretical results.
> > These results **collectively provide a complete, end-to-end understanding** of the mechanistic interpretation and training dynamics of how a two-layer NN learns modular addition. We have added a temporary conclusion section in the appendix and **will move it to the main body** if the paper is accepted. We will also include a roadmap in the appendix guiding the readers to the results and their logical relationship in the revision.
> >
> > [1] Nanda, N., Chan, L., Lieberum, T., Smith, J. and Steinhardt, J., 2023. Progress measures for grokking via mechanistic interpretability. arXiv preprint arXiv:2301.05217.

---

### Official Review · Reviewer_xW8s · 2025-10-30

**Soundness:** 1
**Presentation:** 1
**Contribution:** 1
**Rating:** 0
**Confidence:** 3

**Summary:**

The paper explores a one-hot encoded modular addition task $(a+b) mod 23$ with a 2-layer fully connected neural network with a hidden size of 512. With this network, they explore 3 questions, encompassing mechanistic interpretability, training dynamics, and grokking. They explore these questions from the parameters perspective, empirically finding that a neuron's parameters form a trigonometric pattern, and identify a set of properties: phase-alignment, model symmetry, majority-voting scheme, and a lottery ticket mechanism.

**Strengths:**

The paper explores an interesting set of questions.

Highlights potentially interesting findings.

Work around activation functions is intriguing.

**Weaknesses:**

The paper introduces the conceptions `phase alignment, where a neuron’s output phase is twice its input phase, and phase symmetry, where phases are uniformly distributed among neurons sharing the same frequency.` however, before this introduction, the term `phase` is not concretely defined in this context, which makes this section hard to parse.

The paper states on line `136` that `We begin with the most striking observation: a global trigonometric pattern in parameters that consistently emerges across all training runs with random initialization.` however, no empirical evidence is provided to support this claim.

On line `148` it states `In Figure 7b, we zoom in on the learned parameters of the first five neurons`; however Figure 7b looks at 3 Neurons; how are these neurons selected? In addition, how many neurons have this pattern? How often was this pattern observed over multiple runs? Line `150-151` goes on to state `The plots show that these parameters are well approximated by cosine curves, shifted by phases φm, ψm, and scaled by magnitudes αm, βm.` however, only 1.9% of the total neurons are represented in Figure 7a, and only 0.5859375% for Figure 7b. Although these neurons may be represented this way, how are the other neurons represented? This is especially important to show when the following text reads `this suggests that the trained neural network learns to solve modular addition by embedding a trigonometric structure into its parameters`. The paper then uses this finding to build the rest of the paper. In addition, I was unable to reproduce the main findings in the paper, i.e., that **all** the networks' input and output parameters formed a `trigonometric structure` and found neurons that did not form a `trigonometric structure` which suggests the `trigonometric structure` is not a requirement for the model to learn this task.

The paper only explores $(a+b) mod 23$, which limits the generality of the findings. Additional modular tasks should be explored to further support the findings, such as subtraction, division, and multiplication. Given that the paper explores modular arithmetic in a non-standard setup, where the input data is one-hot encoded. It is unclear how the findings will generalize to larger networks, more practical problems, or networks that use embedding layers.

The paper ends abruptly with no concluding section and does not provide clear takeaways nor relates the findings back to the current literature.

**Questions:**

Please see weakness and more concretely:

Why is only p=23 explored in a network with a hidden width of 512 explored? Do these results hold when using a range of $p$ values, i.e $29, 31, 37, 41,.., 83, 89, 97$ and hidden widths of $32, 64, 128, 256$?

Can you explore  additional modular arithmetic tasks such as subtraction, division, and multiplication? This would help improve the generality of the findings. To also help substantiate claims around phase alignment,  model symmetry, and the lottery ticket mechanism can this be explored in the case of CIFAR 10 and CIFAR 100 [1].

A lot of the work is then based on the finding that the problem is solved by `embedding a trigonometric structure into its parameters`. Given the model is significantly overparameterized, can you explore and report what happens when the first (input) layer is frozen (completely random) during training (with all data)?

[1] Krizhevsky, A. and Hinton, G., 2009. Learning multiple layers of features from tiny images.

---

> ### Author Response · Authors · 2025-11-25
>
> We thank Reviewer xW8s for the comments and suggestions. But we would like to point out that some of the comments based on **fundamental misunderstandings** of our experimental setup and are verifiably false.
>
>
> ### Response to Weaknesses
>
>
> > The paper introduces the conceptions "phase alignment, where a neuron’s output phase is twice its input phase", and "phase symmetry, where phases are uniformly distributed among neurons sharing the same frequency." However, before this introduction, the term "phase" is not concretely defined in this context, which makes this section hard to parse.
>
> We note that the mathematical definitions of "phase" are in Equation (3.1) and the paragraph after that. Then, after introducing the phases $\phi_m$ and $\psi_m$, we introduce the notion of _phase alignment_ and _phase symmetry_ in Observations 2 and 3 respectively. Admittedly, we state these observations in Introduction without mathematical details in Introduction, the goal is to introduce the main results, and thus they inevitablly appear. And it is an academic  convention that introduction is rather nontechnical.
>
>
>
> > The paper states on line 136 that "We begin with the most striking observation: a global trigonometric pattern in parameters that consistently emerges across all training runs with random initialization." however, no empirical evidence is provided to support this claim.
>
> We note that the first two observations – _trigonometric parameterization_ and  _phase alignment_ – have been previously explored in the literature (Gromov, 2023; Nanda et al.,
> 2023; Yip et al., 2024). We include them for completeness. See Lines 132 and 133. These observations have been validated empirically in privous works, and more importantly, in our work. In Figure 7 (b), we show that the neurons implements sinusoidal functions. In Figure 8 (a), we show that the Fourier coiefficients of each weight $\theta_m$ and $\xi_m$ only have a single frequency component. This proves that Observation 1 holds. We defer these results to the appendix because this observation is known in the literature and it is not our contribution, although it serves as the foundation for all the results.
>
> As for Observation 2, immediately after its statement, we show the scatter plots of $(2\phi_m, \psi_m)$ for $m \in [512]$. This is a figure generated by our empirical experiment. This figures shows that $\{(2\phi_m, \psi_m)\}_{m\in[512]}$ lies on the line $y=x$, validating Observation 2.
>
> As we mentioned above, the fact that the neuron weights have trignometric pattern is a **foundational insight** gained by existing works on training neural networks for modular arithmetics. Seen from the comment we just responded to and the subsequent comments, we are **worried that the reviewer is unfamiliar with the literature and make uninformed assessments**. The claim that "no empirical evidence is provided to support this claim" is **factually incorrect**.

---

> > ### Author Response · Authors · 2025-11-25
> >
> > > On line 148 it states "In Figure 7b, we zoom in on the learned parameters of the first five neurons"; however Figure 7b looks at 3 Neurons; how are these neurons selected? In addition, how many neurons have this pattern? ...
> >
> > The comment questions reproducibility of our experiments. We address this comments in a few bullet points as follows.
> >
> > - "In Figure 7b, we zoom in on the learned parameters of the first five neurons". Here "five" is a typo and it should be three. These three neurons are the first three of the total 512 neurons. That is, neurons with indices $1, 2$, and $3$.
> > - "how many neurons have this pattern? How often was this pattern observed over multiple runs?" All 512 neurons have this pattern, and this pattern appears in all our experiments, regardless the choices of of the random seed or choice of $p$.
> >
> > - "however, only 1.9% of the total neurons are represented in Figure 7a, and only 0.5859375% for Figure 7b." While it seems we only report a shockingly small subset of parameters by considering numbers like 1.9% and 0.58%, we would like to assure the reviewer that **all the neurons follow that same pattern**, **the pattern is robust**, and **our experiments are reproducable**.
> >
> > 	Note that the number of neurons is 512. We can plot Figure 7 (a) and (b) for all these 512 neurons in theory, but it will take huge space. Therefore, we only plot a (unpicked) subset of neurons. Note that due to random initialization, picking the first three or five neurons is equivalent to picking a random subset of neurons.
> >
> >
> >
> > - "In addition, I was unable to reproduce the main findings in the paper, ... learn this task." (**Reproducibility**) First, we note that All our code and experiments are publicly available for verification:  [https://anonymous.4open.science/r/modular-addition-feature-learning](https://anonymous.4open.science/r/modular-addition-feature-learning). We are happy to resolve all questions the review have about replicating our experiments using our code.
> >
> > 	In addition, our experiment setup, e.g., 2-layer MLP and one-hot encoding, is adopt from existing works (e.g., Google PAIR: [https://pair.withgoogle.com/explorables/grokking/](https://pair.withgoogle.com/explorables/grokking/). These **established baselines** have already identified the **trigonometric parameter structure is known to reliably emerge**. Our contribution is not discovering the pattern, but providing the rigorous **mechanistic interpretation** and **formal theory** for this observed phenomenon.
> >
> >
> > - **Activation Function and Theoretical Consistency.** Our conjecture on why the reviewer may not reproduce the result in our paper is the **choice of activation**. As we claim in Section 3 for empirical results, we set $P=23$ and use a two-layer neural network with width $M=512$ and **ReLU activation**; if you train from scratch with ReLU, it will **consistently converge to the trigonometric pattern**. In the later paper, we use a **quadratic activation** for theoretical analysis, especially dynamics analysis,  due to the intractable nature of the ReLU function, a switch that is **widely adopted in theoretical literature** [1] [2] . We argue the reasonability of this switch on two points: 1) **Observation 4** verifies that the activation switch **preserves the perfect accuracy**; 2) even with quadratic activation, under small initialization, during the initial stage of training, the trigonometric pattern will **fully emerge**, formalized in **Theorem 5.2**. We discuss the failure mode of quadratic activation—the emergence and subsequent disappearance of the Fourier trig pattern at the end of training—in Appendix C.2: _The failure of the quadratic activation stems from the significant disparity in growth rates among neurons due to the nature of the quadratic function. Specifically, a few neurons with more well-aligned initial phases grow faster in magnitude and come to dominate the output, leaving an insufficient number of neurons to support diversification._ But the understanding of quadratic activation during the initial stage provide a clear theoretical interpretation beyond the learned parametrization but from a training dynamics perspective explains why such trig pattern consistently emerges under different activations and random initialization.
> > - **Scope and Generalization.** Restricting our investigation to modular addition was **necessary for foundational depth**. Even in this simplified setting, the detailed **mechanistic solution** (including **phase-alignment** and **symmetry**) and the complex **training dynamics** were largely uncharacterized. Our theoretical proofs (Section 4) already demonstrate that the derived feature structure is **independent of the modulus $P$**, negating the need for exhaustive $P$-sweeps. Generalizing these specific Fourier-based findings to complex, non-arithmetic tasks like CIFAR-10 is wholly outside the scope of this essential foundational work.

---

> > > ### Author Response · Authors · 2025-11-25
> > >
> > > > The paper only explores $a+b \mod 23$, which limits the generality of the findings. Additional modular tasks should be explored to further support the findings, such as subtraction, division, and multiplication. Given that the paper explores modular arithmetic in a non-standard setup, where the input data is one-hot encoded. It is unclear how the findings will generalize to larger networks, more practical problems, or networks that use embedding layers.
> > >
> > >
> > > As we explained above,  **our setup is standard** (e.g., the same as in Google PAIR) while we provide novel empirical and theoretical insights towards building a comprehensive mechanistic interpretability of how two-layer MLP is trained to solve the task of modular addition. The number $p=23$ is not essential - our results hold for any choice of prime $p$. We believe that it might be problematic to conclude that our setup is nonstandard for two reasons:
> > >
> > > - (a) Existing works also adopt this setup, while leaving many questions open;
> > > - (b) Given that the mechanism of modular addition is far from a complete understanding, we focus on the foundamental setup with one-hot embedding of the data, **following the first principle**, and the results obtained lay the foundation for generation to more complex/realistic setups.
> > >
> > > 	Existing works studing this problem also only focus on the problem of modular addition only. It is possible that other modular operations admits different working methanisms. They are deferred to future research.
> > >
> > > >The paper ends abruptly with no concluding section and does not provide clear takeaways nor relates the findings back to the current literature.
> > >
> > > We omit the conclusion section due to space limit. We will add it in the revised version in Appendix. We believe the introduction as well as the color boxes in the main paper have provided clear takeways. We have also compared our work to the current literature in detail in Appendix A.2 and B.
> > >
> > >
> > > ### Response to Questions
> > >
> > > > Why is only p=23 explored in a network with a hidden width of 512 explored? Do these results hold when using a range of
> > >  values, i.e 29, 31, ...
> > >  and hidden widths of 32, 64, ...256?
> > >
> > >
> > > We choose $p=23$ without loss of generality. Our results also hold for any other choice of prime $p$. As for the hidden width $M$, we need it to be sufficiently large. Theoretically, our Theorem 5.2 requires that $\log M / M $ has to be small, which means that $M$ has to be sufficiently large. Empirically, we also need $M$ to be large enough such that the neurons are diversified enough (Definition 4.1) such that the majority-voting scheme is effective. This is also related to Obervation 3 - the phenomenon of _phase symmetry_ also requires $M$ to be sufficiently large.
> > >
> > > We do not have a theoretical or empirical result showing how the smallest working $M$ scales with $p$. This is a limitation of our work. But our work offers some insight for determining whether $M$ is too small -- if we run the training and observe that Phenomenon 3 does not appear, then it is very likely that $M$ is too small.
> > >
> > >
> > > > Can you explore additional modular arithmetic tasks such as subtraction, division, and multiplication? This would help improve the generality of the findings. To also help substantiate claims around phase alignment, model symmetry, and the lottery ticket mechanism can this be explored in the case of CIFAR 10 and CIFAR 100 [1].
> > >
> > > As we mentioned above, extending to other modular arithmetic tasks  is a future work, as they might require different mechanisms. Existing works also study the modular addition task as in our work, while we provide new findings that complement the literature.
> > >
> > > Moreover, we believe that question that "To also help substantiate claims around phase alignment, model symmetry, and the lottery ticket mechanism can this be explored in the case of CIFAR 10 and CIFAR 100 [1]" **is irrelavant**.  These mechanisms are only for the task of modular addition, and **we do not claim anything beyond this particular task**. This task is a foundamental for understanding feature learning of neural networks, with no particular application in mind. It seems **unfair** to expect that the insights gained from modular addition to also hold in imaging. This ignores the fact that these two tasks are very different.
> > >
> > > The fact that the reviewer asks this question, **specifically about applications to image datasets like CIFAR 10**, seem to suggest that the review is **unfamiliar with the literature** on modular addition and thus unsuitable for evaluating this work.

---

> > > > ### Author Response · Authors · 2025-11-25
> > > >
> > > > > A lot of the work is then based on the finding that the problem is solved by embedding a trigonometric structure into its parameters. Given the model is significantly overparameterized, can you explore and report what happens when the first (input) layer is frozen (completely random) during training (with all data)?
> > > >
> > > > **Frozen First Layer.** Our conjecture regarding the requested experiment (freezing the first layer) is that the model will still quickly learn a perfect accuracy solution, but this solution will **not adhere to the identified trigonometric pattern**. This is because the **phase alignment dynamics**, which we revealed in our paper as crucial for the emergence of the Fourier feature set, fundamentally **requires interaction between the layers** to correctly set the feature phases. Freezing the input weights eliminates this required dynamic and thus the learned model may not be generalizable.
> > > >
> > > >
> > > >
> > > >
> > > > [1] Tian, Yuandong. "Composing Global Solutions to Reasoning Tasks via Algebraic Objects in Neural Nets." _The Thirty-ninth Annual Conference on Neural Information Processing Systems_.
> > > > [2] Morwani, Depen, et al. "Feature emergence via margin maximization: case studies in algebraic tasks." _arXiv preprint arXiv:2311.07568_ (2023).

---

> ### Comment · Reviewer_xW8s · 2025-11-27
> **Response 1**
>
> Thank you for taking the time to respond to my weaknesses and questions.
>
> # Weakness 1 and 2
> On Weakness 1, I appreciate that the introduction is not meant to be technical. However, the simple addition of `Phase as defined in Equation 3.1` would suffice to remove any uncertainty that a reader may have. Especially given the word `phase` is used 9 times within the introduction.
>
>
> In response to Weakness 2, it was not entirely clear that `two—trigonometric parameterization` and `phase alignment` which were highlighted in blue, indicating importance and differentiation, directly related to `global trigonometric pattern` where `trigonometric pattern` is additionally highlighted in blue. Given the highlighting in blue, it was understood that the `trigonometric pattern` was something else and new, thus the statement that no empirical evidence was shown. Thank you for clarifying that the `trigonometric pattern` was in reference to `trigonometric parameterization`, I now understand and see the Figures 7(b) provides this empirical evidence of this finding and this was referencing previous work.
>
> This part can be improved by not introducing new terminology and instead continuing the use of `trigonometric parameterization` and state something along the following lines `We begin with the most striking observation from literature: a global trigonometric parameterization that consistently emerges across all training runs with random initialization.`. This improves the clarity of the writing and better indicates that this part of the paper explores work already shown in the literature, which was not initially clear.
>
> ## Surrounding Reproducibility:
> Thank you for providing the codebase. I have a better understanding of the trigonometric structure that was measured and can now state that **I can reproduce the finding that the neurons do match this structure**.
>
> However, for experiment p_23_dmlp_512_ReLU_random_scale_0.1_decay_0_08142058 (as provided in the codebase), when measuring the Pearson correlation between the actual pattern and the fitted pattern (for the first layer), that there is a mean correlation of 0.9068, a standard deviation of 0.0750 and min of 0.5223 and max of 0.9893. The min of 0.5223 is not particularly strong. For this layer the breakdown of correlation, $\rho$, is the following:
>
> - 0 neurons have a $\rho < 0.5$
>
> - 2 neurons have a $0.5 \le \rho < 0.6$
>
> - 16 neurons have a $0.6 \le \rho < 0.7$
>
> - 31 neurons have a $0.7 \le \rho < 0.8$
>
> - 71 neurons have a $0.8 \le \rho < 0.9$
>
> - 392 neurons have a $0.9 \le \rho < 1.0$
>
> This suggests that not all neurons have a strong trigonometric structure at the end of training; however, the majority do. Can you explain why some neurons do not exhibit this behavior strongly?

---

> ### Comment · Reviewer_xW8s · 2025-11-27
> **Response 2**
>
> ### Non-standard setup response
> Grokking was originally explored with Transforms [1] and has subsequently been explored with this architecture [2,3,4,5]. However, I do accept that one-hot encoding is used in literature to understand aspects of the phenomena as well. I should note that the Google Pair work is not cited within the paper. Would the authors expect the same results as found in the paper to emerge using different architectures, or is it specific to their setup?
>
> ### Response to  lack of conclusion:
>  I would like to remind the authors that a conclusion is important and integral to aiding the reader to understand the key takeaways of the paper. Providing color boxes in the main paper does not suffice to offer a conclusion or signposts of conclusions. Reading the conclusion in the paper it states `Third, we characterize grokking as a three-stage process where weight decay prunes non-feature frequencies` however [5] states that `weight decay` is not required for grokking to occur. Could the authors please explain their results considering [5] that shows weight decay is not required for grokking?
>
> ### Surrounding Scope and Generalization
>
> The aim of the question was to understand how applicable and generalizable these findings are. It is clear from the response that the methods and factors identified here are restricted to a modular addition and do not generalize to cases more complex than this. Given that it is unsure whether the work would extend to additional cases of subtraction, division or multiplication, it is hard to see the direct utility of the work, as it does not answer core questions surrounding grokking but only a specific case.  Could the authors explain how this paper improves the general understanding of grokking and how the findings are generalizable?
>
> ### Response to Model and dataset size:
>
> Thank you for responding to this question. By sufficiently large what is meant by this? Is there a number that signifies sufficiently large, at what point can this be considered? In relation to Theorem 5.2. that requires $\frac{\log(M)}{M}$ has to be sufficiently small, therefore a hidden width of 1 would suffice, as this would result in a value of 0. This suggests that Theorem 5.2, is at odds with the requirement that `we also need to be large enough such that the neurons are diversified enough (Definition 4.1) such that the majority-voting scheme is effective`. The statement that if we run it and do not observe the phenomenon of phase symmetry does not give much credence to the insights as they do not provide a dimensionality as to when and at what scale we can expect this to occur. It would be interesting to understand how and when these observations hold and for what hidden dimensions?
>
> ### Frozen Layers
>
> The response to the Frozen First Layer, is not adequate and does not answer the question. Given the conjecture that the model can perfectly learn the function without adhering to the identified trigonometric pattern, does this not suggest that these trigonometric patterns are not required for the model to learn the function? Especially since the theory is based of the training with 100% of the data. Could you additionally explain why you think that the model `may not be generalizable` when training with a frozen layer?

---

> ### Comment · Reviewer_xW8s · 2025-11-27
> **Response 3**
>
> ### Additional Comments:
>
> On Line 120 (original paper): `we initialize all parameters using PyTorch’s default method (Paszke et al., 2019), and then normalize` In the code base provided the models are initialized using:
>
> ```python
> self.W_in = nn.Parameter(self.init_scale * torch.randn(d_mlp, d_model)/np.sqrt(d_model))
> self.W_out = nn.Parameter(self.init_scale * torch.randn(d_vocab, d_mlp)/np.sqrt(d_model))
> ```
>
> However the default intilisation of Linear layer in Pytorch is:
> $\mathcal{U}(-k,k)$, where $k=\frac{1}{in\_features}$
>
> See Pytorch docs here: https://docs.pytorch.org/docs/stable/generated/torch.nn.Linear.html
>
> ### Conclusion
>
> As I have increased my score to 2, as I was able to reproduce the core findings with the aid of the author's codebase. However, I still have questions about the usefulness of the insights, how they can be leveraged to additional modular tasks, how this analysis holds when layers are frozen, and with different model sizes. Additionally, I have questions around the need for weight decay given that [5], has shown that it is not a requirement for grokking.
>
> ### References
>
>
> [1] Power, A., Burda, Y., Edwards, H., Babuschkin, I. and Misra, V., 2022. Grokking: Generalization beyond overfitting on small algorithmic datasets. arXiv preprint arXiv:2201.02177.
>
> [2] Nanda, N., Chan, L., Lieberum, T., Smith, J. and Steinhardt, J., 2023. Progress measures for grokking via mechanistic interpretability. arXiv preprint arXiv:2301.05217.
>
> [3] Lee, J., Kang, B.G., Kim, K. and Lee, K.M., 2024. Grokfast: Accelerated grokking by amplifying slow gradients. arXiv preprint arXiv:2405.20233.
>
> [4] Thilak, V., Littwin, E., Zhai, S., Saremi, O., Paiss, R. and Susskind, J., 2022. The slingshot mechanism: An empirical study of adaptive optimizers and the grokking phenomenon. arXiv preprint arXiv:2206.04817.
>
> [5] Kumar, T., Bordelon, B., Gershman, S.J. and Pehlevan, C., 2023, September. Grokking as the transition from lazy to rich training dynamics. In The twelfth international conference on learning representations.

---

### Official Review · Reviewer_Ykz3 · 2025-11-01

**Soundness:** 1
**Presentation:** 1
**Contribution:** 2
**Rating:** 2
**Confidence:** 5

**Summary:**

This paper studies 1 hidden layer, one hot encoded networks and seeks to describe the learned solution mechanistically and explain how the training dynamics result in that solution being learned.

**Strengths:**

- This work addresses a question that the field currently considers to be of high importance: why do deep neural networks learn the features they learn on modular addition?

**Weaknesses:**

*I am concerned with the paper overclaiming its novelty, particularly with respect to their claimed mechanistic interpretation*.

This paper claims multiple results are novel, but I know some were done by other published papers. Furthermore, some results claimed as novel are in disagreement with results from other published papers.

Thus, there are significant issues with this paper:

1. At least five prior works of high relevance aren't cited, which leads to 2.

2. There are **multiple claims of novelty that aren't novel**, i.e. other published work has already achieved the result:
Claimed novelty 1. "While individual neurons produce noisy signals, the phase symmetry enables a majority-voting scheme that cancels out noise, allowing the network to robustly identify the correct sum."
Claimed novelty 2. "We prove that these properties allow the network to collectively approximate an indicator function on the correct logic for the modular addition task."

Claimed novelties 1 and 2 are already known and aren't novel. They were detailed by [1], which provided a mathematical model for how networks get the correct answer, using the uniform phase assumption to cancel out noise (claimed novelty 1) to prove the correct logit becomes "dirac" (i.e. an indicator, novelty 2). I believe that Gromov's work on this topic (which is cited by this paper) also used random phase cancellation to prove the correct logit would be like a dirac indicator, but I am more familiar with [1] which I know did this.

Also, claims 1 and 2 disagree with empirical and theoretical results in [2], which shows that in 2-layer networks the logits are **not an indicator on the correct output logit**. [2] gives both empirical evidence and a theoretical proof that on average, O(log(n)) different frequencies are learned, and the size of the margins as a function of frequencies that are learned is O(log(n)). Thus, in multilayer networks, the margins are not an indicator.

3. a lack of scope (the authors only study 1 hidden layer networks, though this is unclear from a first reading of their paper, which claims they study 2 layer networks, which can't be the case due to it being already established and proven that networks with >= 2 hidden layers learn O(log(n)) frequencies to solve this task [2]. 1 layer networks learn *all* (p-1)/2 frequencies (Morwani et al.), while multi-layer networks have been observed to learn substantially fewer (Nanda et al., Chughtai et al., [2]), and the gradient dynamics explaining why this happens are considered an open problem. Only studying 1 hidden layer networks makes their results fail to generalize to multilayer networks, and can't explain the aforementioned open problem.

4. False claims are made, for example, on line 99, the authors state either one hot encoded inputs or a trainable embedding matrix can be used, but switching from one hot encodings to a trainable embedding causes the network to learn O(log(n)) frequencies [2], and not the n-1/2 frequencies result of Morwani et al., were trainable embeddings to be used, observation 3 and definition 4.1 both become false.

Less significant, but still issues:

5. This paper claims a "full" mechanistic understanding of models trained on modular addition, and presents 6 empirical observations, but lacks convincing empirical evidence supporting the mechanistic interpretation and observations. Some of this empirical evidence is relegated to the appendix (but should be adjacent to the observations and in the main paper). The experiments remain unconvincing due to reasons like: experiments not over multiple random seeds, some experiments seem to require training on the entire dataset (what do the plots in the first section look like when using standard ML train test splits?), lack of quantitative statistical / causal testing supporting observations 1-6.

6. This paper spends a significant amount of space claiming the aforementioned "full" mechanistic understanding (claims 1 and 2). This space should instead be used to incorporate convincing experimental evidence supporting what's necessary for the main result of the paper: their claims about how training dynamics unfold.

7. This work does not have a related work section, as it's located in the appendix. Furthermore, this work incorrectly claims a "complete" discussion of related work, while missing citations to at least 5 other works ([1,2,3,4,5]). The authors state: "A complete discussion on related works is deferred to A.2 due to space limit."

8. There is no limitations section or conclusion; the last section is titled "TRAINING DYNAMICS FOR FEATURE EMERGENCE".

In summary, this paper makes multiple claims of novelty where the result is already well known and established. Of particular note, is the claim of a "full" mechanistic understanding, but [1,3], and especially [2] all together provide a more complete understanding than what is presented in this paper. **I believe this paper is not ready for publication at this time and needs a rewrite and restructuring beyond the scope of what can occur during reviews.**

That said, **there are open problems remaining related to the gradient training dynamics on modular addition**: if their claim for the training dynamics holds up robustly under quantitative analyses with depth, over many random seeds, then their work would be the first paper (to my knowledge) to resolve *how* networks learn the features on modular addition. [5] is an uncited paper that attempted to explain the gradient dynamics on modular addition and was (to my knowledge) only accepted at a workshop. It used lotka-volterra ODEs to attempt their arguments.

A successful paper on the gradient dynamics is worthy of publication, without needing to claim novel mechanistic interpretations.

[1] Grokking modular polynomials https://arxiv.org/pdf/2406.03495

[2] Uncovering a Universal Abstract Algorithm for Modular Addition in Neural Networks
 https://arxiv.org/abs/2505.18266

[3] Towards a unified and verified understanding of group-operation networks https://arxiv.org/abs/2410.07476

[4] Modular addition without black-boxes: Compressing explanations of MLPs that compute numerical integration https://arxiv.org/abs/2412.03773

[5] Survival of the Fittest Representation: A Case Study with Modular Addition https://openreview.net/forum?id=2WfiYQlZDa

**Questions:**

Q1. Is my understanding correct that you trained networks with one hot encoded inputs and 1 hidden layer, i.e. the same networks trained by Morwani et al.?

Q2: How is this different from the Survival of the fittest work, which also uses ODEs (Lotka Volterra) [5]?

---

> ### Author Response · Authors · 2025-11-25
>
> We thank Reviewer Ykz3 for the detailed feedbacks. We would like to point out that the reviewer's claims regarding non-novelty and disagreement with prior work are based on some confusion. We would like to ellaborate on that aspect as follows.
>
> ---
>
> ## Response to Weaknesses
>
> >1. At least five prior works of high relevance aren't cited, which leads to 2.
>
> Thanks for pointing out the missing related work and we added them in the revised version
>
> >2. There are multiple claims of novelty that aren't novel, i.e. other published work has already achieved the result...."
>
>
> We maintain that our claims represent a genuine advance over previous work due to their detailed quantitative and rigorous nature, the specifics of which will be detailed in the following response. Regarding noise cancellation and phase symmetry, we are the first to a) **quantitatively** characterize how diversified frequencies and phases cancel out noise and b) identify the **minimal symmetry condition** for phases, which is essential for **finite-neuron** analysis, rather than relying on uniform phase distributions  (Grokking modular polynomials https://arxiv.org/pdf/2406.03495). Building on this, we c) **rigorously prove** that these properties allow a finite-neuron network to approximate the indicator function after softmax using the **learned parametrization**. Finally, d) we experimentally verify all preceding conditions under an overparameterized regime ($M \gg p$)  and rigorously prove that these phenomena arise from the dynamics of **gradient-based training**, moving beyond analyses focused only on the representation results after convergence.
>
>
> > Claimed novelties 1 and 2 are already known and aren't novel. They were detailed by [1], which provided a mathematical model for how networks get the correct answer, using the uniform phase assumption to cancel out noise (claimed novelty 1) to prove the correct logit becomes "dirac" (i.e. an indicator, novelty 2). I believe that Gromov's work on this topic (which is cited by this paper) also used random phase cancellation to prove the correct logit would be like a dirac indicator, but I am more familiar with [1] which I know did this.
>
> The reviewer states that [1] (Grokking modular polynomials) "provided a mathematical model for **how networks get the correct answer**, using the uniform phase assumption to cancel out noise". This argument is **not entirely correct**. Section 2.1 in [1] **provides a construction** of a two-layer neural network that solves modular addition, where the weights are sinusoidal and the **phases are i.i.d. and uniform**. But this is **merely a construction result**. It neither guarantees that **gradient training will find this specific solution** nor establishes the **necessary conditions** for such uniformity to actually occur. To narrow this gap, [1] empirically argue the learned network is similar to the constructed one by looking into the loss and prediction accuracy. **But there is no comparison of neural network parameters that substantiate this claim**.
>
> In contrast, our work empirically show that, after gradient-based training, symmetric phases (uniformly distributed phases) appear (Observation 2). Motivated by this observation, with **finite network width**, we establish a much weaker technical condition than uniformity, i.e., Definition 4.1, finite sum $\sum_{m} \exp(i \cdot \iota \phi_m) = 0$ for harmonics $\iota \in \{2, 4\}$, showing that as long as the $M$ phases are sufficiently diverse, **gradient-based training provably learns a two-layer neural network that solves modular addition** and the noise can be cancelled out. By saying [1] shows "how networks get the correct answer", the reviewer vaguely insinuates that [1] has gradient-based training results, but this is not true.
>
>
> As for claim 2 -- "We prove that these properties allow the network to collectively approximate an indicator function on the correct logic for the modular addition task" -- we note that by the construction in [1], the resulting network implments a **modular Kronecker Delta function**, which is Kronecker Delta function  up to
> integer multiplies of the modular base $p$. That is,
> $$
> \delta^{p}(x) = \begin{cases} 1 & x = r \cdot p , r \in \mathbb{Z} \\
> 0 & \mathrm{otherwise}
> \end{cases}.
> $$
> This work shows that the **constructed network itself is equal to a modular Kronecker delta function** by direct calculation. And again, this is a construction/expressivity result. In contrast, we prove that gradient-based training **provably learns** a desired network, which uses phase symmetry (cancellation) to build an aggregated predictor that is close to an indicator (modular Kronecker Delta). Moreover, such an indicator approximation argument is based on **both frequency and phase diversification**, whose proof is different from that in [1]. See Proposition 4.2 for details.
>
> **In summary, both these two critical claims are novel in our work and are not covered by [1].**

---

> ### Author Response · Authors · 2025-11-25
>
> (continuing)
>
> Moreover, in the following, we further discuss these two claims with more technical depth.
>
> 1. **Demystifying Phase Symmetry: The "Balanced Vector" Argument.**
>     The reviewer suggests our findings are standard because prior work assumes uniform phases to cancel noise. This misses a critical distinction between *infinite* and *finite* widths.
>     * **Concrete Difference:** Prior mean-field theories (Wang & Wang, 2025) require the phases to form a **perfect continuous circle** (a uniform probability density) to integrate noise to zero. This effectively requires infinite width ($N \to \infty$).
>     * **Our Finite Solution:** We prove that perfect uniformity is unnecessary. Our **Definition 4.1** shows the network only needs the **discrete vector sums** of **specific harmonics** to cancel out, which is in essence a much weaker and more practical condition.
>     * **Specifically:** Think of the phases as weights on a wheel. You don't need a continuous tire (infinite neurons) to be balanced; you just need the weights to be spaced such that (i) their center of mass is zero. Specifically, we prove that if the finite sum $\sum_{m} \exp(i \cdot \iota \phi_m) = 0$ for harmonics $\iota \in \{2, 4\}$, the noise vanishes with quadratic activation.
>     * **Evidence:** **Figure 1b (right table)** provides concrete empirical proof of this discrete cancellation. The average values of $\cos(\iota \phi_m)$ and $\sin(\iota \phi_m)$ for the trained network are not just "random"—they are remarkably close to zero, confirming that the network learns this specific "balancing act" with finite neurons.
>
> We have revised the manuscript to clarify the novelty.
>
>
> 2. **The Mechanics of the Indicator: Signal vs. Noise Peaks.**
>     The reviewer claims our "indicator function" finding contradicts prior work showing logits are noisy. **Proposition 4.2** resolves this by quantifying exactly *how* the indicator emerges from the noise.
>     * **The Equation:** We derive the **exact** form of the logit $f(x,y)[j]$ in Proposition (4.2):
>      $$
>      \frac{f[j]}{N} \propto -1 +\underbrace{\frac{p}{2}\mathbb{1}(x+y \\mod p=j)}_{\text{Huge Signal}} + \underbrace{\frac{p}{4}(\mathbb{1}(2x\mod p=j) + \mathbb{1}(2y\mod p=j))}_{\text{Small Spurious Noise}}
>      $$
>     * **The Concrete Mechanism:** This equation reveals that while there *are* spurious noise peaks at $2x$ and $2y$ (as noted in prior work), the **Signal-to-Noise gap** is massive, scaling with the network width $N$ and modulus $p$.
>     * **Result:** Because the Softmax function is exponential, this linear gap ($\Theta(aNp)$) suppresses the smaller noise peaks at $2x, 2y$ to near zero, forcing the output to become a "Dirac" (indicator) function on the correct answer. **Figure 4** visually confirms this: the diagonal (correct signal) is bright red/intense, while the spurious peaks are faint background noise, exactly as predicted by our equation. In comparison, as far as we know, previous results stay in writing the predictor as a **sum of cosine functions**, making the essence of neuron cancellation unknown.

---

> > ### Author Response · Authors · 2025-11-25
> >
> > > Also, claims 1 and 2 disagree with empirical and theoretical results in [2], which shows that in 2-layer networks the logits are not an indicator on the correct output logit. [2] gives both empirical evidence and a theoretical proof that on average, O(log(n)) different frequencies are learned, and the size of the margins as a function of frequencies that are learned is O(log(n)). Thus, in multilayer networks, the margins are not an indicator.
> >
> > - **Our Architecture:** We would like to clarify that throughout our paper, we study a **2-layer MLP** (one hidden layer) with **one-hot encoded inputs**, trained with both full dataset which matches the setting analyzed in Morwani et al. (2023) and further in Tian (2024), and also dataset with train-test splitting (Nanda et al. (2023)). Under the full dataset setting, the nn is expected to reliably converge to a solution utilizing **all $k \in \{1, \dots, (P-1)/2\}$ frequencies**, given a sufficient neuron budget.
> >
> > - **On $O(\log N)$ Frequency Behavior in [2]:** The $O(\log N)$ frequency result arises from **deep architectures** (e.g., deep Transformers or MLPs with **trainable embeddings**) where the solution exhibits a **low-rank structure**. We did not study these architectures in our work, as our primary focus is to obtain a systematic and clear mechanistic understanding of the 2-layer MLP case. The low-rank solution in deep architectures is likely learned due to **implicit regularization** effects of deep neural networks (Timor et al., 2023), and is outside the scope of our current analysis, and is an important direction for future work.
> >
> > 	- In addition, [2] empirically validates that two-layer (1 hidden layer) NN, after training, finds all $(p-1)/2$ frequencies, which corroborates our finding. Similar results are also reported in [6][7].
> > 	- **When $O(\log N)$ Frequency Occurs: Interplay of Hyperparameters.** We acknowledge the finding in [2] that, using trainable embeddings or deeper MLPs, networks may learn only a handful, or $O(\log N)$, of frequencies (as shown in Figure 4 of [2]). However, we argue that their experimental scope is **not comprehensive**. Our own extensive experiments reveal that the conclusion regarding the emergence of a low-rank frequency structure is determined by a complex **interplay between the modulus $p$, the neural network width $M$, the initial scale**, and the **embedding dimension $d_{\mathrm{ebd}}$**. Generally, the low-rank structure (i.e., $O(\log N)$ frequencies) is more likely to occur when 1) The ratio **$p/M$ is small**, 2)The **initial scale is not very small**, and 3) The **embedding dimension $d_{\mathrm{ebd}}$ is small**.
> >
> > 		Crucially, in the specific setup of our paper—a sufficiently wide two-layer neural network with **small initialization**—we observe that **all $(p-1)/2$ frequencies are learned**, even with trainable embeddings. While we do not yet have a conclusive theory fully governing this conditional behavior, exploring this full dependency is an important direction for future research. In the revised version, we have **removed the description of trainable embeddings** to eliminate potential confusion. We want to unequivocally highlight that all of our experimental results are based on the use of one-hot embeddings, ensuring consistency and clarity regarding the architectural setup used throughout our analysis. Moreover, theoretical results in [2] are also about existence, leaving the question whether gradient-based training finds the desired solution open. In contrast, we provide positive gradient-based training result that close this gap in the simpler two-layer setting.

---

> > > ### Author Response · Authors · 2025-11-25
> > >
> > > > 3. a lack of scope (the authors only study 1 hidden layer networks, though this is unclear from a first reading of their paper, which claims they study 2 layer networks, which can't be the case due to it being already established and proven that networks with >= 2 hidden layers learn O(log(n)) frequencies to solve this task [2]. 1 layer networks learn all (p-1)/2 frequencies (Morwani et al.), while multi-layer networks have been observed to learn substantially fewer (Nanda et al., Chughtai et al., [2]), and the gradient dynamics explaining why this happens are considered an open problem. Only studying 1 hidden layer networks makes their results fail to generalize to multilayer networks, and can't explain the aforementioned open problem.
> > >
> > > The first sentence of abstract states that "We present a comprehensive analysis of how two-layer neural networks learn features to solve the modular addition task". So we are clear that we use two-layer neural networks. Moreover, it is the convention that **"two-layer NN" is the same as "one-hidden-layer NN"**, not "two-hidden-layer NN". Two-layer NN simply means NNs with two weight matrices, and thus only has one hidden layers. See, e.g., https://francisbach.com/gradient-descent-neural-networks-global-convergence/ for the definition.  This confusion might led the reviewer think that our results are about deep neural networks.
> > >
> > >
> > > As we mentioned about, the claim that deeper MLPs learn only $O(\log n)$ features are not rigorously substatiated by [2], because it relies on extensively testing various choices of network architectures, e.g., width and depth. This is not arguely in the experiment in [2] and only established theoretically in terms of existence. In contrast, we focus on the two-layer simpler model and demystifies **how gradient-based training learns** a model that solves modular addition and **what this predictor is**. This is an open question before our work.
> > >
> > > Following the first principle, we should first answer the simpler question, i.e., the two-layer NN model. It is indeed possible that more complex models solve this task using a different mechanism. This seems slightly out of the scope of our work and should be left for future work because the analysis of gradient-based training will be different.

---

> > > > ### Author Response · Authors · 2025-11-25
> > > >
> > > > > 4. False claims are made, for example, on line 99, the authors state either one hot encoded inputs or a trainable embedding matrix can be used, but switching from one hot encodings to a trainable embedding causes the network to learn O(log(n)) frequencies [2], and not the n-1/2 frequencies result of Morwani et al., were trainable embeddings to be used, observation 3 and definition 4.1 both become false.
> > > >
> > > > As we stated above, the $O(\log N)$ claim is theoretical and only about existence. The experiments in [2] do not substantiate this quantatitive claim. The concrete claim in their experimental part is that, when training a two-layer NN  with learnable embedding matrices, only a handful frequencies are found.
> > > > But according to our experiments, this claim involves a complicated interplay between $p$ and $M$ -- the number of frequencies found depends on the size of $M$. We tested with a large $M$ and found all frequences with trainable embedding matrices. When $M$ is smaller, we indeed observe that not all frequencies are found. **We have revised the related claim to make it mroe rigorous.** Moreover, we note that our main theory and experiment results are not about this setup -- we mainly focus on the one-hot embeedding case.
> > > >
> > > > > 5. This paper claims a "full" mechanistic understanding of models trained on modular addition, and presents 6 empirical observations, but lacks convincing empirical evidence supporting the mechanistic interpretation and observations. Some of this empirical evidence is relegated to the appendix (but should be adjacent to the observations and in the main paper). The experiments remain unconvincing due to reasons like: experiments not over multiple random seeds, some experiments seem to require training on the entire dataset (what do the plots in the first section look like when using standard ML train test splits?), lack of quantitative statistical / causal testing supporting observations 1-6.
> > > >
> > > > We note that all our code and experiments are publicly available for verification:  [https://anonymous.4open.science/r/modular-addition-feature-learning](https://anonymous.4open.science/r/modular-addition-feature-learning). We have conduced extensive experiments with various randomseeds, network sizes, and values of $p$. The emprical observations are stable. We include results with $p = 47$ and $M=1024$ in Appexndix I to illustrate such robustness. All of our reported empirical observations are substantiated by experiments, and we include plots showing them respectively.
> > > >
> > > > Moreover, our empirical results do not rely on having the full data -- they are valid with train-test split. However, the assumption of full-data training is needed only for the theory of gradient-based training. The main reason is that we need the symmetry of the $p^2$ data points to simplify the dynamics and make them tractable.

---

> > > > > ### Author Response · Authors · 2025-11-25
> > > > >
> > > > > (continuing)
> > > > >
> > > > > In the following, we further clarify the empirical observations.
> > > > >
> > > > >
> > > > > **1. Quantitative Validation of Learned Features (Observations 1–3)**
> > > > > The reviewer claims our observations are qualitative. On the contrary, we employ precise statistical metrics to quantify structural emergence across all neurons ($M=512$).
> > > > > * **Observation 1 (Fourier Sparsity):** We do not rely on visual inspection. We quantify "single-frequency" emergence using the **Inverse Participation Ratio (IPR)** of the Fourier coefficients. As shown in **Figure 3d**, the IPR grows and stabilizes, providing a quantitative measure of how the network concentrates energy into specific frequencies over time.
> > > > > * **Observation 2 (Phase Alignment):** We validate the $2\phi_m \approx \psi_m$ relationship globally. **Figure 1a** provides a scatter plot of $(2\phi_m, \psi_m)$ for all neurons, showing they lie precisely on the $y=x$ line. We further quantify this alignment error over training in **Figure 3c**, tracking the metric $|\sin(\mathcal{D}^*_m)|$ (where $\mathcal{D}^*_m$ is the phase difference) as it converges to zero.
> > > > > * **Observation 3 (Symmetry):** We test the "uniform distribution" hypothesis numerically. **Figure 1b (table)** reports the average values of harmonic moments (e.g., $\cos(\iota \phi_m)$). Values such as **0.0123** and **-0.0500** confirm that the phases cancel out statistically, satisfying the "vector sum" condition required for our mechanistic proof.
> > > > >
> > > > > **2. Causal Testing of Dynamics (Observations 5–6)**
> > > > > The reviewer suggests our dynamics are merely observational. We provide **causal** evidence by manipulating initial conditions to predict outcomes, supported by theoretical guarantees.
> > > > > * **Observation 6 (Lottery Ticket):** We rigorously test the causal link between initialization and the final learned frequency. **Figure 2b** presents a phase diagram showing that the "winning" frequency is deterministically predicted by the initial magnitude $\beta^k(0)$ and phase misalignment $\mathcal{D}^k(0)$.
> > > > > * **Theoretical Guarantee:** This is not a random seed artifact. **Theorem 5.2** and **Corollary C.1**  prove that under small initialization, the frequency with the smallest misalignment *must* dominate exponentially. The "competition" is a deterministic outcome of the gradient flow equations we derived, not a stochastic fluke.
> > > > >
> > > > > **3. Robustness Testing (Observation 4)**
> > > > > * **Observation 4 (Activation Invariance):** To address concerns about model specificity, **Table 1** provides quantitative testing of **9 different activation functions** (including $x^2, x^4, e^x$, etc.). The model achieves 100% accuracy and near-zero loss across these variations, proving the mechanism relies on the even-order harmonics of the activation, not the specific ReLU nonlinearity.
> > > > >
> > > > > **4. Clarification on Dataset Usage**
> > > > > * **Full vs. Split Data:** The reviewer conflates our *mechanistic characterization* (Sections 3.1, 4, 5) with our *generalization analysis* (Section 3.2). We intentionally use the full dataset in early sections to isolate the **converged solution structure** (the "what") without the noise of sampling error. However, **Section 3.2** explicitly addresses **Grokking** using a standard 75/25 train-test split, documenting the transition from memorization to generalization (see **Figure 3a, 3b**).

---

> > > > > > ### Author Response · Authors · 2025-11-25
> > > > > >
> > > > > > > 6. This paper spends a significant amount of space claiming the aforementioned "full" mechanistic understanding (claims 1 and 2). This space should instead be used to incorporate convincing experimental evidence supporting what's necessary for the main result of the paper: their claims about how training dynamics unfold.
> > > > > >
> > > > > > As we explaiend above, claims 1 and 2 are our genuine novel contribution, and they contribute to the theoretical results of gradient-based training. These claims are important because they help to answer these two questions: (a) What is the predictor implemented  by the learned neural network that solves the  modular addition task? (b) How does gradient-based training finds this predictor? We need to first answer (a) and then leverage (a) to show (b).
> > > > > >
> > > > > >
> > > > > > > 7. This work does not have a related work section, as it's located in the appendix. Furthermore, this work incorrectly claims a "complete" discussion of related work, while missing citations to at least 5 other works ([1,2,3,4,5]). The authors state: "A complete discussion on related works is deferred to A.2 due to space limit."
> > > > > >
> > > > > >
> > > > > >
> > > > > > Thanks for pointing out the missing related work and we added them in the revised version.
> > > > > >
> > > > > >
> > > > > > > 8. There is no limitations section or conclusion; the last section is titled "TRAINING DYNAMICS FOR FEATURE EMERGENCE".
> > > > > >
> > > > > >
> > > > > > We have added a temporary conclusion section in the appendix and **will move it to the main body** if the paper is accepted.
> > > > > >
> > > > > >
> > > > > > > That said, there are open problems remaining related to the gradient training dynamics on modular addition: if their claim for the training dynamics holds up robustly under quantitative analyses with depth, over many random seeds, then their work would be the first paper (to my knowledge) to resolve how networks learn the features on modular addition. [5] is an uncited paper that attempted to explain the gradient dynamics on modular addition and was (to my knowledge) only accepted at a workshop. It used lotka-volterra ODEs to attempt their arguments.
> > > > > >
> > > > > >
> > > > > >
> > > > > > We appreciate it that the reviewer acknowledge that gradient training dynamics on modular addition task is an open problem. Our work aims to solve this open problem in the context of two-layer neural network. Our experiments are robust to random seed, $M$, and $p$, as long as $M$ is sufficiently large. These experiments show that the neural network learns a full-frequencies solution, and we prove that gradient-based training provably finds this solution. More concretely, we make the following contributions:
> > > > > >
> > > > > > - A **neuron-wise, closed-form characterization** of learning dynamics for this architecture, distinct from the mean-field (infinite-neuron) analyses found in prior work (e.g., Wang & Wang, 2025), where taking the limit of infinite neurons simplifies the analysis of majority voting.
> > > > > > - A rigorous mechanistic explanation of **_how_** the model converges to this full-frequency solution via ODE dynamics in the frequency domain. Our analysis reveals that different frequencies within a single neuron **compete** during training, ultimately leading each neuron to specialize in a single frequency.
> > > > > > - A detailed elucidation of the "lottery ticket" mechanism, demonstrating how random initialization determines **_which_** specific frequency survives in each neuron to dominate the final representation. We validate this empirically in Figure 2, showing how initial phase mismatch and magnitude influence growth dynamics and final frequency selection.
> > > > > >
> > > > > > ## Response to Questions
> > > > > >
> > > > > >
> > > > > >
> > > > > >
> > > > > > Q1. Architecture Confirmation
> > > > > >
> > > > > > Yes, your understanding is correct. We study networks with **one-hot encoded inputs and one hidden layer** (a 2-layer MLP), which is the same setup that reliably produces the full Fourier solution, consistent with work by Morwani et al.
> > > > > >
> > > > > > Q2. Difference from Survival of the Fittest Work (Lotka-Volterra ODEs)
> > > > > >
> > > > > > Our work differs fundamentally from the Lotka-Volterra approach in the analysis of training dynamics: The Lotka-Volterra model is a **simplified population modeling tool** that tracks feature survival abstractly. In contrast, our approach tracks the **exact gradient flow**—a continuous version of Gradient Descent. By performing a **full theoretical analysis**, we show that if the step size is small enough, our theory can **exactly predict what will happen during the training process**, tracking the evolution of the specific Fourier parameters (phases and magnitudes) themselves. This provides a detailed, causal explanation of the feature selection, unlike a generalized ODE model.
> > > > > >
> > > > > > [6] Tian, Yuandong. "Composing Global Solutions to Reasoning Tasks via Algebraic Objects in Neural Nets." _The Thirty-ninth Annual Conference on Neural Information Processing Systems_.
> > > > > >
> > > > > > [7] Morwani, Depen, et al. "Feature emergence via margin maximization: case studies in algebraic tasks." _arXiv preprint arXiv:2311.07568_ (2023).

---

### Author Response · Authors · 2025-11-12
**Reviews unavailable**

Dear Area Chair,

Many thanks for your efforts in coordinating the review process. We’re writing to check on the availability of the reviews for Submission 20794. They don’t appear in the system yet on our side.
Could you advise on the expected timeline or any steps we should take?

Authors of Submission 20794

---

### Meta-Review · Area_Chair_C7rR · 2026-01-06

**Summary:**

This work is a nice and seemingly well-written paper about mechanistic interpretability. All reviewers have concerns about the novelty of this paper, including the only positive reviewer (after seeing the reviews from other reviewers). This makes it very difficult for me to recommend acceptance, given the low score and unanimous criticisms.

It would have been nice if I were capable of determining (1) how novel the contribution is; (2) if they are novel, how significant they are. Both are unfortunately impossible due to limited time and energy. For point (1), it does look like most of the insights are not novel, even if the authors go a bit deeper into the theory. For point (2), it is unclear how these deeper technical issues are significant. At best, they only apply to two-layer networks (and after many empirically justified assumptions). Even if the authors proved something extremely strong about two-layer networks, it is still debatable whether that is a sufficiently significant contribution, and I think that is better left to a very serious full-round review.

Another problem is that the paper seems to cover way too many things, and I cannot find a single strong enough message to support it. The paper is primarily empirical, but the authors seem to claim that the main contribution is theoretical in the rebuttal. Even if I personally like this paper, I think it is much better to leave this paper to another round of serious reviews.

**Reviewer Concerns:**

Some attempts are made to address the concerns, but I feel they are not convincing enough to raise scores.

**Reviewer Scores:**

I do not think they will change their opinion.

---

### Decision · Program_Chairs · 2026-01-26

Reject